# Graphons, mergeons, and so on!

**Justin Eldridge**     **Mikhail Belkin**     **Yusu Wang**
The Ohio State University
{eldridge, mbelkin, yusu}@cse.ohio-state.edu

## Abstract

In this work we develop a theory of hierarchical clustering for graphs. Our modeling assumption is that graphs are sampled from a graphon, which is a powerful and general model for generating graphs and analyzing large networks. Graphons are a far richer class of graph models than stochastic blockmodels, the primary setting for recent progress in the statistical theory of graph clustering. We define what it means for an algorithm to produce the "correct" clustering, give sufficient conditions in which a method is statistically consistent, and provide an explicit algorithm satisfying these properties.

## 1   Introduction

A fundamental problem in the theory of clustering is that of defining a cluster. There is no single answer to this seemingly simple question. The right approach depends on the nature of the data and the proper modeling assumptions. In a statistical setting where the objects to be clustered come from some underlying probability distribution, it is natural to define clusters in terms of the distribution itself. The task of a clustering, then, is twofold – to identify the appropriate cluster structure of the distribution and to recover that structure from a finite sample. Thus we would like to say that a clustering is good if it is in some sense close to the ideal structure of the underlying distribution, and that a clustering method is *consistent* if it produces clusterings which converge to the true clustering, given larger and larger samples. Proving the consistency of a clustering method deepens our understanding of it, and provides justification for using the method in the appropriate setting.

In this work, we consider the setting in which the objects to be clustered are the vertices of a graph sampled from a *graphon* – a very general random graph model of significant recent interest. We develop a statistical theory of graph clustering in the graphon model; To the best of our knowledge, this is the first general consistency framework developed for such a rich family of random graphs. The specific contributions of this paper are threefold. First, we define the clusters of a graphon. Our definition results in a graphon having a tree of clusters, which we call its *graphon cluster tree*. We introduce an object called the *mergeon* which is a particular representation of the graphon cluster tree that encodes the heights at which clusters merge. Second, we develop a notion of consistency for graph clustering algorithms in which a method is said to be consistent if its output converges to the graphon cluster tree. Here the graphon setting poses subtle yet fundamental challenges which differentiate it from classical clustering models, and which must be carefully addressed. Third, we prove the existence of consistent clustering algorithms. In particular, we provide sufficient conditions under which a graphon estimator leads to a consistent clustering method. We then identify a specific practical algorithm which satisfies these conditions, and in doing so present a simple graph clustering algorithm which provably recovers the graphon cluster tree.

**Related work.** Graphons are objects of significant recent interest in graph theory, statistics, and machine learning. The theory of graphons is rich and diverse; A graphon can be interpreted as a generalization of a weighted graph with uncountably many nodes, as the limit of a sequence of finite graphs, or, more importantly for the present work, as a very general model for generating

unweighted, undirected graphs. Conveniently, any graphon can be represented as a symmetric, measurable function $W : [0,1]^2 \to [0,1]$, and it is this representation that we use throughout this paper.

The graphon as a graph limit was introduced in recent years by [16], [5], and others. The interested reader is directed to the book by Lovász [15] on the subject. There has also been a considerable recent effort to produce consistent estimators of the graphon, including the work of [20], [8], [2], [18], and others. We will analyze a simple modification of the graphon estimator proposed by [21] and show that it leads to a graph clustering algorithm which is a consistent estimator of the graphon cluster tree.

Much of the previous statistical theory of graph clustering methods assumes that graphs are generated by the so-called *stochastic blockmodel*. The simplest form of the model generates a graph with $n$ nodes by assigning each node, randomly or deterministically, to one of two communities. An edge between two nodes is added with probability $\alpha$ if they are from the same community and with probability $\beta$ otherwise. A graph clustering method is said to achieve *exact recovery* if it identifies the true community assignment of every node in the graph with high probability as $n \to \infty$. The blockmodel is a special case of a graphon model, and our notion of consistency will imply exact recovery of communities.

Stochastic blockmodels are widely studied, and it is known that, for example, spectral methods like that of [17] are able to recover the communities exactly as $n \to \infty$, provided that $\alpha$ and $\beta$ remain constant, or that the gap between them does not shrink too quickly. For a summary of consistency results in the blockmodel, see [1], which also provides information-theoretic thresholds for the conditions under which exact recovery is possible. In a related direction, [4] examines the ability of spectral clustering to withstand noise in a hierarchical block model.

**The density setting.** The problem of defining the underlying cluster structure of a probability distribution goes back to Hartigan [12] who considered the setting in which the objects to be clustered are points sampled from a density $f : \mathcal{X} \to \mathbb{R}^+$. In this case, the *high density clusters* of $f$ are defined to be the connected components of the upper level sets $\{x : f(x) \geq \lambda\}$ for any $\lambda > 0$. The set of all such clusters forms the so-called *density cluster tree*. Hartigan [12] defined a notion of consistency for the density cluster tree, and proved that single-linkage clustering is *not* consistent. In recent years, [9] and [14] have demonstrated methods which *are* Hartigan consistent. [10] introduced a distance between a clustering of the data and the density cluster tree, called the *merge distortion metric*. A clustering method is said to be *consistent* if the trees it produces converge in merge distortion to density cluster tree. It is shown that convergence in merge distortion is stronger than Hartigan consistency, and that the method of [9] is consistent in this stronger sense.

In the present work, we will be motivated by the approach taken in [12] and [10]. We note, however, that there are significant and fundamental differences between the density case and the graphon setting. Specifically, it is possible for two graphons to be equivalent in the same way that two graphs are: up to a relabeling of the vertices. As such, a graphon $W$ is a representative of an equivalence class of graphons modulo appropriately defined relabeling. It is therefore necessary to define the clusters of $W$ in a way that does not depend upon the particular representative used. A similar problem occurs in the density setting when we wish to define the clusters not of a single density function, but rather of a *class* of densities which are equal almost everywhere; Steinwart [19] provides an elegant solution. But while the domain of a density is equipped with a meaningful metric – the mass of a ball around a point $x$ is the same under two equivalent densities – the ambient metric on the vertices of a graphon is not useful. As a result, approaches such as that of [19] do not directly apply to the graphon case, and we must carefully produce our own. Additionally, we will see that the procedure for sampling a graph from a graphon involves latent variables which are in principle unrecoverable from data. These issues have no analogue in the classical density setting, and present very distinct challenges.

**Miscellany.** Due to space constraints, most of the (rather involved) technical details are in the appendix. We will use $[n]$ to denote the set $\{1, \ldots, n\}$, $\triangle$ for the symmetric difference, $\mu$ for the Lebesgue measure on $[0,1]$, and bold letters to denote random variables.

## 2 The graphon model

In order to discuss the statistical properties of a graph clustering algorithm, we must first model the process by which graphs are generated. Formally, a *random graph model* is a sequence of random variables $\mathbf{G}_1, \mathbf{G}_2, \ldots$ such that the range of $\mathbf{G}_n$ consists of undirected, unweighted graphs with node set $[n]$, and the distribution of $\mathbf{G}_n$ is invariant under relabeling of the nodes – that is, isomorphic graphs occur with equal probability. A random graph model of considerable recent interest is the *graphon model*, in which the distribution over graphs is determined by a symmetric, measurable function $W : [0, 1]^2 \to [0, 1]$ called a *graphon*. Informally, a graphon $W$ may be thought of as the weight matrix of an infinite graph whose node set is the continuous unit interval, so that $W(x, y)$ represents the weight of the edge between nodes $x$ and $y$.

Interpreting $W(x, y)$ as a probability suggests the following graph sampling procedure: To draw a graph with $n$ nodes, we first select $n$ points $\mathbf{x}_1, \ldots, \mathbf{x}_n$ at random from the uniform distribution on $[0, 1]$ – we can think of these $\mathbf{x_i}$ as being random "nodes" in the graphon. We then sample a random graph $\mathbf{G}$ on node set $[n]$ by admitting the edge $(i, j)$ with probability $W(\mathbf{x}_i, \mathbf{x}_j)$; by convention, self-edges are not sampled. It is important to note that while we begin by drawing a set of nodes $\{\mathbf{x_i}\}$ from the graphon, the graph as given to us is labeled by integers. Therefore, the correspondence between node $i$ in the graph and node $\mathbf{x_i}$ in the graphon is latent.

It can be shown that this sampling procedure defines a distribution on finite graphs, such that the probability of graph $G = ([n], E)$ is given by

$$\mathbb{P}_W(\mathbf{G} = G) = \int_{[0,1]^n} \prod_{(i,j)\in E} W(x_i, x_j) \prod_{(i,j)\notin E} \left[ 1 - W(x_i, x_j) \right] \prod_{i\in[n]} dx_i. \qquad (1)$$

For a fixed choice of $x_1, \ldots, x_n \in [0, 1]$, the integrand represents the likelihood that the graph $G$ is sampled when the probability of the edge $(i, j)$ is assumed to be $W(x_i, x_j)$. By integrating over all possible choices of $x_1, \ldots, x_n$, we obtain the probability of the graph.

A very general class of random graph models may be represented as graphons. In particular, a random graph model $\mathbf{G}_1, \mathbf{G}_2, \ldots$ is said to be *consistent* if the random graph $\mathbf{F}_{k-1}$ obtained by deleting node $k$ from $\mathbf{G}_k$ has the same distribution as $\mathbf{G}_k$. A random graph model is said to be *local* if whenever $S, T \subset [k]$ are disjoint, the random subgraphs of $\mathbf{G}_k$ induced by $S$ and $T$ are independent random variables. A result of Lovász and Szegedy [16] is that any consistent, local random graph model is equivalent to the distribution on graphs defined by $\mathbb{P}_W$ for some graphon $W$; the converse is true as well. That is, any such random graph model is equivalent to a graphon.

A particular random graph model is not uniquely defined by a graphon – it is clear from Equation 1 that two graphons $W_1$ and $W_2$ which are equal almost everywhere (i.e., differ on a set of measure zero) define the same distribution on graphs. In fact, the distribution defined by $W$ is unchanged by "relabelings" of $W$'s nodes. More formally, if $\Sigma$ is the sigma-algebra of Lebesgue measurable subsets of $[0, 1]$ and $\mu$ is the Lebesgue measure, we say that a relabeling function $\varphi : ([0, 1], \Sigma) \to ([0, 1], \Sigma)$ is *measure preserving* if for any measurable set $A \in \Sigma, \mu(\varphi^{-1}(A)) = \mu(A)$. We define the relabeled graphon $W^\varphi$ by $W^\varphi(x, y) = W(\varphi(x), \varphi(y))$. By analogy with finite graphs, we say that graphons $W_1$ and $W_2$ are *weakly isomorphic* if they are equivalent up to relabeling, i.e., if there exist measure preserving maps $\varphi_1$ and $\varphi_2$ such that $W_1^{\varphi_1} = W_2^{\varphi_2}$ almost everywhere. Weak isomorphism is an equivalence relation, and most of the important properties of a graphon in fact belong to its equivalence class. For instance, a powerful result of [15] is that two graphons define the same random graph model if and only if they are weakly isomorphic.

An example of a graphon $W$ is shown in Figure 1a. It is conventional to plot the graphon as one typically plots an adjacency matrix: with the origin in the upper-left corner. Darker shades correspond to higher values of $W$. Figure 1b depicts a graphon $W^\varphi$ which is weakly isomorphic to $W$. In particular, $W^\varphi$ is the relabeling of $W$ by the measure preserving transformation $\varphi(x) = 2x \mod 1$. As such, the graphons shown in Figures 1a and 1b define the same distribution on graphs. Figure 1c shows the adjacency matrix $A$ of a graph of size $n = 50$

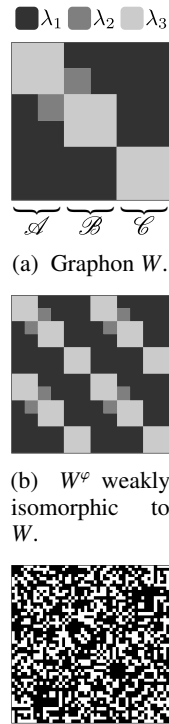

(a) Graphon $W$.

(b) $W^\varphi$ weakly isomorphic to $W$.

(c) An instance of a graph adjacency sampled from $W$.

Figure 1

sampled from the distribution defined by the equivalence class containing $W$ and $W^\varphi$. Note that it is in principle not possible to determine from $A$ alone which graphon $W$ or $W^\varphi$ it was sampled from, or to what node in $W$ a particular column of $A$ corresponds to.

# 3   The graphon cluster tree

We now identify the cluster structure of a graphon. We will define a graphon's clusters such that they are analogous to the maximally-connected components of a finite graph. It turns out that the collection of all clusters has hierarchical structure; we call this object the *graphon cluster tree*. We propose that the goal of clustering in the graphon setting is the recovery of the graphon cluster tree.

**Connectedness and clusters.** Consider a finite weighted graph. It is natural to cluster the graph into connected components. In fact, because of the weighted edges, we can speak of the clusters of the graph at various levels. More precisely, we say that a set of nodes $A$ is *internally connected* – or, from now on, just *connected* – at level $\lambda$ if for every pair of nodes in $A$ there is a path between them such that every node along the path is also in $A$, and the weight of every edge in the path is at least $\lambda$. Equivalently, $A$ is *connected* at level $\lambda$ if and only if for every partitioning of $A$ into disjoint, non-empty sets $A_1$ and $A_2$ there is an edge of weight $\lambda$ or greater between $A_1$ and $A_2$. The clusters at level $\lambda$ are then the largest connected components at level $\lambda$.

A graphon is, in a sense, an infinite weighted graph, and we will define the clusters of a graphon using the example above as motivation. In doing so, we must be careful to make our notion robust to changes of the graphon on a set of zero measure, as such changes do not affect the graph distribution defined by the graphon. We base our definition on that of Janson [13], who defined what it means for a graphon to be connected as a whole. We extend the definition in [13] to speak of the connectivity of subsets of the graphon's nodes at a particular height. Our definition is directly analogous to the notion of internal connectedness in finite graphs.

**Definition 1** (Connectedness)**.** *Let $W$ be a graphon, and let $A \subset [0, 1]$ be a set of positive measure. We say that $A$ is* disconnected at level $\lambda$ *if there exists a measurable $S \subset A$ such that $0 < \mu(S) < \mu(A)$, and $W < \lambda$ almost everywhere on $S \times (A \setminus S)$. Otherwise, we say that $A$ is* connected at level $\lambda$.

We now identify the clusters of a graphon; as in the finite case, we will frame our definition in terms of maximally-connected components. We begin by gathering all subsets of $[0, 1]$ which should belong to some cluster at level $\lambda$. Naturally, if a set is connected at level $\lambda$, it should be in a cluster at level $\lambda$; for technical reasons, we will also say that a set which is connected at all levels $\lambda' < \lambda$ (though perhaps not at $\lambda$) should be contained in a cluster at level $\lambda$, as well. That is, for any $\lambda$, the collection $\mathfrak{A}_\lambda$ of sets which should be contained in some cluster at level $\lambda$ is $\mathfrak{A}_\lambda = \{ A \in \Sigma : \mu(A) > 0$ and $A$ is connected at every level $\lambda' < \lambda \}$. Now suppose $A_1, A_2 \in \mathfrak{A}_\lambda$, and that there is a set $A \in \mathfrak{A}_\lambda$ such that $A \supset A_1 \cup A_2$. Naturally, the cluster to which $A$ belongs should also contain $A_1$ and $A_2$, since both are subsets of $A$. We will therefore consider $A_1$ and $A_2$ to be equivalent, in the sense that they should be contained in the same cluster at level $\lambda$. More formally, we define a relation $\circ\!\!-\!\!\circ_\lambda$ on $\mathfrak{A}_\lambda$ by $A_1 \circ\!\!-\!\!\circ_\lambda A_2 \iff \exists A \in \mathfrak{A}_\lambda$ s.t. $A \supset A_1 \cup A_2$. It can be verified that $\circ\!\!-\!\!\circ_\lambda$ is an equivalence relation on $\mathfrak{A}_\lambda$; see Claim 9 in Appendix B.

Each equivalence class $\mathscr{A}$ in the quotient space $\mathfrak{A}_\lambda/\!\circ\!\!-\!\!\circ_\lambda$. consists of connected sets which should intuitively be clustered together at level $\lambda$. Naturally, we will define the clusters to be the largest elements of each class; in some sense, these are the maximally-connected components at level $\lambda$. More precisely, suppose $\mathscr{A}$ is such an equivalence class. It is clear that in general no single member $A \in \mathscr{A}$ can contain all other members of $\mathscr{A}$, since adding a null set (i.e., a set of measure zero) to $A$ results in a larger set $A'$ which is nevertheless still a member of $\mathscr{A}$. However, we can find a member $A^* \in \mathscr{A}$ which contains all but a null set of every other set in $\mathscr{A}$. More formally, we say that $A^*$ is an *essential maximum* of the class $\mathscr{A}$ if $A^* \in \mathscr{A}$ and for every $A \in \mathscr{A}$, $\mu(A \setminus A^*) = 0$. $A^*$ is of course not unique, but it is unique up to a null set; i.e., for any two essential maxima $A_1, A_2$ of $\mathscr{A}$, we have $\mu(A_1 \triangle A_2) = 0$. We will write the set of essential maxima of $\mathscr{A}$ as $\operatorname{ess\,max} \mathscr{A}$; the fact that the essential maxima are well-defined is proven in Claim 10 in Appendix B. We then define clusters as the maximal members of each equivalence class in $\mathfrak{A}_\lambda/\!\circ\!\!-\!\!\circ_\lambda$:

**Definition 2** (Clusters)**.** *The set of clusters at level $\lambda$ in $W$, written $\mathbb{C}_W(\lambda)$, is defined to be the countable collection $\mathbb{C}_W(\lambda) = \{ \operatorname{ess\,max} \mathscr{A} : \mathscr{A} \in \mathfrak{A}_\lambda/\!\circ\!\!-\!\!\circ_\lambda \}$.*

Note that a cluster $\mathscr{C}$ of a graphon is not a subset of the unit interval per se, but rather an *equivalence class* of subsets which differ only by null sets. It is often possible to treat clusters as sets rather than equivalence classes, and we may write $\mu(\mathscr{C})$, $\mathscr{C} \cup \mathscr{C}'$, etc., without ambiguity. In addition, if $\varphi : [0, 1] \to [0, 1]$ is a measure preserving transformation, then $\varphi^{-1}(\mathscr{C})$ is well-defined.

For a concrete example of our notion of a cluster, consider the graphon $W$ depicted in Figure 1a. $A$, $B$, and $C$ represent sets of the graphon's nodes. By our definitions there are three clusters at level $\lambda_3$: $\mathscr{A}$, $\mathscr{B}$, and $\mathscr{C}$. Clusters $\mathscr{A}$ and $\mathscr{B}$ merge into a cluster $\mathscr{A} \cup \mathscr{B}$ at level $\lambda_2$, while $\mathscr{C}$ remains a separate cluster. Everything is joined into a cluster $\mathscr{A} \cup \mathscr{B} \cup \mathscr{C}$ at level $\lambda_1$.

We have taken care to define the clusters of a graphon in such a way as to be robust to changes of measure zero to the graphon itself. In fact, clusters are also robust to measure preserving transformations. The proof of this result is non-trivial, and comprises Appendix C.

**Claim 1.** *Let $W$ be a graphon and $\varphi$ a measure preserving transformation. Then $\mathscr{C}$ is a cluster of $W^\varphi$ at level $\lambda$ if and only if there exists a cluster $\mathscr{C}'$ of $W$ at level $\lambda$ such that $\mathscr{C} = \varphi^{-1}(\mathscr{C}')$.*

**Cluster trees and mergeons.** The set of all clusters of a graphon at any level has hierarchical structure in the sense that, given any pair of distinct clusters $\mathscr{C}_1$ and $\mathscr{C}_2$, either one is "essentially" contained within the other, i.e., $\mathscr{C}_1 \subset \mathscr{C}_2$, or $\mathscr{C}_2 \subset \mathscr{C}_1$, or they are "essentially" disjoint, i.e., $\mu(\mathscr{C}_1 \cap \mathscr{C}_2) = 0$, as is proven by Claim 8 in Appendix B. Because of this hierarchical structure, we call the set $\mathbb{C}_W$ of all clusters from any level of the graphon $W$ the *graphon cluster tree* of $W$. It is this tree that we hope to recover by applying a graph clustering algorithm to a graph sampled from $W$.

We may naturally speak of the height at which pairs of distinct clusters merge in the cluster tree. For instance, let $\mathscr{C}_1$ and $\mathscr{C}_2$ be distinct clusters of $\mathbb{C}$. We say that the *merge height* of $\mathscr{C}_1$ and $\mathscr{C}_2$ is the level $\lambda$ at which they are joined into a single cluster, i.e., $\max\{\lambda : \mathscr{C}_1 \cup \mathscr{C}_2 \in \mathbb{C}(\lambda)\}$. However, while the merge height of clusters is well-defined, the merge height of individual points is not. This is because the cluster tree is not a collection of sets, but rather a collection of equivalence classes of sets, and so a point does not belong to any one cluster more than any other. Note that this is distinct from the classical density case considered in [12], [9], and [1], where the merge height of any pair of points is well-defined.

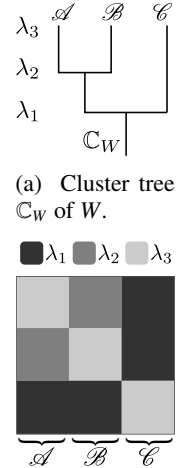

(a) Cluster tree $\mathbb{C}_W$ of $W$.

(b) Mergeon $M$ of $\mathbb{C}_W$.

Figure 2

Nevertheless, consider a measurable function $M : [0, 1]^2 \to [0, 1]$ which assigns a merge height to every pair of points. While the value of $M$ on any given pair is arbitrary, the value of $M$ on sets of positive measure is constrained. Intuitively, if $\mathscr{C}$ is a cluster at level $\lambda$, then we must have $M \geq \lambda$ almost everywhere on $\mathscr{C} \times \mathscr{C}$. If $M$ satisfies this constraint for every cluster $\mathscr{C}$ we call $M$ a *mergeon* for $\mathbb{C}$, as it is a graphon which determines a particular choice for the merge heights of every pair of points in $[0, 1]$. More formally:

**Definition 3** (Mergeon). *Let $\mathbb{C}$ be a cluster tree. A* mergeon[1] *of $\mathbb{C}$ is a graphon $M$ such that for all $\lambda \in [0, 1]$, $M^{-1}[\lambda, 1] = \bigcup_{\mathscr{C} \in \mathbb{C}_W(\lambda)} \mathscr{C} \times \mathscr{C}$, where $M^{-1}[\lambda, 1] = \{(x, y) \in [0, 1]^2 : M(x, y) \geq \lambda\}$.*

An example of a mergeon and the cluster tree it represents is shown in Figure 2. In fact, the cluster tree depicted is that of the graphon $W$ from Figure 1a. The mergeon encodes the height at which clusters $\mathscr{A}$, $\mathscr{B}$, and $\mathscr{C}$ merge. In particular, the fact that $M = \lambda_2$ everywhere on $\mathscr{A} \times \mathscr{B}$ represents the merging of $\mathscr{A}$ and $\mathscr{B}$ at level $\lambda_2$ in $W$.

It is clear that in general there is no unique mergeon representing a graphon cluster tree, however, the above definition implies that two mergeons representing the same cluster tree are equal almost everywhere. Additionally, we have the following two claims, whose proofs are in Appendix B.

**Claim 2.** *Let $\mathbb{C}$ be a cluster tree, and suppose $M$ is a mergeon representing $\mathbb{C}$. Then $\mathscr{C} \in \mathbb{C}(\lambda)$ if and only if $\mathscr{C}$ is a cluster in $M$ at level $\lambda$. In other words, the cluster tree of $M$ is also $\mathbb{C}$.*

**Claim 3.** *Let $W$ be a graphon and $M$ a mergeon of the cluster tree of $W$. If $\varphi$ is a measure preserving transformation, then $M^\varphi$ is a mergeon of the cluster tree of $W^\varphi$.*

# 4 Notions of consistency

We have so far defined the sense in which a graphon has hierarchical cluster structure. We now turn to the problem of determining whether a clustering algorithm is able to recover this structure when applied to a graph sampled from a graphon. Our approach is to define a distance between the infinite graphon cluster tree and a finite clustering. We will then define consistency by requiring that a consistent method converge to the graphon cluster tree in this distance for all inputs minus a set of vanishing probability.

**Merge distortion.** A *hierarchical clustering* $\mathsf{C}$ of a set $S$ – or, from now on, just a *clustering* of $S$ – is hierarchical collection of subsets of $S$ such that $S \in \mathsf{C}$ and for all $C, C' \in \mathsf{C}$, either $C \subset C'$, $C' \subset C$, or $C \cap C' = \emptyset$. Suppose $\mathsf{C}$ is a clustering of a finite set $S$ consisting of graphon nodes; i.e, $S \subset [0, 1]$. How might we measure the distance between this clustering and a graphon cluster tree $\mathbb{C}$? Intuitively, the two trees are close if every pair of points in $S$ merges in $\mathsf{C}$ at about the same level as they merge in $\mathbb{C}$. But this informal description faces two problems: First, $\mathbb{C}$ is a collection of equivalence classes of sets, and so the height at which any pair of points merges in $\mathbb{C}$ is not defined. Recall, however, that the cluster tree has an alternative representation as a *mergeon*. A mergeon *does* define a merge height for every pair of nodes in a graphon, and thus provides a solution to this first issue. Second, the clustering $\mathsf{C}$ is not equipped with a height function, and so the height at which any pair of points merges in $\mathsf{C}$ is also undefined. Following [10], our approach is to *induce* a merge height function on the clustering using the mergeon in the following way:

**Definition 4** (Induced merge height)**.** *Let $M$ be a mergeon, and suppose $S$ is a finite subset of* $[0, 1]$*. Let $\mathsf{C}$ be a clustering of $S$. The* merge height function on $\mathsf{C}$ induced by $M$ *is defined by* $\hat{M}_{\mathsf{C}}(s, s') = \min_{u,v \in \mathsf{C}(s,s')} M(u, v)$*, for every $s, s' \in S \times S$, where $\mathsf{C}(s, s')$ denotes the smallest cluster $C \in \mathsf{C}$ which contains both $s$ and $s'$.*

We measure the distance between a clustering $\mathsf{C}$ and the cluster tree $\mathbb{C}$ using the *merge distortion*:

**Definition 5.** *Let $M$ be a mergeon, $S$ a finite subset of* $[0, 1]$*, and $\mathsf{C}$ a clustering of $S$. The* merge distortion *is defined by* $d_S(M, \hat{M}_{\mathsf{C}}) = \max_{s,s' \in S, \, s \neq s'} |M(s, s') - \hat{M}_{\mathsf{C}}(s, s')|$.

Defining the induced merge height and merge distortion in this way leads to an especially meaningful interpretation of the merge distortion. In particular, if the merge distortion between $\mathsf{C}$ and $\mathbb{C}$ is $\epsilon$, then any two clusters of $\mathbb{C}$ which are separated at level $\lambda$ but merge below level $\lambda - \epsilon$ are correctly separated in the clustering $\mathsf{C}$. A similar result guarantees that a cluster in $\mathbb{C}$ is connected in $\mathsf{C}$ at within $\epsilon$ of the correct level. For a precise statement of these results, see Claim 5 in Appendix A.4.

**The label measure.** We will use the merge distortion to measure the distance between $\mathsf{C}$, a hierarchical clustering of a graph, and $\mathbb{C}$, the graphon cluster tree. Recall, however, that the nodes of a graph sampled from a graphon have integer labels. That is, $\mathsf{C}$ is a clustering of $[n]$, and not of a subset of $[0, 1]$. Hence, in order to apply the merge distortion, we must first relabel the nodes of the graph, placing them in direct correspondence to nodes of the graphon, i.e., points in $[0, 1]$.

Recall that we sample a graph of size $n$ from a graphon $W$ by first drawing $n$ points $\mathbf{x_1}, \ldots, \mathbf{x_n}$ uniformly at random from the unit interval. We then generate a graph on node set $[n]$ by connecting nodes $i$ and $j$ with probability $W(\mathbf{x_i}, \mathbf{x_j})$. However, the nodes of the sampled graph are not labeled by $\mathbf{x_1}, \ldots, \mathbf{x_n}$, but rather by the integers $1, \ldots, n$. Thus we may think of $\mathbf{x_i}$ as being the "true" latent label of node $i$. In general the latent node labeling is not recoverable from data, as is demonstrated by the figure to the right. We might suppose that the graph shown is sampled from the graphon above it, and that node 1 corresponds to $a$, node 2 to $b$, node 3 to $c$, and node 4 to $d$. However, it is just as likely that node 4 corresponds to $d'$, and so neither labeling is more "correct". It is clear, though, that some labelings are less likely than others. For instance, the existence of the edge $(1, 2)$ makes it impossible that 1 corresponds to $a$ and 2 to $c$, since $W(a, c)$ is zero.

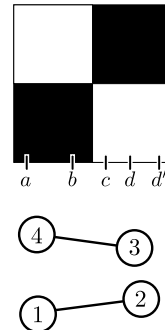

Therefore, given a graph $G = ([n], E)$ sampled from a graphon, there are many possible relabelings of $G$ which place its nodes in correspondence with nodes of the graphon, but some are more likely than others. The merge distortion depends which labeling of $G$ we assume, but, intuitively, a good clustering of $G$ will have small distortion with respect to highly probable labelings, and only have large distortion on improbable labelings. Our approach is to assign a probability to every pair $(G, S)$ of a graph and possible labeling. We will thus be able to measure the probability mass of the set of

pairs for which a method performs poorly, i.e., results in a large merge distortion.

More formally, let $\mathfrak{G}_n$ denote the set of all undirected, unweighted graphs on node set $[n]$, and let $\Sigma^n$ be the sigma-algebra of Lebesgue-measurable subsets of $[0, 1]^n$. A graphon $W$ induces a unique product measure $\Lambda_{W,n}$ defined on the product sigma-algebra $2^{\mathfrak{G}_n} \times \Sigma^n$ such that for all $\mathcal{G} \in 2^{\mathfrak{G}_n}$ and $S \in \Sigma^n$:

$$\Lambda_{W,n}(\mathcal{G} \times S) = \sum_{G \in \mathcal{G}} \left( \int_S \mathcal{L}_W(S|G)\, dS \right), \text{ where } \mathcal{L}_W(S \mid G) = \prod_{(i,j) \in E(G)} W(x_i, x_j) \prod_{(i,j) \notin E(G)} \left[ 1 - W(x_i, x_j) \right],$$

where $E(G)$ represents the edge set of the graph $G$. We recognize $\mathcal{L}_W(S \mid G)$ as the integrand in Equation 1 for the probability of a graph as determined by a graphon. If $G$ is fixed, integrating $\mathcal{L}_W(S \mid G)$ over all $S \in [0, 1]^n$ gives the probability of $G$ under the model defined by $W$.

We may now formally define our notion of consistency. First, some notation: If $\mathsf{C}$ is a clustering of $[n]$ and $S = (x_1, \ldots, x_n)$, write $\mathsf{C} \circ S$ to denote the relabeling of $\mathsf{C}$ by $S$, in which $i$ is replaced by $x_i$ in every cluster. Then if $f$ is a hierarchical graph clustering method, $f(G) \circ S$ is a clustering of $S$, and $\hat{M}_{f(G) \circ S}$ denotes the merge function induced on $f(G) \circ S$ by $M$.

**Definition 6** (Consistency). *Let $W$ be a graphon and $M$ be a mergeon of $W$. A hierarchical graph clustering method $f$ is said to be a* consistent *estimator of the graphon cluster tree of $W$ if for any fixed $\epsilon > 0$, as $n \to \infty$, $\Lambda_{W,n}\left( \{(G, S) : d_S(M, \hat{M}_{f(G) \circ S}) > \epsilon\} \right) \to 0$.*

The choice of mergeon for the graphon $W$ does not affect consistency, as any two mergeons of the same graphon differ on a set of measure zero. Furthermore, consistency is with respect to the random graph model, and not to any particular graphon representing the model. The following claim, the proof of which is in Appendix B, makes this precise.

**Claim 4.** *Let $W$ be a graphon and $\varphi$ a measure preserving transformation. A clustering method $f$ is a consistent estimator of the graphon cluster tree of $W$ if and only if it is a consistent estimator of the graphon cluster tree of $W^\varphi$.*

**Consistency and the blockmodel.** If a graph clustering method is consistent in the sense defined above, it is also consistent in the stochastic blockmodel; i.e., it ensures strict recovery of the communities with high probability as the size of the graphs grow large. For instance, suppose $W$ is a stochastic blockmodel graphon with $\alpha$ along the block-diagonal and $\beta$ everywhere else. $W$ has two clusters at level $\alpha$, merging into one cluster at level $\beta$. When the merge distortion between the graphon cluster tree and a clustering is less than $\alpha - \beta$, which will eventually be the case with high probability if the method is consistent, the two clusters are totally disjoint in $\mathsf{C}$; this implication is made precise by Claim 5 in Appendix A.4.

# 5 Consistent algorithms

We now demonstrate that consistent clustering methods exist. We present two results: First, we show that any method which is capable of consistently estimating the probability of each edge in a random graph leads to a consistent clustering method. We then analyze a modification of an existing algorithm to show that it consistently estimates edge probabilities. As a corollary, we identify a graph clustering method which satisfies our notion of consistency. Our results will be for graphons which are piecewise Lipschitz (or weakly isomorphic to a piecewise Lipschitz graphon):

**Definition 7** (Piecewise Lipschitz). *We say that $\mathcal{B} = \{B_1, \ldots, B_k\}$ is a* block partition *if each $B_i$ is an open, half-open, or closed interval in $[0, 1]$ with positive measure, $B_i \cap B_j$ is empty whenever $i \neq j$, and $\bigcup \mathcal{B} = [0, 1]$. We say that a graphon $W$ is* piecewise $\mathsf{c}$-Lipschitz *if there exists a set of blocks $\mathcal{B}$ such that for any $(x, y)$ and $(x', y')$ in $B_i \times B_j$, $|W(x, y) - W(x', y')| \leq \mathsf{c}(|x - x'| + |y - y'|)$.*

Our first result concerns methods which are able to consistently estimate edge probabilities in the following sense. Let $\mathbf{S} = (\mathbf{x_1}, \ldots, \mathbf{x_n})$ be an ordered set of $n$ uniform random variables drawn from the unit interval. Fix a graphon $W$, and let $\mathbf{P}$ be the random matrix whose $ij$ entry is given by $W(\mathbf{x_i}, \mathbf{x_j})$. We say that $\mathbf{P}$ is the random *edge probability matrix*. Assuming that $W$ has structure, it is possible to estimate $\mathbf{P}$ from a single graph sampled from $W$. We say that an estimator $\hat{\mathbf{P}}$ of $\mathbf{P}$ is *consistent* in max-norm if, for any $\epsilon > 0$, $\lim_{n \to \infty} \mathbb{P}(\max_{i \neq j} |\mathbf{P}_{ij} - \hat{\mathbf{P}}_{ij}| > \epsilon) = 0$. The following non-trivial theorem, whose proof comprises Appendix D, states that any estimator which is consistent in this sense leads to a consistent clustering algorithm:

**Theorem 1.** *Let $W$ be a piecewise c-Lipschitz graphon. Let $\hat{\mathbf{P}}$ be a consistent estimator of $\mathbf{P}$ in max-norm. Let $f$ be the clustering method which performs single-linkage clustering using $\hat{\mathbf{P}}$ as a similarity matrix. Then $f$ is a consistent estimator of the graphon cluster tree of $W$.*

Estimating the matrix of edge probabilities has been a direction of recent research, however we are only aware of results which show consistency in mean squared error; That is, the literature contains estimators for which $1/n^2 \|\mathbf{P} - \hat{\mathbf{P}}\|_F^2$ tends to zero in probability. One practical method is the neighborhood smoothing algorithm of [21]. The method constructs for each node $i$ in the graph $\mathbf{G}$ a neighborhood of nodes $\mathbf{N_i}$ which are similar to $i$ in the sense that for every $i' \in \mathbf{N_i}$, the corresponding column $\mathbf{A_{i'}}$ of the adjacency matrix is close to $\mathbf{A_i}$ in a particular distance. $\mathbf{A_{ij}}$ is clearly not a good estimate for the probability of the edge $(i, j)$, as it is either zero or one, however, if the graphon is piecewise Lipschitz, the average $\mathbf{A_{i'j}}$ over $i' \in \mathbf{N_{ij}}$ will intuitively tend to the true probability. Like others, the method of [21] is proven to be consistent in mean squared error. Since Theorem 1 requires consistency in max-norm, we analyze a slight modification of this algorithm and show that it consistently estimates $\mathbf{P}$ in this stronger sense. The technical details are in Appendix E.

---

**Algorithm 1** Clustering by nbhd. smoothing

---

**Require:** Adjacency matrix $A$, $C \in (0, 1)$
  *% Step 1: Compute the estimated edge*
  *% probability matrix $\hat{P}$ using neighborhood*
  *% smoothing algorithm based on [21]*
  $n \leftarrow \text{Size}(A)$
  $h \leftarrow C \sqrt{(\log n)/n}$
  **for** $i \neq j \in [n] \times [n]$ **do**
    $\hat{A} \leftarrow A$ after setting row/column $j$ to zero
    **for** $i' \in [n] \setminus \{i, j\}$ **do**
      $d_j(i, i') \leftarrow \max_{k \neq i, i', j} |(\hat{A}^2/n)_{ik} - (\hat{A}^2/n)_{i'k}|$
    **end for**
    $q_{ij} \leftarrow h$th quantile of $\{d_j(i, i') : i' \neq i, j\}$
    $N_{ij} \leftarrow \{i' \neq i, j : d_j(i, i') \leq q_{ij}(h)\}$
  **end for**
  **for** $(i, j) \in [n] \times [n]$ **do**
    $\hat{P}_{ij} \leftarrow \frac{1}{2} \left( \frac{1}{N_{ij}} \sum_{i' \in N_{ij}} A_{i'j} + \frac{1}{N_{ji}} \sum_{j' \in N_{ji}} A_{ij'} \right)$
  **end for**
  *% Step 2: Cluster $\hat{P}$ with single linkage*
  $\mathsf{C} \leftarrow$ the single linkage clusters of $\hat{P}$
  **return** $\mathsf{C}$

---

**Theorem 2.** *If the graphon $W$ is piecewise Lipschitz, the modified neighborhood smoothing algorithm in Appendix E is a consistent estimator of $\mathbf{P}$ in max-norm.*

As a corollary, we identify a practical graph clustering algorithm which is a consistent estimator of the graphon cluster tree. The algorithm is shown in Algorithm 1, and details are in Appendix E.2. Appendix F contains experiments in which the algorithm is applied to real and synthetic data.

**Corollary 1.** *If the graphon $W$ is piecewise Lipschitz, Algorithm 1 is a consistent estimator of the graphon cluster tree of $W$.*

# 6 Discussion

We have presented a consistency framework for clustering in the graphon model and demonstrated that a practical clustering algorithm is consistent. We now identify two interesting directions of future research. First, it would be interesting to consider the extension of our framework to *sparse* random graphs; many real-world networks are sparse, and the graphon generates dense graphs. Recently, however, sparse models which extend the graphon have been proposed; see [7, 6]. It would be interesting to see what modifications are necessary to apply our framework in these models.

Second, it would be interesting to consider alternative ways of defining the ground truth clustering of a graphon. Our construction is motivated by interpreting the graphon $W$ not only as a random graph model, but also as a similarity function, which may not be desirable in certain settings. For example, consider a "bipartite" graphon $W$, which is one along the block-diagonal and zero elsewhere. The cluster tree of $W$ consists of a single cluster at all levels, whereas the ideal bipartite clustering has two clusters. Therefore, consider applying a transformation $S$ to $W$ which maps it to a "similarity" graphon. The goal of clustering then becomes the recovery of the cluster tree of $S(W)$ given a random graph sampled from $W$. For instance, let $S : W \mapsto W^2$, where $W^2$ is the operator square of the bipartite graphon $W$. The cluster tree of $S(W)$ has two clusters at all positive levels, and so represents the desired ground truth. In general, any such transformation $S$ leads to a different clustering goal. We speculate that, with minor modification, the framework herein can be used to prove consistency results in a wide range of graph clustering settings.

**Acknowledgements.** This work was supported by NSF grant IIS-1550757.

## Footnotes

[1]The definition given here involves a slight abuse of notation. For a precise – but more technical – version, see Appendix A.2.

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
