[Supplementary Material]

# A Technical details

## A.1 Measurable sets modulo null sets

Let $(\Omega, \Sigma, \mu)$ be a measure space. Let $A, A'$ be any measurable sets and define $\sim_\emptyset$ to be the relation $A \sim_\emptyset A' \Leftrightarrow \mu(A \triangle A') = 0$; that is, two measurable sets are equivalent under $\sim_\emptyset$ if they differ by a null set. Write $\Sigma/\sim_\emptyset$ for the quotient space of measurable sets by $\sim_\emptyset$, and denote by $[A]_\emptyset$ the equivalence class containing the set $A$. Throughout, we use script letters such as $\mathscr{A}$ to denote these equivalence classes of measurable sets modulo null sets.

We can often use the normal set notation to manipulate such classes without ambiguity. For instance, if $\mathscr{A}$ and $\mathscr{A}'$ are two classes in $\Sigma/\sim_\emptyset$, we define $\mathscr{A} \cup \mathscr{A}'$ to be $[A \cup A']_\emptyset$, where $A$ and $A'$ are arbitrary members of $\mathscr{A}$ and $\mathscr{A}'$. $\mathscr{A} \cap \mathscr{A}'$ and $\mathscr{A} \setminus \mathscr{A}'$ are defined similarly. We can define $\mathscr{A} \times \mathscr{A}'$ in this manner too; note that the result is an equivalence class in $\Sigma \times \Sigma/\sim_\emptyset$, where the relation $\sim_\emptyset$ is implicitly assumed to be with respect to the product measure $\mu \times \mu$. Similarly, we can unambiguously order such equivalence classes. For example, we write $\mathscr{A} \subset \mathscr{A}'$ to denote $\mu(\mathscr{A} \setminus \mathscr{A}') = 0$.

In some instances it will be more convenient to work with sets as opposed to equivalence classes of sets. In such cases we will use a *section* map $\rho$ which returns an (often arbitrary) member of the class, $\rho(\mathscr{A})$.

## A.2 A precise definition of a mergeon

In Definition 3, we introduce the *mergeon* of a cluster tree $\mathbb{C}$ as a graphon $M$ satisfying

$$M^{-1}[\lambda, 1] = \bigcup_{\mathscr{C} \in \mathbb{C}_W(\lambda)} \mathscr{C} \times \mathscr{C}$$

for all $\lambda \in [0, 1]$. This definition involves a slight abuse of notation. In particular, $\mathscr{C}$ is an equivalence class of sets modulo null sets. Therefore, as descibed in the previous subsection, $\mathscr{C} \times \mathscr{C}$ is defined to be an equivalence class of measurable subsets of $[0, 1] \times [0, 1]$ modulo null sets. On the other hand, $M^{-1}[\lambda, 1]$ is simply a measurable subset of $[0, 1] \times [0, 1]$. Therefore, the left and the right of the above equation are two different types of objects, and we are being imprecise in equating them.

A precise statement of this definition is as follows:

**Definition 8** (Mergeon, rigorous). *A mergeon of $\mathbb{C}$ is a graphon $M$ such that for all $\lambda \in [0, 1]$,*

$$M^{-1}[\lambda, 1] \triangle \bigcup_{\mathscr{C} \in \mathbb{C}_W(\lambda)} \rho(\mathscr{C}) \times \rho(\mathscr{C})$$

*is a null set, where $M^{-1}[\lambda, 1] = \{(x, y) : M(x, y) \geq \lambda\}$, $\triangle$ is the symmetric difference operator, and $\rho$ is an arbitrary section map. Equivalently, a mergeon of $\mathbb{C}$ is a graphon $M$ such that for all $\lambda \in [0, 1]$,*

$$[M^{-1}[\lambda, 1]]_\emptyset = \bigcup_{\mathscr{C} \in \mathbb{C}_W(\lambda)} \mathscr{C} \times \mathscr{C},$$

*where $[M^{-1}[\lambda, 1]]_\emptyset$ is the equivalence class of measurable subsets of $[0, 1] \times [0, 1]$ modulo null sets which contains $M^{-1}[\lambda, 1]$.*

In these more rigorous definitions we compare sets to sets, or equivalence classes to equivalence classes, and thus precisely define the mergeon.

## A.3 Strict cluster trees and their mergeons

A graphon cluster tree is a hierarchical collection of equivalence classes of sets. It is sometimes useful to instead to work with a hierarchical collection of subsets of $[0, 1]$. We may always do so by choosing a section map $\rho$ and applying it to every cluster in the cluster tree. Though the choice of representative of a given cluster is often arbitrary, it will sometimes be useful to choose it in such a way that the cluster tree has strictly nested structure, as made precise in the following definition.

**Definition 9** (Strict section). *Let $\mathbb{C}$ be a cluster tree. A strict section $\tilde{\rho} : \mathbb{C} \to \Sigma$ is a function which selects a unique representative from each cluster $\mathscr{C}$ such that if:*

1. $\mu(\mathscr{C} \cap \mathscr{C}') = 0 \Rightarrow \tilde{\rho}(\mathscr{C}) \cap \tilde{\rho}(\mathscr{C}) = \emptyset$,

2. $\mathscr{C} \subset \mathscr{C}' \Rightarrow \tilde{\rho}(\mathscr{C}) \subset \tilde{\rho}(\mathscr{C}')$, and

3. *(Technical condition)* $\tilde{\rho}(\mathscr{C}) = \bigcap \{\tilde{\rho}(\mathscr{C}') : \mathscr{C}' \supset \mathscr{C}\}$.

*The* strict cluster tree $\tilde{\mathbb{C}}$ induced by applying $\tilde{\rho}$ to $\mathbb{C}$ is defined by $\tilde{\mathbb{C}}(\lambda) = \{\tilde{\rho}(\mathscr{C}) : \mathscr{C} \in \mathbb{C}(\lambda)\}$.

Claim 14 in Appendix B proves that it is always possible to construct a strict section. Furthermore, given a cluster tree and a strict section, there is a *unique* mergeon representing the strict cluster tree, defined as follows:

**Definition 10** (Strict mergeon)**.** *Let $\mathbb{C}$ be a cluster tree, and suppose $\tilde{\rho}$ is a strict section for the clusters of $\mathbb{C}$. Then M is a* strict mergeon *of the strict cluster tree induced by $\tilde{\rho}$ if, for every $\lambda \in [0, 1]$,*

$$M^{-1}[\lambda, 1] = \bigcup_{\mathscr{C} \in \mathbb{C}_W(\lambda)} \tilde{\rho}(\mathscr{C}) \times \tilde{\rho}(\mathscr{C}).$$

Because any two mergeons of the same cluster tree differ only on a null set, we are typically free to assume that a mergeon is strict without much loss. Making this assumption will simplify some statements and proofs.

### A.4   Merge distortion and cluster structure

Definition 4 introduces the merge height induced on a clustering by a mergeon. There are, of course, other approachs to inducing a merge height on a clustering, but our definition allows for a particular interpretation of the merge distortion in terms of the cluster structure that is recovered by the finite clustering, as the following claim makes precise. It is convenient to state the claim using the notion of a *strict mergeon* defined in Appendix A.3; analogous (but equally as strong) claims can be made for general mergeons and cluster trees.

**Claim 5.** *Let $\mathbb{C}$ be cluster tree, and let $\tilde{\mathbb{C}}$ be a strict cluster tree obtained by applying a strict section $\tilde{\rho}$ to each cluster of $\mathbb{C}$. Let M be the strict mergeon representing $\tilde{\mathbb{C}}$. Let $S = (x_1, \ldots, x_n)$ with each $x_i \in [0, 1]$ and suppose $\mathsf{C}$ is a clustering of S. Let $\hat{M}$ be the induced merge height on $\mathsf{C}$. If $d_S(M, \hat{M}) < \epsilon$, we then have:*

1. Connectedness*: If C is a cluster of $\tilde{\mathbb{C}}$ at level $\lambda$ and $|S \cap C| \geq 2$, then the smallest cluster in $\mathsf{C}$ which contains all of $S \cap C$ is contained within $C' \cap S$, where $C'$ is the cluster of $\tilde{\mathbb{C}}$ at level $\lambda' = \lambda - \epsilon$ which contains C.*

2. Separation*: If $C_1$ and $C_2$ are two clusters of $\tilde{\mathbb{C}}$ at level $\lambda$ such that $C_1$ and $C_2$ merge at level $\lambda' < \lambda - \epsilon$, then if $|C_1 \cap S|, |C_2 \cap S| \geq 2$, the smallest cluster in $\mathsf{C}$ containing $C_1 \cap S$ and the smallest cluster containing $C_2 \cap S$ are disjoint.*

The proof of this claim is found in Appendix B.

## B   Proofs

In the following proofs we will often work with the clusters of a graphon. Recall that, formally, a cluster is not a subset of $[0, 1]$, but rather an equivalence class of measurable subsets of $[0, 1]$ which differ by null sets. Nevertheless, we will often speak as though the clusters are in fact subsets of $[0, 1]$; we can typically do so without issue. For instance, we might say "$C \subset [0, 1]$ is a cluster of a graphon $W$ at level $\lambda$." Technically-speaking, this is incorrect. However, we might interpret the above statement as saying that there exists a cluster $\mathscr{C}$ at level $\lambda$ such that $C$ is a representative of $\mathscr{C}$.

**Claim 2.** *Let $\mathbb{C}$ be a cluster tree, and suppose M is a mergeon representing $\mathbb{C}$. Then $\mathscr{C} \in \mathbb{C}(\lambda)$ if and only if $\mathscr{C}$ is a cluster in M at level $\lambda$. In other words, the cluster tree of M is also $\mathbb{C}$.*

*Proof.* Let $C$ be an arbitrary representative of the cluster $\mathscr{C}$. By definition of the mergeon, all but a null set of $C \times C$ is contained within $M^{-1}[\lambda, 1]$, and therefore $M \geq \lambda$ almost everywhere on $C \times C$. This implies that $C$ is connected at level $\lambda$ in $M$, which in turn implies that $C$ is contained in some cluster $C'$ of $M$ at level $\lambda$. By definition, $C'$ is connected in $M$ at every level $\lambda' < \lambda$, and so Claim 12

in Appendix B implies that $C'$ is contained in some cluster of $W$ at every level $\lambda' < \lambda$. Claim 13 in Appendix B then implies that $C'$ is contained in some cluster of $W$ at level $\lambda$. In other words, $C$ is a cluster of $W$ at level $\lambda$, and $C \subset C'$, so the fact that $C'$ is contained in a cluster of $W$ at level $\lambda$ implies that $C'$ differs from $C$ by at most a null set. Hence $C$ is a cluster of $M$.

Now suppose $C$ is a cluster of $M$ at level $\lambda$. Then $C$ is connected in $M$ at every level $\lambda' < \lambda$, and so Claim 12 implies that $C$ is contained in some cluster of $W$ at every level $\lambda' < \lambda$. Claim 13 then implies that $C$ is contained in some cluster of $W$ at level $\lambda$. Let $C'$ be this cluster. Then the above implies that $C'$ is a cluster at level $\lambda$ in $M$. But $C \subset C'$, and $C$ is a cluster of $M$, so it must be that $C$ and $C'$ differ by a null set, and hence $C$ is a cluster of $W$.

$\square$

**Claim 3.** *Let $W$ be a graphon and $M$ a mergeon of the cluster tree of $W$. If $\varphi$ is a measure preserving transformation, then $M^\varphi$ is a mergeon of the cluster tree of $W^\varphi$.*

*Proof.* One one hand, the function defined by

$$M_0^{-1}[\lambda, 1] = \bigcup_{C \in \mathbb{C}_W(\lambda)} \varphi^{-1}(C) \times \varphi^{-1}(C)$$

is a mergeon of $W^\varphi$, since $C$ is a cluster of $W$ if and only if $\varphi^{-1}(C)$ is a cluster of $W^\varphi$ by Claim 1 in the body of the paper. Now consider the pullback $M^\varphi$ and its upper level set

$$(M^\varphi)^{-1}[\lambda, 1] = \{(x, y) : M^\varphi(x, y) \geq \lambda\}$$
$$= \{(x, y) : M(\varphi(x), \varphi(y)) \geq \lambda\},$$

which, by definition of the mergeon, is

$$= \left\{ (x, y) : (\varphi(x), \varphi(y)) \in \bigcup_{C \in \mathbb{C}_W(\lambda)} C \times C \right\}.$$

It is well-known that if $\varphi$ is a measure preserving map, then the transformation defined by $\Phi : (x, y) \mapsto (\varphi(x), \varphi(y))$ is also measure preserving and measurable. Therefore we have

$$= \Phi^{-1} \left( \bigcup_{C \in \mathbb{C}_W(\lambda)} C \times C \right).$$

Since preimages commute with arbitrary unions:

$$= \bigcup_{C \in \mathbb{C}_W(\lambda)} \Phi^{-1}(C \times C)$$

Some thought will show that $\Psi^{-1}(C \times C) = \varphi^{-1}(C) \times \varphi^{-1}(C)$, such that:

$$= \bigcup_{C \in \mathbb{C}_W(\lambda)} \varphi^{-1}(C) \times \varphi^{-1}(C)$$

Comparing this to the definition of $M_0$ above, which was a mergeon of $W^\varphi$, we see that $M^\varphi$ is a mergeon of $W^\varphi$.

$\square$

**Claim 4.** *Let $W$ be a graphon and $\varphi$ a measure preserving transformation. A clustering method $f$ is a consistent estimator of the graphon cluster tree of $W$ if and only if it is a consistent estimator of the graphon cluster tree of $W^\varphi$.*

*Proof.* Let $M$ be a mergeon of the cluster tree of $W$ and fix any $\epsilon > 0$. Consider the set

$$F = \left\{ (G, S) : d_S\left(M, \hat{M}_{f(G) \circ S}\right) > \epsilon \right\},$$

which is the set of graph/sample pairs for which the merge distortion between the clustering and the mergeon $M$ is greater than $\epsilon$. Consistency with respect to the cluster tree of $W$ requires that $\Lambda_{W,n}(F) \to 0$ as $n \to \infty$. Now recall that $M^\varphi$ is a mergeon of the cluster tree of $W^\varphi$, and consider

$$F_\varphi = \left\{ (G, S) : d_S \left( M^\varphi, \hat{M}^\varphi_{f(G) \circ S} \right) > \epsilon \right\}$$

where $\hat{M}^\varphi_{f(G) \circ S}$ is the merge height induced on the clustering $f(G) \circ S$ by the mergeon $M^\varphi$. $F_\varphi$ is the set of graph/sample pairs for which the merge distortion between the clustering and the mergeon $M^\varphi$ is greater than $\epsilon$. Consistency with respect to the cluster tree of $W^\varphi$ requires that $\Lambda_{W^\varphi,n}(F_\varphi) \to 0$ as $n \to \infty$. It will therefore be sufficient to show that $\Lambda_{W,n}(F) = \Lambda_{W^\varphi,n}(F_\varphi)$ to prove the claim.

Now we compute the measure under $\Lambda_{W^\varphi,n}$ of $F_\varphi$:

$$\Lambda_{W^\varphi,n}(F_\varphi) = \sum_{G \in \mathbb{G}_n} \int_{F_\varphi(G)} \mathcal{L}_{W^\varphi}(S \mid G) \, dS,$$

where $F_\varphi(G)$ denotes the *section* of $F_\varphi$ by graph $G$, that is, the set $F_\varphi(G) = \{ S : (G, S) \in F_\varphi \}$. It is easy to see that $\mathcal{L}_{W^\varphi}(S \mid G) = \mathcal{L}_W(\varphi(S), G)$, such that:

$$\Lambda_{W^\varphi,n}(F_\varphi) = \sum_{G \in \mathbb{G}_n} \int_{F_\varphi(G)} \mathcal{L}_W(\varphi(S) \mid G) \, dS,$$

Since $M^\varphi(x, y) = M(\varphi(x), \varphi(y))$, we have

$$d_S \left( M^\varphi, \hat{M}^\varphi_{f(G) \circ S} \right) = d_{\varphi(S)} \left( M, \hat{M}_{f(G) \circ \varphi(S)} \right)$$

such that

$$F_\varphi = \left\{ (G, S) : d_{\varphi(S)} \left( M, \hat{M}_{f(G) \circ \varphi(S)} \right) > \epsilon \right\}.$$

Now consider the section of $F$ by $G$, defined by $F(G) = \{ S : (G, S) \in F \}$. It is clear that $F_\varphi(G) = \varphi^{-1}(F(G))$ for every graph $G$. Therefore,

$$\Lambda_{W^\varphi,n}(F_\varphi) = \sum_{G \in \mathbb{G}_n} \int_{\varphi^{-1}(F(G))} \mathcal{L}_W(\varphi(S) \mid G) \, dS.$$

Now, it is a property of measure preserving maps that $\int_{\varphi^{-1}(A)} f(\varphi(x)) \, d\mu(x) = \int_A f(x) \, d\mu(x)$; See, for example, [3]. Therefore, we have

$$\begin{aligned}
\Lambda_{W^\varphi,n}(F_\varphi) &= \sum_{G \in \mathbb{G}_n} \int_{\varphi^{-1}(F(G))} \mathcal{L}_W(\varphi(S) \mid G) \, dS \\
&= \sum_{G \in \mathbb{G}_n} \int_{F(G)} \mathcal{L}_W(S \mid G) \, dS \\
&= \Lambda_{W,n}(F)
\end{aligned}$$

which proves the claim.

$\square$

**Claim 5.** *Let $\mathbb{C}$ be cluster tree, and let $\tilde{\mathbb{C}}$ be a strict cluster tree obtained by applying a strict section $\tilde{\rho}$ to each cluster of $\mathbb{C}$. Let $M$ be the strict mergeon representing $\tilde{\mathbb{C}}$. Let $S = (x_1, \ldots, x_n)$ with each $x_i \in [0, 1]$ and suppose $\mathsf{C}$ is a clustering of $S$. Let $\hat{M}$ be the induced merge height on $\mathsf{C}$. If $d_S(M, \hat{M}) < \epsilon$, we then have:*

1. *Connectedness: If $C$ is a cluster of $\tilde{\mathbb{C}}$ at level $\lambda$ and $|S \cap C| \geq 2$, then the smallest cluster in $\mathsf{C}$ which contains all of $S \cap C$ is contained within $C' \cap S$, where $C'$ is the cluster of $\tilde{\mathbb{C}}$ at level $\lambda' = \lambda - \epsilon$ which contains $C$.*

2. *Separation: If $C_1$ and $C_2$ are two clusters of $\tilde{\mathbb{C}}$ at level $\lambda$ such that $C_1$ and $C_2$ merge at level $\lambda' < \lambda - \epsilon$, then if $|C_1 \cap S|, |C_2 \cap S| \geq 2$, the smallest cluster in $\mathsf{C}$ containing $C_1 \cap S$ and the smallest cluster containing $C_2 \cap S$ are disjoint.*

*Proof.* First we prove connectedness. Let $\hat{C}$ be the smallest cluster in $\tilde{\mathbb{C}}$ containing $C \cap S$. Suppose $\hat{C}$ contains a point $y$ from outside of $C'$. Let $x, x'$ be any two distinct points in $C \cap S$. Then necessarily $M(x, y) < \lambda' = \lambda - \epsilon$, as, since $M$ is strict, $M(x, y) \geq \lambda$ if and only if $x, y$ are in the same cluster of $\tilde{\mathbb{C}}$ at level $\lambda$. Hence the merge distortion is at least $M(x, x') - M(x, y) > \epsilon$, which is a contradiction.

Separation follows from connectedness. Let $\hat{C}_1$ be the smallest cluster in the clustering containing $C_1 \cap S$, and similarly for $\hat{C}_2$. Let $\tilde{C}_1$ and $\tilde{C}_2$ be the clusters at level $\lambda - \epsilon$ which contain $C_1$ and $C_2$. Then $\tilde{C}_1 \cap \tilde{C}_2 = \emptyset$, since $C_1$ and $C_2$ merge below $\lambda - \epsilon$. Furthermore, by connectedness, $C_1 \cap S \subset \tilde{C}_1$ and $C_2 \cap S \subset \tilde{C}_2$. Hence they are disjoint. $\qquad\square$

**Claim 6.** *Let $(\Omega, \Sigma, \mu)$ be a measure space, with $\mu$ a finite measure (i.e., $\mu(\Omega) < \infty$). Let $\mathfrak{A} \subset \Sigma$ be closed under countable unions. Define the set of* essential maxima *of $\mathfrak{A}$ by*

$$\text{ess max } \mathfrak{A} = \{M \in \mathfrak{A} : \mu(A \setminus M) = 0 \quad \forall A \in \mathfrak{A}\}.$$

*Then if $\mathfrak{A}$ is nonempty,* ess max $\mathfrak{A}$ *is nonempty. Furthermore, for any $M, M' \in$ ess max $\mathfrak{A}$, $\mu(M \triangle M') = 0$.*

*Proof.* The claim holds trivially if $\mathfrak{A}$ is empty, so suppose it is not. Let $\tau = \sup_{A \in \mathfrak{A}} \mu(A)$, and note that $\tau$ is finite since $\mu(\Omega)$ is finite. Then for every $n \in \mathbb{N}^+$, there exists a set $A_n \in \mathfrak{A}$ such that $\tau - \mu(A_n) < 1/n$. Construct the sequence $\langle B_n \rangle_{n \in \mathbb{N}^+}$ by defining $B_n = \bigcup_{i=1}^n A_i$. Then $B_n \in \mathfrak{A}$ for every $n$, since it is the countable union of sets in $\mathfrak{A}$. Furthermore, $B_n \subseteq B_{n+1}$, and $\lim_{n \to \infty} \mu(B_n) = \tau$. Define $M = \bigcap_{n=1}^\infty B_n = \bigcap_{n=1}^\infty A_n$. Then $M \in \mathfrak{A}$ since it is a countable union of elements of $\mathfrak{A}$, and by continuity of measure $\mu(M) = \tau$.

First we show that for any set $A \in \mathfrak{A}$, $\mu(A \setminus M) = 0$. Suppose for a contradiction that $\mu(A \setminus M) \neq 0$. We have $A \cup M = (A \setminus M) \cup M$, such that $\mu(A \cup M) = \mu(A \setminus M) + \mu(M) > \tau$. But $A \cup M$ is in $\mathfrak{A}$, since $\mathfrak{A}$ is closed under unions. This violates the fact that $\tau$ is the supremal measure of any set in $\mathfrak{A}$, and hence it must be that $\mu(A \setminus M) = 0$. Therefore $M \in$ ess max $\mathfrak{A}$.

Now suppose $M'$ is an arbitrary element of ess max $\mathfrak{A}$. We have just seen that $\mu(M' \setminus M)$ must be zero, since $M' \in \mathfrak{A}$. Likewise, $\mu(M \setminus M') = 0$. Therefore

$$\mu(M \triangle M') = \mu((M \setminus M') \cup (M' \setminus M)) = \mu(M \setminus M') + \mu(M' \setminus M) = 0$$

where we used the fact that $\mu$ is an additive set function and $M \setminus M'$ and $M' \setminus M$ disjoint. It is also clear that if $M \in$ ess max $\mathfrak{A}$ and $N$ is any null set, then $M \cup N$ and $M \setminus N$ are also essential maxima. $\qquad\square$

**Claim 7.** *Let $(\Omega, \Sigma, \mu)$ be a measure space, with $\mu$ a finite measure (i.e., $\mu(\Omega) < \infty$). Let $\mathfrak{A} \subset \Sigma$ be closed under countable intersections. Define the set of* essential minima *of $\mathfrak{A}$ by*

$$\text{ess min } \mathfrak{A} = \{M \in \mathfrak{A} : \mu(M \setminus A) = 0 \quad \forall A \in \mathfrak{A}\}.$$

*Then if $\mathfrak{A}$ is nonempty,* ess min $\mathfrak{A}$ *is nonempty. Furthermore, for any $M, M' \in$ ess min $\mathfrak{A}$, $\mu(M \triangle M') = 0$.*

*Proof.* The proof is analogous the that of Claim 6 for ess max – we simply switch $\tau$ from a supremum to an infimum and construct a descending sequence. It is therefore omitted.

$\qquad\square$

**Claim 8.** *Let $W$ be a graphon, and suppose $A$ and $A'$ are measurable sets with positive measure, and that each is connected at level $\lambda$ in $W$. If $\mu(A \cap A') > 0$, then $A \cup A'$ is connected at level $\lambda$.*

*Proof.* Suppose $\mu(A \cap A') > 0$ and that $A \cup A'$ is disconnected at level $\lambda$. Then, by definition, there exists a measurable set $S \subset A \cup A'$ such that $0 < \mu(S) < \mu(A \cup A')$ and $W < \lambda$ almost everywhere on $S \times ((A \cup A') \setminus S)$.

It is either the case that $0 < \mu(A \cap S) < \mu(A)$ or $0 < \mu(A \cap S') < \mu(S')$, as otherwise we would have $\mu(S) = \mu(A \cup A')$. If $0 < \mu(A \cap S) < \mu(A)$, then $A \setminus S$ is of positive measure. Since $W < \lambda$ almost everywhere on $S \times ((A \cup A') \setminus S)$, it follows that $W < \lambda$ almost everywhere on $S \times (A \setminus S)$. This implies that $A$ is not connected at level $\lambda$. Likewise, if $0 < \mu(A' \cap S) < \mu(A')$, it follows that $W < \lambda$ almost everywhere on $S \times (A' \setminus S)$, and hence $A'$ is disconnected at level $\lambda$. Both cases lead to contradictions, and so it must be that $A \cup A'$ is connected at level $\lambda$. $\qquad\square$

**Claim 9.** *The relation $\multimap_\lambda$ is an equivalence relation on $\mathfrak{A}_\lambda$.*

*Proof.* The symmetry and reflexive properties of $\multimap_\lambda$ are clear. We need only prove that $\multimap_\lambda$ is transitive. Suppose $A_1 \multimap_\lambda A_2$ and $A_2 \multimap_\lambda A_3$. By the definition of $\multimap_\lambda$, there exist measurable sets $B_{12}, B_{23} \in \mathfrak{A}_\lambda$ such that $B_{12} \supset A_1 \cup A_2$ and $B_{23} \supset A_2 \cup A_3$. Since both $B_{12}$ and $B_{23}$ contain $A_2$, a set of positive measure, their intersection is not null. Furthermore, $B_{12}$ and $B_{23}$ are each connected at every level $\lambda' < \lambda$, by virtue of being in $\mathfrak{A}_\lambda$. Hence we can use Claim 8 in Appendix B to conclude that their union $B_{12} \cup B_{23}$ is connected at every level $\lambda' < \lambda$, and is hence an element of $\mathfrak{A}_\lambda$. Since $A_1 \cup A_3 \subset B_{12} \cup B_{23}$, we have $A_1 \multimap_\lambda A_3$.

$\square$

**Claim 10.** *Let $\mathscr{C}$ be an equivalence class in $\mathfrak{A}_\lambda / \multimap_\lambda$. Then $\operatorname{ess\,max} \mathscr{C}$ is well-defined and non-empty.*

*Proof.* We will invoke Claim 6 in Appendix B to show that $\operatorname{ess\,max} \mathscr{C}$ has the desired properties. To do so, we need only show that $\mathscr{C}$ is closed under countable unions. Let $\mathscr{F} \subset \mathscr{C}$ be a countable subset of $\mathscr{C}$, and let $F = \bigcup \mathscr{F}$. We will show that $F$ is connected at every level $\lambda' < \lambda$ and is thus contained in $\mathscr{F}$ by its definition.

Suppose $F$ is disconnected at some level $\lambda' < \lambda$. Then there exists a set $S \subset F$ such that $0 < \mu(S) < \mu(F)$ and $W < \lambda'$ almost everywhere on $S \times (F \setminus S)$. Now, there must exist sets $F_1, F_2 \in \mathscr{F}$ such that $\mu(S \cap F_1) > 0$ and $\mu((F \setminus S) \cap F_2) > 0$. Let $F_{12} = F_1 \cup F_2$. Note that $\mathscr{F}$ is closed under finite union by the definition of $\multimap_\lambda$, and so $F_{12} \in \mathscr{F}$, meaning that $F_{12}$ is connected at $\lambda'$. Furthermore, $F_{12} \subset F$, so that $(F_{12} \cap S) \cup (F_{12} \cap (F \setminus S)) = F_{12}$. But by assumption $W < \lambda'$ almost everywhere on $S \times (F \setminus S)$, and so in particular $W < \lambda'$ almost everywhere on $(F_{12} \cap S) \times (F_{12} \cap (F \setminus S))$. But this implies that $F_{12}$ is disconnected at level $\lambda'$, which is a contradiction. It must therefore be the case that $F$ is connected at every $\lambda' < \lambda$, and so $F \in \mathfrak{A}_\lambda$.

It is clearly true that for all $F' \in \mathscr{F}$, $F' \multimap_\lambda F$, since $F = F \cup F'$. We therefore have that $F \in \mathscr{F}$.

$\square$

**Claim 11.** *Let $W$ be a graphon, and suppose $C_1, C_2 \in \mathbb{C}_W(\lambda)$ are two distinct clusters at level $\lambda$, such that $\mu(C_1 \cap C_2) = 0$. Let $M$ be a mergeon of $W$. Then $M < \lambda$ almost everywhere on $C_1 \times C_2$.*

*Proof.* Consider the set

$$(C_1 \times C_2) \cap M^{-1}[\lambda, 1] = (C_1 \times C_2) \cap \left( \bigcup_{C \in \mathbb{C}_W(\lambda)} C \times C \right)$$

$$= \bigcup_{C \in \mathbb{C}_W(\lambda)} (C_1 \times C_2) \cap (C \times C)$$

$$= \bigcup_{C \in \mathbb{C}_W(\lambda)} (C_1 \cap C) \times (C_2 \cap C)$$

Because distinct clusters intersect only on a null set, we can have either $C_1 \cap C$ or $C_2 \cap C$ being non-negligible, but not both simultaneously for the same choice of $C$. Hence every set in the union is a null set, and since $\mathbb{C}_W(\lambda)$ is countable, the union as a whole is a null set. Therefore $\mu \times \mu(M^{-1}[\lambda, 1] \cap (C_1 \times C_2)) = 0$, and so it must be that $M < \lambda$ almost everywhere on $C_1 \times C_2$. $\square$

**Claim 12.** *Let $W$ be a graphon and suppose $M$ is a mergeon of $W$. Suppose $A$ is connected at level $\lambda$ in $M$. Then $A$ is contained in some cluster $C$ in $W$ at level $\lambda$.*

*Proof.* First, it must be the case that $\mu(A \setminus \bigcup \mathbb{C}_W(\lambda)) = 0$. Suppose not. Let $R = A \setminus \bigcup \mathbb{C}_W(\lambda)$. Since $A$ is connected, it follows that there is a subset $Q \subset R \times (A \setminus R)$ of positive measure such that $M \geq \lambda$ on $Q$. We then have that

$$Q \cap M^{-1}[\lambda, 1] = Q \cap \bigcup_{C \in \mathbb{C}_W(\lambda)} C \times C$$

is not null. Since $\mathbb{C}_W(\lambda)$ is a countable set, it follows that there must be a cluster $C \in \mathbb{C}_W(\lambda)$ such that $Q \cap (C \times C)$ is not null. But $Q \subset R \times (A \setminus R)$, so this implies that $(R \times (A \setminus R)) \cap (C \times C)$ is not null. We have the identity:

$$(R \times (A \setminus R)) \cap (C \times C) = (R \cap C) \times ((A \setminus R) \cap C),$$

which then implies that $(R \cap C) \times ((A \setminus R) \cap C)$ is not null. However, this is a contradiction, since $R \cap C$ is necessarily a set of measure zero by the definition of $R$. Hence it must be that all of $A$ excluding a null set is contained within $\bigcup \mathbb{C}_W(\lambda)$.

Let $\hat{A} = A \cap \bigcup \mathbb{C}_W(\lambda)$. Then $\hat{A}$ is equivalent to $A$ in that it differs only by a set of measure zero, however, it is contained entirely within $\bigcup \mathbb{C}_W(\lambda)$. Since $\hat{A}$ is a set of positive measure, there must exist a $\hat{C} \in \mathbb{C}_W(\lambda)$ such that $\mu(\hat{A} \cap \hat{C}) > 0$. We will show that $\mu(\hat{A} \setminus \hat{C}) = 0$.

Let $S = \hat{A} \cap \hat{C}$, and let $T = \hat{A} \setminus S$. Note that $T \cap \hat{C}$ is null. Suppose for a contradiction that $T$ is not null. Since $T$ is contained within $\bigcup \mathbb{C}_W(\lambda)$, we may decompose it as the union

$$T = \bigcup_{C \in \mathbb{C}_W(\lambda)} T \cap C$$

hence we have

$$
\begin{aligned}
S \times T &= \bigcup_{C \in \mathbb{C}_W(\lambda)} S \times (T \cap C) \\
&= \bigcup_{C \in \mathbb{C}_W(\lambda)} (\hat{A} \cap \hat{C}) \times (T \cap C) \\
&= \bigcup_{C \in \mathbb{C}_W(\lambda)} (\hat{A} \times T) \cap (\hat{C} \times C).
\end{aligned}
$$

But $M < \lambda$ almost everywhere on $\hat{C} \times C$ whenever $C$ and $\hat{C}$, are disjoint clusters. Hence $M^{-1}[\lambda, 1] \cap (S \times T)$ is equal, up to a null set, to the set $M^{-1}[\lambda, 1] \cap (\hat{A} \times T) \cap (\hat{C} \times \hat{C})$. Using the identity once again, this is the set $M^{-1}[\lambda, 1] \cap \left[ (\hat{A} \cap \hat{C}) \times (\hat{C} \cap T) \right]$. But $\hat{C} \cap T$ is null, so that this set is null. This is a contradiction, as it implies that $M < \lambda$ almost everywhere on $S \times T$, but $S \cup T = \hat{A}$ is connected at level $\lambda$. Therefore it must be that $T$ is null, and hence $\mu(\hat{A} \setminus \hat{C}) = 0$. This implies that $\mu(A \setminus \hat{C}) = 0$, and so $A$ is contained in some cluster of $W$ at level $\lambda$, namely $C$. □

**Claim 13.** *Suppose a set $A$ of positive measure is contained in a cluster at every level $\lambda' < \lambda$. Then $A$ is contained in a cluster at level $\lambda$.*

*Proof.* We may construct a sequence $C_1, C_2, \ldots$ of clusters such that $C_i$ is a cluster at level $\lambda - 1/n$, and $C_i$ contains $A$. Then the intersection $C = \bigcap_{i=1}^{\infty} C_i$ is connected at all levels $\lambda' < \lambda$, as otherwise there would exist a $\lambda^* < \lambda$ at which $C$ is disconnected, but this would imply that $C_i$ is disconnected for any $i$ such that $\lambda - 1/i > \lambda^*$. Furthermore, $C$ has positive measure, since the measure of every $C_i$ is at least $\mu(A)$. Therefore, $C$ is contained in some cluster at level $\lambda$, and $C$ contains $A$. Hence $A$ is in some cluster at level $\lambda$. □

**Claim 14.** *Let $\mathbb{C}$ be a cluster tree. There exists a section function $\tilde{\rho}$ on $\mathbb{C}$ such that if*

1. *$\mu(\mathscr{C} \cap \mathscr{C}') = 0 \Rightarrow \tilde{\rho}(\mathscr{C}) \cap \tilde{\rho}(\mathscr{C}) = \emptyset$,*

2. *$\mathscr{C} \subset \mathscr{C}' \Rightarrow \tilde{\rho}(\mathscr{C}) \subset \tilde{\rho}(\mathscr{C}')$, and*

3. *(Technical condition) $\tilde{\rho}(\mathscr{C}) = \bigcap \{ \tilde{\rho}(\mathscr{C}') : \mathscr{C}' \supset \mathscr{C} \}$.*

*Proof.* We construct such a section function on the clusters at rational levels and extend it to $[0, 1]$. Let $\mathbb{Q}_{[0,1]} = \mathbb{Q} \cap [0, 1]$. Define

$$\hat{\mathbb{C}} = \{ \mathscr{C} \in \mathbb{C}(\lambda) : \lambda \in \mathbb{Q}_{[0,1]} \}$$

that is, $\hat{\mathbb{C}}$ is the set of all clusters from every rational level. Note that this is a countable collection. For any cluster $\mathscr{C} \in \hat{\mathbb{C}}$, define $P_{\mathscr{C}}$ to be the set of clusters in $\hat{\mathbb{C}}$ which have null intersection with $\mathscr{C}$. That is:

$$P_{\mathscr{C}} = \{ \mathscr{C}' \in \hat{\mathbb{C}} : \mu(\mathscr{C} \cap \mathscr{C}') = 0 \}.$$

Let $\rho_0$ be an arbitrary section function, and define the section function $\rho_1 : \hat{\mathbb{C}} \to \Sigma$ as follows:

$$\rho_1(\mathscr{C}) = \rho_0(\mathscr{C}) \setminus \bigcup P_\mathscr{C}.$$

Furthermore, let $\mathscr{C}_0$ be the equivalence class of sets differing from $[0,1]$ by a null set, and define $\rho_1(\mathscr{C}_0) = [0,1]$; This will ensure that all pairs of points have a well-defined merge height. The intersection of $\rho_0(\mathscr{C})$ and $\bigcup P_\mathscr{C}$ is null, by definition of $P_\mathscr{C}$ and the fact that it is a countable set. Therefore, $\rho_1(\mathscr{C}) \triangle \rho_0(\mathscr{C})$ is null, and $\rho_1(\mathscr{C})$ is hence a valid representative of $\mathscr{C}$. Furthermore, for any $\mathscr{C}, \mathscr{C}' \in \hat{\mathbb{C}}$ such that $\mu(\mathscr{C} \cap \mathscr{C}') = 0$, we have $\rho_1(\mathscr{C}) \cap \rho_2(\mathscr{C}') = \emptyset$.

We now define the section function on all levels in $[0,1]$. For a cluster $\mathscr{C}$ at any level, define its set of ancestors in $\hat{\mathbb{C}}$ to be

$$\mathfrak{A}_\mathscr{C} = \{\mathscr{C}' \in \hat{\mathbb{C}} : \mu(\mathscr{C}' \setminus \mathscr{C}) = 0\}.$$

Then define

$$\tilde{\rho}(\mathscr{C}) = \bigcap_{\mathscr{C}' \in \mathfrak{A}_\mathscr{C}} \rho_1(\mathscr{C}')$$

Hence $\tilde{\rho}$ trivially satisfies the third condition of the claim.

We must argue that $\tilde{\rho}(\mathscr{C})$ is a valid representative of $\mathscr{C}$. First, suppose that $\mathscr{C}$ is a cluster at level $\lambda$. Then $\mathfrak{A}_\mathscr{C}$ contains a cluster from every rational level below $\lambda$, and $\tilde{\rho}(\mathscr{C})$ is contained in a representative of each of them. It follows that $\tilde{\rho}(\mathscr{C})$ is contained in a cluster representative at every level $\lambda' < \lambda$. Hence, by Claim 13 in Appendix B, $\tilde{\rho}(\mathscr{C})$ is contained in a cluster representative at level $\lambda$. But $\mathscr{C}$ is essentially contained in all of its ancestors. Therefore, it must be that $\tilde{\rho}(\mathscr{C}) \triangle \mathscr{C}$ is null and so $\tilde{\rho}(\mathscr{C})$ is a valid representative of $\mathscr{C}$.

Now we show that $\tilde{\rho}$ has the desired properties. Suppose $\mathscr{C}$ and $\mathscr{C}'$ have null intersection, and without loss of generality, assume that they are clusters at the same level $\lambda$. Let $\lambda' < \lambda$ be the maximal level at which their intersection is not null. Then there is some rational level $\tilde{\lambda}$ between $\lambda'$ and $\lambda$. Hence $\tilde{\rho}(\mathscr{C})$ is strictly contained in the representative $\rho_1(\tilde{\mathscr{C}})$ of some cluster $\tilde{\mathscr{C}}$ at level $\tilde{\lambda}$, and similarly, $\tilde{\rho}(\mathscr{C})$ is strictly contained in $\rho_1(\tilde{\mathscr{C}}')$ at the same level. Necessarily, $\tilde{\mathscr{C}}$ and $\tilde{\mathscr{C}}'$ have null intersection, and so $\rho_1(\tilde{\mathscr{C}})$ and $\rho_1(\tilde{\mathscr{C}}')$ are strictly disjoint. Therefore, so also are $\tilde{\rho}(\mathscr{C})$ and $\tilde{\rho}(\mathscr{C}')$.

Furthermore, suppose that $\mathscr{C}$ and $\mathscr{C}'$ are such that $\mu(\mathscr{C}' \setminus \mathscr{C}) = 0$. Suppose without loss of generality that $\lambda > \lambda'$ (if $\lambda = \lambda'$ then $\tilde{\rho}(\mathscr{C}) = \tilde{\rho}(\mathscr{C}')$). Then the ancestors of $\mathscr{C}$ include the ancestors of $\mathscr{C}'$, and so the intersection of the ancestors of $\mathscr{C}$ is a subset of the intersection of the ancestors of $\mathscr{C}'$. This proves that $\tilde{\rho}(\mathscr{C}) \subset \tilde{\rho}(\mathscr{C}')$.

$\square$

## C    Clusters under measure preserving transformations

In this section we show that there is a bijection between the clusters of two weakly isomorphic graphons. In particular, we show that if $\varphi$ is a measure preserving transformation, then $\mathscr{C}$ is a cluster of $W^\varphi$ at level $\lambda$ if and only if there exists a cluster $\mathscr{C}'$ at level $\lambda$ of $W$ such that $\mathscr{C} = \varphi^{-1}(\mathscr{C}')$. This is made non-trivial by the fact that a measure preserving transformation is in general not injective. For instance, $\varphi(x) = 2x \mod 1$ defines a measure preserving transformation, but is not an injection. Even worse, it is possible for a measure preserving transformation to map a set of zero measure to a set of positive measure – it is only the measure of the *preimage* which must be preserved.

### C.1    Claims

We will mitigate the fact that $\varphi$ may not be injective by working whenever possible with sets whose image is necessarily stable under even non-injective measure preserving transformations, in the sense that $\varphi^{-1}(\varphi(A)) = A$. We will show that such stability is a property of sets which contain all of their so-called *twin* points, defined as follows. Two points $x$ and $x'$ are *twins* in $W$ if $W(x, y) = W(x', y)$ for almost every $y \in [0, 1]$. We say that a set $A$ *separates* twins if there exist twins $x$ and $x'$ such that $x \in A$ and $x' \notin A$. The relation of being twins is an equivalence relation on $[0, 1]$.

We will define a probability space on the equivalence classes of the twin relation as follows (see [15] for the full construction):

**Definition 11.** *Let $W$ be a graphon. The* twin measure space *$(\Omega_W, \Sigma_W, \mu_W)$ is defined as follows. Let $\Omega_W$ be the set of equivalence classes under the twin relation in $W$, and let $\psi_W(x)$ denote the equivalence class in $\Omega_W$ containing $x$. If $\Sigma$ is the sigma-algebra of Lebesgue measurable subsets of $[0, 1]$, create a new sigma-algebra by defining*

$$\Sigma_W = \{\psi_W(X) : X \in \Sigma, X \text{ does not separate twins in } W\}.$$

*Furthermore, we define the measure $\mu_W(A) = \mu(\psi_W^{-1}(A))$ for $A \in \Sigma_W$. It can be shown that, with this measure, $\psi_W$ is measure preserving.*

We note in passing that the random graph model defined by any graphon $W$ can also be represented as a $\Sigma_W$-measurable function $W_T : \Omega_W \times \Omega_W \to [0, 1]$ defined on the probability space $(\Omega_W, \Sigma_W, \mu_W)$, as is shown by [15]. $W_T$ is called a "twin-free" graphon, since no two points in $\Omega_W$ are twins in $W_T$. In this representation, two twin-free graphons are weakly isomorphic if there exists a measure preserving *bijection* relating them. Our definitions of connectedness, clusters, mergeons, etc. can be formulated for twin-free graphons with minor modifications, and the existence of the measure preserving bijection between twin-free graphons means that clusters transfer trivially between weakly isomorphic graphons. In a sense, the twin-free setting is a more natural one for the considerations of the current section; We leave a more in-depth investigation of this direction to future work.

We now prove some useful properties of the map $\psi_W$.

**Claim 15.** *Suppose $A \subset [0, 1]$ does not separate twins in $W$. Then $\psi_W^{-1}(\psi_W(A)) = A$ and $A$ is $\Sigma$-measurable.*

*Proof.* It is clear that $A \subset \psi_W^{-1}(\psi_W(A))$. Now let $x \in \psi_W^{-1}(\psi_W(A))$. Then there exists a $y \in A$ such that $\psi_W(x) = \psi_W(y)$. But then $x$ and $y$ are twins in $W^\varphi$. Since $A$ does not separate twins, $x \in A$, proving that $A = \psi_W^{-1}(\psi_W(A))$.

Now we prove that $A$ is $\Sigma$-measurable. $\psi_W$ is a measurable function, and so the inverse image of any $\Sigma_W$-measurable set is $\Sigma$-measurable. We have that $\psi_W(A)$ is $\Sigma_W$-measurable, since $A$ does not separate twins. Hence $\psi_W^{-1}(\psi_W(A)) = A$ is $\Sigma$-measurable. $\qquad\square$

**Claim 16.** *Let $W$ be a graphon and let $A \subset [0, 1]$. Then $\psi_W^{-1}(\psi_W(A))$ is $\Sigma$-measurable.*

*Proof.* It is clear that $\psi_W^{-1}(\psi_W(A))$ does not separate twins in $W$. Hence it is $\Sigma$-measurable by the previous claim. $\qquad\square$

**Claim 17.** *Let $W$ be a graphon and $\varphi$ a measure preserving transformation. Suppose $A$ does not separate twins in $W^\varphi$. Then*

1. *$\varphi^{-1}(\varphi(A)) = A$, and*

2. *$\mu(\varphi(A)) = \mu(A)$.*

*Proof.* For the first claim, we know that $A \subseteq \varphi^{-1}(\varphi(A))$. Now we show the other inclusion. Let $x \in \varphi^{-1}(\varphi(A))$. Then there exists an $x'$ in $\varphi(A)$ such that $\varphi(x) = \varphi(x')$. But then $x$ and $x'$ are twins, such that $x$ and $x'$ are both in $A$. Hence $x \in A$, proving the claim. The second claim follows immediately since $\varphi$ is measure preserving. That is, $\mu(\varphi^{-1}(\varphi(A))) = \mu(\varphi(A))$, but since $\varphi^{-1}(\varphi(A)) = A$, $\mu(\varphi(A)) = \mu(A)$. $\qquad\square$

Therefore, sets which do not separate twins are stable under measure preserving transformations. An arbitrary set $C$ may separate twins, however we can always find a set containing all of $C$ except for a null set, and which does not separate twins. We call the smallest such set the *family* of $C$.

**Definition 12.** *Let $W$ be a graphon and let $(\Omega_W, \Sigma_W, \mu_W)$ be the corresponding twin measure space for $W$. For any $\Sigma$-measurable set $C$, construct the collection*

$$\mathcal{F}_C = \{A \in \Sigma_W : \mu(C \setminus \psi_W^{-1}(A)) = 0\}.$$

*We define the* family *of $C$, written $\mathsf{Fam}_W\, C$, as*

$$\mathsf{Fam}_W\, C = \{\psi_W^{-1}(X) : X \in \operatorname{ess\,min} \mathcal{F}_C\}.$$

Recall that the ess min $\mathfrak{A}$ of a collection of sets $\mathfrak{A}$ is defined to be the set

$$\text{ess min } \mathfrak{A} = \{M \in \mathfrak{A} : \mu(M \setminus A) = 0 \quad \forall A \in \mathfrak{A}\}.$$

See Claim 7 in Appendix B for the properties of the essential minima of a class of sets.

It is clear that $\mathsf{Fam}_W C$ cannot be empty, as $\psi_W^{-1}(\Omega_W)$ must contain almost all of $C$. To be rigorous, we must argue that $\mathcal{F}_C$ is closed under countable intersections so that it has a well-defined set of essential minima. To see this, let $\mathcal{F}$ be any countable subset of $\mathcal{F}_C$. Define $D = \bigcap \mathcal{F}$. Then $D \in \Sigma_W$ since it is a sigma-algebra, and we have

$$\begin{aligned}
\mu\left(C \setminus \psi_W^{-1}(D)\right) &= \mu\left(C \setminus \psi_W^{-1}\left(\bigcap \mathcal{F}\right)\right), \\
&= \mu\left(C \setminus \bigcap_{F \in \mathcal{F}} \psi_W^{-1}(F)\right), \\
&= \mu\left(\bigcup_{F \in \mathcal{F}} C \setminus F\right), \\
&= 0,
\end{aligned}$$

where the last step follows because each $F$ has the property that $C \setminus F$ is a null set, and the union of countably many null sets is null. Hence $D \in \mathcal{F}_C$.

Also note that for any $A, A' \in \mathsf{Fam}_W C$, it must be that $\mu(A \triangle A') = 0$. This is because $A = \psi_W^{-1}(B)$ and $A' = \psi_W^{-1}(B')$ for some $B, B' \in \text{ess min } \mathcal{F}_C$. As seen above, $\mu_W(B \triangle B') = 0$. Since $\psi_W^{-1}(B \setminus B') = \psi_W^{-1}(B) \setminus \psi_W^{-1}(B')$, it must be that $\psi_W^{-1}(B) \triangle \psi_W^{-1}(B') = A \triangle A'$ is a null set. Furthermore, it is clear that for any $A \in \mathsf{Fam}_W C$, $A$ does not separate twins in $W$.

A key result is that the family of any representative of a cluster differs from the representative by a null set, and is therefore itself a representative. That is, we can always find a representative of a cluster which does not separate twins.

**Claim 18.** *Let $W$ be a graphon and suppose $\mathscr{C}$ is a cluster at level $\lambda$ in $W$. Let $\bar{C} \in \mathsf{Fam}\,\mathscr{C}$. Then $\mu(\mathscr{C} \triangle \bar{C}) = 0$.*

*Proof.* Take an arbitrary representative $C$ of $\mathscr{C}$. Let $(\Omega_W, \Sigma_W, \mu_W)$ be the twin measure space as defined above.

We know that $C \setminus \bar{C}$ is a null set, so we need only show that $\bar{C} \setminus C$ is null. Suppose otherwise. That is, let $R = \bar{C} \setminus C$ and suppose $\mu(R) > 0$. Let $A$ be any subset of $R$ with positive measure. There are two cases: (1) For some $\lambda' < \lambda$, $W < \lambda'$ almost everywhere on $A \times (\bar{C} \setminus A)$, or (2) for every $\lambda' < \lambda$, $W \geq \lambda'$ on some subset of $A \times (\bar{C} \setminus A)$ of positive measure.

Suppose case (1) holds for some $\lambda'$. Then for almost all $a \in A$ it is true that $W(a, y) < \lambda'$ for almost every $y \in \bar{C} \setminus A$. That is, let

$$\hat{A} = \{a \in A : W(a, y) < \lambda' \text{ for almost every } y \in \bar{C} \setminus A\}.$$

Then $\mu(\hat{A}) = \mu(A)$ and $W < \lambda$ almost everywhere on $\hat{A} \times (\bar{C} \setminus \hat{A})$. Define $\bar{A} = \psi_W^{-1}(\psi_W(\hat{A}))$. There are two subcases: Either (1a) $\bar{A} \cap C$ is null, or (1b) it is of positive measure.

Consider the first subcase. Define $D = \psi_W(\bar{C}) \setminus \psi_W(\bar{A})$. We will show that $\psi_W^{-1}(D)$ contains $C$ except for a set of zero measure, and so $\psi_W^{-1}(D) \in \mathsf{Fam}_W C$. But as we will see, $\mu(\psi_W^{-1}(D)) < \mu(\bar{C})$, which cannot be. We have

$$\begin{aligned}
\psi_W^{-1}(D) &= \psi_W^{-1}(\psi_W(\bar{C}) \setminus \psi_W(\bar{A})) \\
&= \psi_W^{-1}(\psi_W(\bar{C})) \setminus \psi_W^{-1}(\psi_W(\bar{A})) \\
&= \bar{C} \setminus \bar{A}
\end{aligned}$$

where the last step follows since $\bar{C}$ and $\bar{A}$ do not separate twins. Therefore,

$$\begin{aligned}
C \cap \psi_W^{-1}(D) &= C \cap (\bar{C} \setminus \bar{A}) \\
&= (C \cap \bar{C}) \cup (C \cap \bar{A})
\end{aligned}$$

But $C \cap \bar{A}$ is a null set, so $\mu(C \cap \psi_W^{-1}(D)) = \mu(C \cap \bar{C}) = \mu(C)$. This implies that $\mu(C \setminus \psi_W^{-1}(D)) = 0$, and hence $\psi_W^{-1}(D) \in \mathsf{Fam}_W\, C$. But $\mu_W(D) = \mu_W(\psi_W(\bar{C}) \setminus \psi_W(\bar{A}))$, and $\psi_W(\bar{A}) \subset \psi_W(\bar{C})$ with $\mu_W(\psi_W(\bar{A})) = \mu(\bar{A}) > 0$. Therefore, $\mu_W(D) < \mu_W(\psi_W(\bar{C}))$, and so $\mu(\psi_W^{-1}(D)) < \mu(C)$. This cannot be, since all elements of $\mathsf{Fam}_W\, C$ differ only by null sets. Hence it cannot be that $\bar{A} \cap C$ is null.

Suppose case (1b) holds, then. That is, suppose $\bar{A} \cap C$ is not null. Then for every $x \in \bar{A} \cap C$ it is true that $W(x, y) < \lambda'$ for almost all $y \in \bar{C} \setminus A$. In particular, since $C \setminus (A \cap C) \subset \bar{C} \setminus A$, we have that $W < \lambda'$ almost everywhere on $(\bar{A} \cap C) \times (C \setminus \bar{A})$. This means that $C$ is disconnected at level $\lambda'$, which violates the assumption that $C$ is a cluster at $\lambda > \lambda'$.

Both subcases lead to contradictions, and so (1) cannot hold. Therefore, it must be that case (2) holds: For every $\lambda' < \lambda$, $W \geq \lambda'$ on some subset of $A \times (\bar{C} \setminus A)$. Furthermore, this must hold for arbitrary $A \subset R$ with positive measure. This implies that $\bar{C}$ is connected at every level $\lambda' < \lambda$, and hence part of a cluster at level $\lambda$. To see this, let $S, T \subset \bar{C}$ such that $S$ has positive measure and $S \cup T = \bar{C}$. Without loss of generality, assume $A \cap S$ is not null – if it is, swap $S$ and $T$. Then $T \cap (\bar{C} \setminus A)$ is not null. Therefore $W \geq \lambda$ on some subset of $S \times T$ with positive measure – namely, $(S \cap A) \times (T \cap (\bar{C} \setminus A))$. Since this holds for arbitrary $S$ and $T$, $\bar{C}$ is connected.

Therefore, both cases lead to contradictions under the assumption that $\mu(R) > 0$. Hence $\mu(R) = 0$, and $\mu(C \triangle \bar{C}) = 0$. $\qquad\square$

The previous claim shows that any cluster has a representative $C$ which does not separate twins, and so $\varphi^{-1}(\varphi(C)) = C$. The next claim shows that there exists a (possibly different) cluster representative $C'$ such that $\varphi(\varphi^{-1}(C')) = C'$.

**Claim 19.** *Let $W$ be a graphon and $\varphi$ a measure preserving transformation. Suppose $\mathscr{C}$ is a cluster of $W$. There exists a representative $C$ of $\mathscr{C}$ such that $\varphi(\varphi^{-1}(C)) = C$.*

*Proof.* First, let $\bar{C} = \mathsf{Fam}_W\, \mathscr{C}$, such that $\bar{C}$ is a representative of $\mathscr{C}$ which does not separate twins. Then $\varphi^{-1}(\bar{C})$ does not separate twins in $W^\varphi$, and so by Claim 17, $\mu(\varphi(\varphi^{-1}(\bar{C}))) = \mu(\bar{C})$. But $\bar{C} \supset \varphi(\varphi^{-1}(\bar{C}))$, such that $\bar{C} \triangle \varphi(\varphi^{-1}(\bar{C})) = 0$. Hence $\varphi(\varphi^{-1}(\bar{C}))$ is a representative of the cluster $\mathscr{C}$. Furthermore, $\varphi^{-1}(\varphi(\varphi^{-1}(\bar{C}))) = \varphi^{-1}(\bar{C})$, such that, defining $C = \varphi(\varphi^{-1}(\bar{C}))$, we have $\varphi(\varphi^{-1}(C)) = C$, as claimed. $\qquad\square$

Recall that a set $C$ is disconnected at level $\lambda$ in a graphon $W$ if there exists a subset $A \subset C$ with $0 < \mu(A) < \mu(C)$ such that $W < \lambda$ almost everywhere on $A \times (C \setminus A)$. It is of course possible, however, that $A$ might separate twins – even if $C$ does not. Therefore we cannot use the above claims to manipulate $A$. The next claim says that if a set $C$ which does not separate twins is disconnected at some level, it is always disconnected by a set $\bar{A}$ which also does not separate twins.

**Claim 20.** *Let $W$ be a graphon and suppose that $C$ is a set of positive measure that does not separate twins. If $C$ is disconnected at level $\lambda$ in $W$, then either $W < \lambda$ almost everywhere on $C \times C$, or there exists a set $\bar{A} \subset C$ such that $\bar{A}$ does not separate twins, $0 < \mu(\bar{A}) < \mu(C)$, and $W < \lambda$ almost everywhere on $\bar{A} \times (C \setminus \bar{A})$.*

*Proof.* Since $C$ is disconnected at level $\lambda$, there exists a subset $S \subset C$ such that $0 < \mu(S) < \mu(C)$ and $W < \lambda$ almost everywhere on $S \times (C \setminus S)$. Define

$$\hat{S} = \{x \in S : W(x, y) < \lambda \text{ for almost every } y \in C \setminus S\}.$$

Since $W < \lambda$ almost everywhere on $S \times (C \setminus S)$, it must be that $\mu(\hat{S}) = \mu(S)$; This is an application of Fubini's theorem. Let $\bar{S} = \psi_W^{-1}(\psi_W(\hat{S}))$. It follows that $\bar{S} \subset C$, and for every $x \in \bar{S}$, $W(x, y) < \lambda$ for almost every $y \in C \setminus S$. Furthermore, $\bar{S}$ does not separate twins, and $\bar{S}$ contains $\hat{S}$ – which is $S$, less a null set – so $\mu(S \setminus \bar{S}) = 0$.

There are two cases: $\mu(\bar{S}) < \mu(C)$, or $\mu(\bar{S}) = \mu(C)$. Suppose the first case holds. Then, since $\mu((C \setminus S) \setminus (C \setminus \bar{S})) = 0$, we have that for every $x \in \bar{S}$, $W(x, y) < \lambda$ for almost every $y \in C \setminus \bar{S}$. Therefore, $W < \lambda$ almost everywhere on $\bar{S} \times (C \setminus \bar{S})$. This proves the claim for the first case, as we may take $\bar{A} = \bar{S}$.

Now suppose $\mu(\bar{S}) = C$, which is to say that $\bar{S}$ differs from $C$ by a null set. Since $W < \lambda$ almost everywhere on $\bar{S} \times (C \setminus S)$, it follows that $W < \lambda$ almost everywhere on $C \times (C \setminus S)$. By symmetry

of $W$, we have $W < \lambda$ almost everywhere on $(C \setminus S) \times C$. This means that $W < \lambda$ almost everywhere on $(C \times C) \setminus (S \times S)$.

Let $T = C \setminus S$. Then $W < \lambda$ almost everywhere on $T \times C = (C \setminus S) \times C$. Define

$$\hat{T} = \{x \in T : W(x, y) < \lambda \text{ for almost every } y \in C \}.$$

Let $\bar{T} = \psi_W^{-1}(\psi_W(\hat{T}))$. Then, by a similar argument used above for $\bar{S}$, $\mu(T \setminus \bar{T}) = 0$, $\bar{T}$ does not separate twins, and $W < \lambda$ almost everywhere on $\bar{T} \times C$.

There are two subcases: First, it may be that $\mu(\bar{T}) = \mu(C)$. If so, then $W < \lambda$ almost everywhere on $C \times C$, which proves the claim. Second, it may be that $\mu(\bar{T}) < \mu(C)$. In this case, we have $W < \lambda$ almost everywhere on $\bar{T} \times (C \setminus \bar{T})$, and so taking $\bar{A} = \bar{T}$ proves the claim.

$\square$

The next two claims shown that the preimage under $\varphi$ of a cluster at level $\lambda$ in a graphon $W$ is connected at every level $\lambda' < \lambda$, and, conversely, a cluster at level $\lambda$ in $W^\varphi$ has a particular representative whose image under $\varphi$ is connected at every level $\lambda' < \lambda$ in $W$.

**Claim 21.** *Let $W$ be a graphon and $\varphi$ a measure preserving transformation. If $\mathscr{C}$ is a cluster at level $\lambda$ in $W$, then $\varphi^{-1}(\mathscr{C})$ is connected at every level $\lambda' < \lambda$ in $W^\varphi$.*

*Proof.* For simplicity, we will work with an representative $C$ of the cluster $\mathscr{C}$. As Claim 19 shows, we may take $C$ to be a representative such that $\varphi(\varphi^{-1}(C)) = C$.

Suppose for a contradiction that $\varphi^{-1}(C)$ is disconnected in $W^\varphi$ at some level $\lambda' < \lambda$. Then by Claim 20 either $W^\varphi < \lambda'$ almost everywhere on $\varphi^{-1}(C) \times \varphi^{-1}(C)$, or there exists a set $\bar{A} \subset \varphi^{-1}(C)$ such that $0 < \mu(\bar{A}) < \mu(\varphi^{-1}(C))$, $W^\varphi < \lambda'$ almost everywhere on $\bar{A} \times (\varphi^{-1}(C) \setminus \bar{A})$, and $\bar{A}$ does not separate twins.

In the first case, $W^\varphi < \lambda'$ almost everywhere on $\varphi^{-1}(C) \times \varphi^{-1}(C)$ implies that $W < \lambda'$ almost everywhere on $C \times C$, which contradicts the fact that $C$ is the representative of a cluster at level $\lambda'$ in $W$.

Suppose the second case, then, where $W^\varphi < \lambda'$ almost everywhere on $\bar{A} \times (\varphi^{-1}(C) \setminus \bar{A})$. Then $W < \lambda'$ almost everywhere on $\varphi(\bar{A}) \times \varphi(\varphi^{-1}(C) \setminus \bar{A})$. We now claim that $\varphi(\varphi^{-1}(C) \setminus \bar{A}) = \varphi(\varphi^{-1}(C)) \setminus \varphi(\bar{A}) = C \setminus \bar{A}$. To see this, note that $\varphi(\varphi^{-1}(C) \setminus \bar{A}) \supset \varphi^{-1}(\varphi(C)) \setminus \varphi(\bar{A})$. However, we have chosen $C$ to be a representative such that $\varphi(\varphi^{-1}(C)) = C$, and so we obtain

$$\varphi(\varphi^{-1}(C) \setminus \bar{A}) \supset C \setminus \bar{A}.$$

On the other hand, suppose $y \in \varphi(\varphi^{-1}(C) \setminus \bar{A})$. This means that there is some $x \in \varphi^{-1}(C) \setminus \bar{A}$ such that $\varphi(x) = y$. But $\varphi^{-1}(C) \setminus \bar{A}$ does not separate twins, so there cannot be an $x' \in \bar{A}$ such that $\varphi(x') = \varphi(x) = y$. Therefore, $y \in \varphi(\varphi^{-1}(C) \setminus \bar{A})$ if $y \in C$ and there is no $a \in \bar{A}$ such that $\varphi(a) = y$. That is, $\varphi(\varphi^{-1}(C) \setminus \bar{A}) \subset C \setminus \bar{A}$. Hence $\varphi(\varphi^{-1}(C) \setminus \bar{A}) = C \setminus \varphi(\bar{A})$.

Therefore $W < \lambda$ almost everywhere on $\varphi(\bar{A}) \times (C \setminus \varphi(\bar{A}))$. Since $\mu(\varphi(\bar{A})) = \mu(\bar{A}) < \mu(C)$ by Claim 17, this implies that $C$ is disconnected at level $\lambda'$ in $W$. Hence $C$ is not the representative of a cluster at level $\lambda$, and so we have derived a contradiction.

Both cases lead to contradictions, and so it must be that $\varphi^{-1}(C)$ is connected in $W^\varphi$ at every level $\lambda' < \lambda$.

$\square$

**Claim 22.** *Let $W$ be a graphon and $\varphi$ be a measure preserving transformation. Suppose $\mathscr{C}$ is a cluster of $W^\varphi$ at level $\lambda$. Let $C \in \mathsf{Fam}_W \mathscr{C}$. Then $\varphi(C)$ is connected at every level $\lambda' < \lambda$ in $W$.*

*Proof.* Suppose for a contradiction that $\varphi(C)$ is not connected at some level $\lambda' < \lambda$ in $W$. Then there exists a set $S \subset \varphi(C)$ such that $0 < \mu(S) < \mu(\varphi(C))$ and $W < \lambda'$ almost everywhere on $S \times (\varphi(C) \setminus S)$. Hence $W^\varphi < \lambda'$ almost everywhere on $\varphi^{-1}(S) \times \varphi^{-1}(\varphi(C) \setminus S) = \varphi^{-1}(S) \times (\varphi^{-1}(\varphi(C)) \setminus \varphi^{-1}(S))$. Since $C$ does not separate twins in $W^\varphi$, we have by Claim 17 that $\varphi^{-1}(\varphi(C)) = C$, and so $W^\varphi < \lambda'$ almost everywhere on $\varphi^{-1}(S) \times (C \setminus \varphi^{-1}(S))$.

Consider $\varphi^{-1}(S)$. We have $C = \varphi^{-1}(\varphi(C))$, and since $S \subset \varphi(C)$, it follows that $\varphi^{-1}(S) \subset C$. Moreover, $\mu(\varphi^{-1}(S)) = \mu(S)$, since $\varphi$ is measure preserving, and $0 < \mu(S) < \mu(\varphi(C)) = \mu(C)$, where the

last equality comes from Claim 17. Hence $C$ is disconnected at level $\lambda'$ in $W$. This contradicts the fact that $C$ is a representative of a cluster at level $\lambda$ in $W$. Hence it must be that $\varphi(C)$ is connected at every level $\lambda' < \lambda$ in $W$.

□

The two previous claims are sufficient to prove the main result of this section.

**Claim 1.** *Let $W$ be a graphon and $\varphi$ a measure preserving transformation. Then $\mathscr{C}$ is a cluster of $W^\varphi$ at level $\lambda$ if and only if there exists a cluster $\mathscr{C}'$ of $W$ at level $\lambda$ such that $\mathscr{C} = \varphi^{-1}(\mathscr{C}')$.*

*Proof.* Suppose $\mathscr{C}$ is a cluster of $W$ at level $\lambda$ and let $C$ be a representative of $\mathscr{C}$. Then according to Claim 21, $\varphi^{-1}(C)$ is connected at every level $\lambda' < \lambda$ in $W^\varphi$, and hence there exists a cluster $\mathscr{C}'$ at level $\lambda$ in $W^\varphi$ which contains $\varphi^{-1}(C)$. Then by Claim 22, there is a representative $C'$ of $\mathscr{C}'$ such that $C'$ does not separate twins and $\varphi(C')$ is connected at every level $\lambda' < \lambda$ in $W$, and so there is a cluster $\mathscr{C}''$ of $W$ at level $\lambda$ such that $\mathscr{C}''$ contains $\varphi(C')$. However, it must be that $\mathscr{C}'' = \mathscr{C}$. To see this, note that we have $\varphi^{-1}(C \cap \varphi(C')) = \varphi^{-1}(C) \cap \varphi^{-1}(\varphi(C')) = \varphi^{-1}(C) \cap C'$. Since $\varphi$ is measure preserving, it follows that $\mu(C \cap \varphi(C')) = \mu(\varphi^{-1}(C) \cap C')$, but $C' \triangle \varphi^{-1}(C)$ is a null set such that $\mu(C \cap \varphi(C')) = \mu(C)$. Thus $\mu(\mathscr{C}') = \mu(\varphi^{-1}(C))$, and so $\varphi^{-1}(C)$ is a representative of the cluster $\mathscr{C}'$. Hence $\varphi^{-1}(\mathscr{C})$ is a cluster at level $\lambda$ of $W^\varphi$.

Now suppose $\mathscr{C}$ is a cluster of $W^\varphi$ at level $\lambda$ and let $C$ be a representative of $\mathscr{C}$ such that $C \in \mathsf{Fam}_W(\mathscr{C})$. Then according to Claim 22, $\varphi(C)$ is connected at every level $\lambda' < \lambda$ in $W$, and hence there exists a cluster $\mathscr{C}'$ in $W$ at level $\lambda$ which contains $\varphi(C)$. By the previous argument, $\varphi^{-1}(\mathscr{C}')$ is a cluster of $W^\varphi$ at level $\lambda$. Since $C \in \mathsf{Fam}_W \mathscr{C}$, $C$ does not separate twins in $W^\varphi$, and so $\varphi^{-1}(\varphi(C)) = C$, and thus $C$ is contained in $\varphi^{-1}(\mathscr{C}')$. Since $C$ is a cluster representative, and thus maximal, it must be that $\varphi^{-1}(\mathscr{C}') = \mathscr{C}$.

□

# D   Sufficient conditions for consistent clustering methods

In this section we prove that any consistent estimator of the edge probability matrix leads to a consistent estimator of the graphon cluster tree. Estimating the graphon or the edge probability matrix is an area of recent research. There are a number of methods in the literature; See, for instance, [20], [8], [2], [18], [21]. Each work in this direction defines a slightly different sense in which the proposed estimator is consistent, but all use some variant of the mean squared error. Convergence in this norm ensures that the estimate is close to the true graphon in aggregate, but still allows the estimate to differ from the ground truth by a large amount on a set of small measure. Since our merge distortion is sensitive to the *largest* error, regardless of measure, consistency of graphon estimators as shown in the literature is not sufficient to show consistency in merge distortion.

In particular we require that the estimator $\hat{\mathbf{P}}$ satisfies

$$\mathbb{P}\left(\max_{i \neq j} \left|\hat{\mathbf{P}}_{ij} - \mathbf{P}_{ij}\right| > \epsilon\right) \to 0$$

for every $\epsilon > 0$ as $n \to \infty$. The probability is with respect to the label measure introduced in Definition 6. That is, to be precise:

$$\mathbb{P}\left(\max_{i \neq j} \left|\hat{\mathbf{P}}_{ij} - \mathbf{P}_{ij}\right| > \epsilon\right) = \Lambda_{W,n}\left(\left\{(G, S) : \max_{i \neq j} |P_{ij} - \hat{P}_{ij}| > \epsilon\right\}\right)$$

It is implicit here that $P$ is induced by the graphon $W$ and the particular labeling $S$, and $\hat{P}$ is a function of the graph, $G$.

Given an estimator $\hat{\mathbf{P}}$, we construct a consistent clustering algorithm as follows. Let $\mathcal{P}_n(i, j)$ be the set of all simple paths between nodes $i$ and $j$ in the complete graph on node set $[n]$. For $p \in \mathcal{P}_n(i, j)$, let $\ell(p)$ denote the length of the path, and let $p_k$ be the label of the $k$th node along the path. For any $i \neq j \in [n] \times [n]$, define the *merge estimate* $\hat{\mathbf{Q}}_{ij}$ by

$$\hat{\mathbf{Q}}_{ij} = \max_{p \in \mathcal{P}_n(i,j)} \min_{1 \leq k \leq \ell(p)} \hat{\mathbf{P}}_{p_k p_{k+1}}.$$

As its name implies, the merge estimate $\hat{\mathbf{Q}}_{ij}$ estimates the height at which nodes $i$ and $j$ merge in the cluster tree. Intuitively, if $\hat{\mathbf{Q}}_{ij}$ is close to the true merge height for every pair $i, j$, we can use $\hat{\mathbf{Q}}$ to construct a clustering which is close to the cluster tree in merge distortion. Specifically, let $\mathbf{H}$ be the weighted graph on node set $[n]$ in which the weight between nodes $i$ and $j$ is given by $\hat{\mathbf{Q}}_{ij}$. We define the clusters of $\mathbf{H}$ at level $\lambda$ to be the connected components of the subgraph induced by removing every edge with weight less than $\lambda$. The clustering $\mathsf{C}_{\hat{\mathbf{Q}}}$ is defined to be the set of all clusters of $\mathbf{H}$ at any level $\lambda$. Equivalently, $\hat{\mathbf{Q}}_{ij}$ is the level at which nodes $i$ and $j$ merge in the single linkage clustering of $\hat{\mathbf{P}}$, when $\hat{\mathbf{P}}$ is treated as a similarity matrix. Thus $\mathsf{C}_{\hat{\mathbf{Q}}}$ is simply the single linkage clustering of $\hat{\mathbf{P}}$.

## D.1 Claims

We state our claims here, and place all technical details and proofs in Appendix D.2 for clarity.

It is sufficient to show that if $|\hat{\mathbf{P}}_{ij} - \mathbf{P}_{ij}| < \epsilon$, $\hat{\mathbf{Q}}_{ij}$ is at most $\epsilon + c$ away from the true merge height $M(x_i, x_j)$, where $c$ is a constant. It is easy to see that the merge distortion between the cluster tree and $\mathsf{C}_{\hat{\mathbf{Q}}}$ cannot be greater than $2(\epsilon + c)$.

**Claim 23.** *Let $W$ be a graphon, $M$ be a mergeon of $W$, and $S = (x_1, \ldots, x_n)$. Suppose $\max_{i \neq j} |M(x_i, x_j) - \hat{Q}_{ij}| < \epsilon$, and let $\mathsf{C}_{\hat{Q}}$ be the clustering defined above of the weighted graph $H$ with weight matrix $\hat{Q}$. Let $\hat{M}$ be the merge height on $\mathsf{C}_{\hat{Q}}$ induced by $M$. Then the merge distortion $d_S(M, \hat{M}) < 2\epsilon$.*

We will now show that $\hat{\mathbf{Q}}_{ij}$ is close to the true merge height for all $ij$ with high probability. First, recall the definition of a piecewise Lipschitz graphon:

**Definition 7** (Piecewise Lipschitz). *We say that $\mathcal{B} = \{B_1, \ldots, B_k\}$ is a* block partition *if each $B_i$ is an open, half-open, or closed interval in $[0, 1]$ with positive measure, $B_i \cap B_j$ is empty whenever $i \neq j$, and $\bigcup \mathcal{B} = [0, 1]$. We say that a graphon $W$ is* piecewise $\mathsf{c}$-Lipschitz *if there exists a set of blocks $\mathcal{B}$ such that for any $(x, y)$ and $(x', y')$ in $B_i \times B_j$, $|W(x, y) - W(x', y')| \leq \mathsf{c}(|x - x'| + |y - y'|)$.*

The idea is that the piecewise Lipschitz graphon is essentially piecewise constant when viewed at small enough scales. As such, we refine the blocks on which the graphon is Lipschitz, creating a new block partition whose blocks are small enough that $W$ varies by only a small amount on each. We define a refinement as follows:

**Definition 13.** *A set of blocks $\mathcal{R} = \{R_i\}$ is a $\Delta$-refinement of a block partition $\mathcal{B} = \{B_i\}$ if for every $R \in \mathcal{R}$, $\Delta \leq \mu(R) \leq 2\Delta$ and there exists some $B \in \mathcal{B}$ such that $B \supseteq R$.*

We can think of the blocks in a refinement as being nodes in a weighted graph, such that the weight between blocks $R$ and $R'$ is approximately the value of $W$ on $R \times R'$. As such, we define a path of blocks in a refinement as follows:

**Definition 14.** *Let $\mathcal{R}$ be a block partition of $[0, 1]$, and suppose $R, R' \in \mathcal{R}$. A $\lambda$-path from $R$ to $R'$ in a graphon $W$ is a sequence $\langle R = R_1, \ldots, R_t = R' \rangle$ of blocks from $\mathcal{R}$ such that, for all $1 \leq i < t$, $W \geq \lambda$ almost everywhere on $R_i \times R_{i+1}$. The elements of the path need not be distinct.*

In piecewise Lipschitz graphons, the existence of a $\lambda$-path between blocks $R$ and $R'$ implies that there exists a set $C$ containing both $R$ and $R'$, and which is connected at level $\lambda$, as the following claim demonstrates. Note that is directly analogous to the case of a finite weighted graph, where a pair of nodes is connected if there is a path between them.

**Claim 24.** *Let $W \in \mathcal{W}_{\mathcal{B}}^{\mathsf{c}}$ and let $M$ be a mergeon of $W$. Let $\mathcal{R}$ be a $\Delta$-refinement of $\mathcal{B}$. Let $\langle R_1, \ldots, R_t \rangle$ be a $\lambda$-path in $\mathcal{R}$. Let $C = R_1 \cup \ldots \cup R_t$. Then $C$ is connected at level $\lambda$ in $W$, and thus $M \geq \lambda$ almost everywhere on $C \times C = (R_1 \cup \ldots \cup R_t) \times (R_1 \cup \ldots \cup R_t)$.*

Conversely, if $R$ and $R'$ are blocks in a $\Delta$-refinement, each of which have non-null intersection with the same cluster $\mathscr{C}$, then there exists a $(\lambda - 2\Delta)$-path of blocks between $R$ and $R'$:

**Claim 25.** *Let $W \in \mathcal{W}_{\mathcal{B}}^{\mathsf{c}}$. Let $\mathcal{R}$ be a $\Delta$-refinement of $\mathcal{B}$, and suppose $R, R' \in \mathcal{R}$ (possibly with $R = R'$). If there exists a cluster $\mathscr{C}$ at level $\lambda$ such that $\mu(\mathscr{C} \cap R) > 0$ and $\mu(\mathscr{C} \cap R') > 0$, then there exists a $(\lambda' - 2\Delta\mathsf{c})$-path $(R = R_1, \ldots, R_t = R')$ between $R$ and $R'$, for any $\lambda' < \lambda$.*

Lastly, the Lipschitz condition on the graphon $W$ also implies that the mergeon does not vary much:

**Claim 26.** *Let $R, R' \in \mathcal{R}$. Let $\lambda$ be the greatest level at which there exists some cluster $\mathscr{C}$ containing a non-negligible piece of both $R$ and $R'$. That is,*

$$\lambda = \sup\{\lambda' : \exists \mathscr{C} \in \mathbb{C}(\lambda') \text{ such that } \mu(R \cap \mathscr{C}) > 0 \text{ and } \mu(R' \cap \mathscr{C}) > 0.\}$$

*Then $\lambda' - 2\Delta\mathsf{c} \le M \le \lambda$ almost everywhere on $R \times R'$.*

Putting these ideas together, we are able to bound the difference between the true merge height of points in a mergeon, and the merge estimate $\hat{Q}$.

**Claim 27.** *Let $W \in \mathscr{W}_{\mathcal{B}}^{\mathsf{c}}$ and let $M$ be a mergeon of $W$. Let $\mathcal{R}$ be a $\Delta$-refinement of $\mathcal{B}$. Let $S = (x_1, \ldots, x_n)$ be an ordered set of elements of $[0, 1]$ such for any $R \in \mathcal{R}$, $R \cap S \ne \emptyset$. Let $P$ be the edge probability matrix, i.e., the matrix whose $(i, j)$ entry is given by $W(x_i, x_j)$, and suppose $\hat{P}$ is such that $\|\hat{P} - P\|_\infty < \epsilon$. Then $\max_{i \ne j} |M(x_i, x_j) - \hat{Q}_{ij}| \le 4\Delta\mathsf{c} + \epsilon$.*

The above holds for a fixed sample $S$ and thus a fixed edge probability matrix $P$. The following theorem considers random $\mathbf{S}$ and $\mathbf{P}$. As with the previous claims in this subsection, the proof of the theorem is in Appendix D.2.

**Theorem 1.** *Let $W$ be a piecewise $\mathsf{c}$-Lipschitz graphon. Let $\hat{\mathbf{P}}$ be a consistent estimator of $\mathbf{P}$ in max-norm. Let $f$ be the clustering method which performs single-linkage clustering using $\hat{\mathbf{P}}$ as a similarity matrix. Then $f$ is a consistent estimator of the graphon cluster tree of $W$.*

## D.2 Proofs

**Claim 23.** *Let $W$ be a graphon, $M$ be a mergeon of $W$, and $S = (x_1, \ldots, x_n)$. Suppose $\max_{i \ne j} |M(x_i, x_j) - \hat{Q}_{ij}| < \epsilon$, and let $\mathsf{C}_{\hat{Q}}$ be the clustering defined above of the weighted graph $H$ with weight matrix $\hat{Q}$. Let $\hat{M}$ be the merge height on $\mathsf{C}_{\hat{Q}}$ induced by $M$. Then the merge distortion $d_S(M, \hat{M}) < 2\epsilon$.*

*Proof.* Take any arbitrary $i \ne j$ in the clustering $\mathsf{C}_{\hat{Q}}$. Let $C$ be the smallest cluster containing both $i$ and $j$. Then $C$ is a cluster in $H$ at level $\hat{Q}_{ij}$. Let $u, v \in C$, $u \ne v$ be such that $M(x_u, x_v) = \min_{u' \ne v' \in C} M(x_{u'}, x_{v'}) = \hat{M}_{ij}$. Then we have that $M(x_i, x_j) \ge M(x_u, x_v)$. On the other hand, $u$ and $v$ are members of $C$, which is a cluster at level $\hat{Q}_{ij}$, so that $\hat{Q}_{uv}\hat{Q}ij$. Hence $\hat{Q}_{uv} > M(x_i, x_j) - \epsilon$. But $\hat{Q}_{uv} < M(x_u, x_v) + \epsilon$. Therefore, $M(x_i, x_j) - M(x_u, x_v) < 2\epsilon$. Therefore, $M(x_i, x_j) - \hat{M}_{ij} < 2\epsilon$. This holds for all $i$ and $j$ simultaneously, since $i$ and $j$ were arbitrary. Hence the merge distortion is less than $2\epsilon$. $\square$

**Claim 24.** *Let $W \in \mathscr{W}_{\mathcal{B}}^{\mathsf{c}}$ and let $M$ be a mergeon of $W$. Let $\mathcal{R}$ be a $\Delta$-refinement of $\mathcal{B}$. Let $\langle R_1, \ldots, R_t \rangle$ be a $\lambda$-path in $\mathcal{R}$. Let $C = R_1 \cup \ldots \cup R_t$. Then $C$ is connected at level $\lambda$ in $W$, and thus $M \ge \lambda$ almost everywhere on $C \times C = (R_1 \cup \ldots \cup R_t) \times (R_1 \cup \ldots \cup R_t)$.*

*Proof.* Let $A$ be an arbitrary measurable subset of $C$ such that $0 < \mu(A) < \mu(C)$. We will show that $W^{-1}[\lambda, 1] \cap A \times (C \setminus A)$ has positive measure, and therefore $C$ is connected at level $\lambda$. Since $C$ is connected at level $\lambda$ in $W$, it must be part of some cluster at level $\lambda$, and so the mergeon is at least $\lambda$ almost everywhere on $C \times C$.

There are two cases: Either 1) There exists a $j \in [t]$ such that $0 < \mu(R_j \cap A) < \mu(R_j)$, or 2) for all $i \in [t]$, either $\mu(R_i \cap A) = 0$ or $\mu(R_i \cap A) = \mu(R_i)$.

Assume the first case: there exists a $j$ such that $R_j$ contains some non-negligible part of $A$, but $\mu(A \cap R_j) < \mu(R_j)$. Since there are at least two elements in the path, there is a $j'$ such that $j' \in [t]$ and $|j - j'| = 1$, that is, $R_{j'}$ is either immediately before or after $R_j$ in the $\lambda$-path. There are two sub-cases:

- $\mu(R_{j'} \cap A) = 0$, such that $R_{j'} \subseteq C \setminus A$. Then $(R_j \cap A) \times R_{j'} \subseteq A \times (C \setminus A)$. Since $\mu(R_j \cap A) > 0$ and $\mu(R_{j'}) > 0$, we have that $\mu((R_j \cap A) \times R_{j'}) > 0$, and since $W$ is at least $\lambda$ a.e. on $R_j \times R_{j'}$, we have that

$$\mu(W^{-1}[\lambda, 1] \cap A \times (C \setminus A)) \ge \mu(W^{-1}[\lambda, 1] \cap (R_j \cap A) \times R_{j'}) > 0.$$

- $\mu(R_{j'} \cap A) > 0$. Then $(R_{j'} \cap A) \times (R_j \setminus A) \subseteq A \times (C \setminus A)$ is a set of positive measure. Since $W$ is at least $\lambda$ a.e. on $R_{j'} \times R_j$, we have:

$$\mu(W^{-1}[\lambda, 1] \cap A \times (C \setminus A)) \geq \mu\left(W^{-1}[\lambda, 1] \cap (R_{j'} \cap A) \times (R_j \setminus A)\right) > 0.$$

Now consider the second case in which, for every $i \in [t]$, $\mu(R_i \cap A = 0)$ or $\mu(R_i \cap A) = \mu(R_i)$. There must exist a $j, j' \in [t]$ such that $|j - j'| = 1$, $\mu(R_j \cap A) = \mu(R_j)$, and $\mu(R_{j'} \cap A) = 0$. If this were not the case, then it would be that either $\mu(R_i \cap A) = \mu(R_i)$ for every $i \in [t]$, or $\mu(R_i \cap A) = 0$ for every $i \in [t]$. But the former of these would imply that $\mu(A) = \mu(C)$, and the latter would imply $\mu(A) = 0$, which we have assumed not to be the case.

Therefore, $R_j \times R_{j'} \subseteq A \times (C \setminus A)$, and this set is of positive measure. Since $W$ is at least $\lambda$ a.e. on $R_j \times R_{j'}$, we once again find

$$\mu(W^{-1}[\lambda, 1] \cap A \times (C \setminus A)) \geq \mu\left(W^{-1}[\lambda, 1] \cap R_j \times R_{j'}\right) > 0.$$

Hence, in every case it is true that $\mu(W^{-1}[\lambda, 1] \cap A \times (C \setminus A))$ has positive measure. Since $A$ was arbitrary, $C$ is connected at level $\lambda$. Hence $M \geq \lambda$ almost everywhere on $C \times C$. $\qquad\square$

**Claim 25.** *Let $W \in \mathscr{W}_{\mathcal{B}}^{\mathsf{c}}$. Let $\mathcal{R}$ be a $\Delta$-refinement of $\mathcal{B}$, and suppose $R, R' \in \mathcal{R}$ (possibly with $R = R'$). If there exists a cluster $\mathscr{C}$ at level $\lambda$ such that $\mu(\mathscr{C} \cap R) > 0$ and $\mu(\mathscr{C} \cap R') > 0$, then there exists a $(\lambda' - 2\Delta\mathsf{c})$-path $(R = R_1, \dots, R_t = R')$ between $R$ and $R'$, for any $\lambda' < \lambda$.*

*Proof.* To be precise, let $C = \rho(\mathscr{C})$ be any representative of the cluster $\mathscr{C}$. Fix a $\lambda' < \lambda$. Let

$$\mathcal{G} = \{R'' \in \mathcal{R} : \mu(R'' \cap C) > 0\}.$$

Then $\mathcal{G}$ contains, in particular, $R_1$ and $R_t$. Since $\mathscr{C}$ is connected at level $\lambda$, it is true that

$$\mu(W^{-1}[\lambda', 1] \cap (R_1 \cap C) \times (C \setminus R_1)) > 0.$$

Since $C \setminus R_1$ is a subset of $(\bigcup \mathcal{G}) \setminus R_1$, there must exist an $R_2 \in \mathcal{G}$ such that

$$\mu(W^{-1}[\lambda', 1] \cap R_1 \times R_2) > 0.$$

Consider $W$ on $R_2$. From above, we know that there is a non-negligible subset of $R_1 \times R_2$ on which $W \geq \lambda'$. Hence there is some point in $R_1 \times R_2$ on which $W \geq \lambda'$. Therefore, due to the Lipschitz condition, we know that $W$ is at least $\lambda' - 2\Delta\mathsf{c}$ everywhere on $R_1 \times R_2$.

Now let $S_2 = R_1 \cup (R_2 \cap C)$. Now, since $\mathscr{C}$ is connected at level $\lambda$, it is true that

$$\mu(W^{-1}[\lambda', 1] \cap S_2 \times (C \setminus S_2)) > 0.$$

By the same logic as above, there must exist an $R_3 \in \mathcal{G}$, $R_3 \neq R_2, R_1$ such that

$$\mu(W^{-1}[\lambda', 1] \cap S_2 \times R_3) > 0.$$

Hence it must be the case that either

$$\mu(W^{-1}[\lambda, 1] \cap R_1 \times R_3) > 0,$$

or

$$\mu(W^{-1}[\lambda, 1] \cap R_2 \times R_3) > 0.$$

In either case, it is true that between any pair chosen from $R_1, R_2, R_3$, there is a $\lambda - 2\Delta\mathsf{c}$ path. The process continues, choosing $R_4, R_5, \dots$ and so on. This process must complete in a finite number of steps, since $\mathcal{G}$ is a finite set. At every step, there exists a $\lambda$-path between any two of the $R_i$. Hence we eventually construct a $\lambda$-path between $R$ and $R'$.

$\qquad\square$

**Claim 26.** *Let $R, R' \in \mathcal{R}$. Let $\lambda$ be the greatest level at which there exists some cluster $\mathscr{C}$ containing a non-negligible piece of both $R$ and $R'$. That is,*

$$\lambda = \sup\{\lambda' : \exists \mathscr{C} \in \mathbb{C}(\lambda') \text{ such that } \mu(R \cap \mathscr{C}) > 0 \text{ and } \mu(R' \cap \mathscr{C}) > 0.\}$$

*Then $\lambda' - 2\Delta\mathsf{c} \leq M \leq \lambda$ almost everywhere on $R \times R'$.*

*Proof.* By the definition of the mergeon it must be that $M \leq \lambda$ almost everywhere on $R \times R'$, since if there existed a $\lambda' > \lambda$ for which $M^{-1}[\lambda', 1] \cap R \times R'$ is not-null, this would imply that there exists some cluster at level $\lambda'$ containing a non-negligible part of both $R$ and $R'$.

Now, by Claim 25, for any $\lambda' < \lambda$ there exists a $(\lambda' - 2\Delta c)$ path between $R$ and $R'$. Hence, by Claim 24, $M \geq \lambda' - 2\Delta c$ almost everywhere on $R \times R'$ for any $\lambda' < \lambda$. $\qquad\square$

**Claim 27.** *Let $W \in \mathscr{W}_{\mathcal{B}}^c$ and let $M$ be a mergeon of $W$. Let $\mathcal{R}$ be a $\Delta$-refinement of $\mathcal{B}$. Let $S = (x_1, \ldots, x_n)$ be an ordered set of elements of $[0, 1]$ such for any $R \in \mathcal{R}$, $R \cap S \neq \emptyset$. Let $P$ be the edge probability matrix, i.e., the matrix whose $(i, j)$ entry is given by $W(x_i, x_j)$, and suppose $\hat{P}$ is such that $\|\hat{P} - P\|_\infty < \epsilon$. Then $\max_{i \neq j} |M(x_i, x_j) - \hat{Q}_{ij}| \leq 4\Delta c + \epsilon$.*

*Proof.* Consider an arbitrary $x_i, x_j \in S$. Let $R_i$ and $R_j$ be the blocks in $\mathcal{R}$ which contain $x_i$ and $x_j$, respectively. Let $\lambda^*$ be the greatest level at which there exists some cluster containing non-negligible parts of both $R_i$ and $R_j$. Therefore, by Claim 26, $M$ is bounded below by $\lambda^* - 2\Delta c$ and above by $\lambda^*$ almost everywhere on $R_i \times R_j$.

First we bound $\hat{Q}_{ij}$ from below. By Claim 25 there exists a $(\lambda' - \Delta c)$-path $\langle R_i = R_1, \ldots, R_t = R_j \rangle$ between $R_i$ and $R_j$, for any $\lambda' < \lambda^*$. By the assumption on $S$, there exists a sample from each element of the path, so that there is a path of samples $\langle x_i = x_1, \ldots, x_t = x_j \rangle$ with the property that, between any two consecutive elements in the path, we have $W(x_k, x_{k+1}) \geq \lambda' - 2\Delta c$ for all $\lambda' < \lambda^*$. Hence $\hat{P}(x_k, x_{k+1}) \geq \lambda^* - 2\Delta c - \epsilon$. Therefore, there exists a path $p$ from $x_i$ to $x_j$ such that $\min_{1 \leq k \leq \ell(p)} \hat{P}_{p_k p_{k+1}} \geq \lambda^* - 2\Delta c$. As a result, $\hat{Q}_{ij} \geq \lambda^* - 2\Delta c - \epsilon$.

We now bound $\hat{Q}_{ij}$ from above. Let $p = \langle x_i = x_1, \ldots, x_t = x_j \rangle$ be a path with cost $\hat{Q}_{ij}$. Let $\langle R_1, \ldots, R_t \rangle$ be the corresponding path of blocks from $\mathcal{R}$, such that $x_k \in R_k$. Then we have $\hat{P}_{x_k x_{k+1}} \geq \hat{Q}_{ij}$, so that $W(x_k, x_{k+1}) \geq \hat{Q}_{ij} - \epsilon$. Hence there is a point in $R_k \times R_{k+1}$ which is at least $\hat{Q}_{ij} - \epsilon$, and by smoothness it follows that $W \geq \hat{Q}_{ij} - 2\Delta c - \epsilon$ almost everywhere on $R_k \times R_{k+1}$. That is, $\langle R_1, \ldots, R_t \rangle$ is a $(\hat{Q}_{ij} - 2\Delta c - \epsilon)$-path. Therefore, Claim 24 implies that the mergeon $M$ is at least $\hat{Q}_{ij} - 2\Delta c - \epsilon$ almost everywhere on $R_i \times R_j$. However, by Claim 26, $M \leq \lambda^*$ almost everywhere on $R_i \times R_j$. Therefore $\hat{Q}_{ij} \leq \lambda^* + 2\Delta c + \epsilon$.

Combining the above bounds, we find that

$$|\hat{Q}_{ij} - \lambda^*| \leq 2\Delta c + \epsilon.$$

The true merge height $M(x_i, x_j)$ is bounded between $\lambda^* - 2\Delta c$ and $\lambda^*$, and so we have

$$|\hat{Q}_{ij} - M(x_i, x_j)| \leq 4\Delta c + \epsilon.$$

$\qquad\square$

**Theorem 1.** *Let $W$ be a piecewise c-Lipschitz graphon. Let $\hat{P}$ be a consistent estimator of $P$ in max-norm. Let $f$ be the clustering method which performs single-linkage clustering using $\hat{P}$ as a similarity matrix. Then $f$ is a consistent estimator of the graphon cluster tree of $W$.*

*Proof.* As stated, $f$ is the clustering method which takes a graph $G$ and returns the clustering $C_{\hat{Q}}$ described at the beginning of the section – the single linkage clustering of the estimated edge probability matrix $\hat{P}$. Let $M$ be a mergeon of $W$. We will show that, for any $\epsilon > 0$.

$$\Lambda_{W,n}\left(\left\{(G, S) : d_S(M, \hat{M}_{f(G) \circ S}) > \epsilon\right\}\right) \to 0,$$

where $\hat{M}_{f(G) \circ S}$ is the merge height function induced on the clustering $f(G) \circ S$ by the mergeon $M$.

First, fix any $\epsilon > 0$. Let $\tilde{\epsilon} = \epsilon/4$. Define

$$H_n = \left\{(G, S) \in \mathfrak{G}_n \times [0, 1]^n : \max_{i \neq j} |\hat{P} - P| < \tilde{\epsilon}\right\},$$

where $P$ is the edge probability matrix induced by $S$ and $\hat{P}$ is the estimate of $P$ computed from $G$. By the assumption that $\hat{P}$ is consistent in $\infty$-norm, we have $\Lambda_{W,n}(H_n) \to 1$ as $n \to \infty$.

Now let $\Delta = \epsilon/16\mathsf{c}$. Let $\mathcal{B}$ be the block partition on which $W$ is piecewise $\mathsf{c}$-Lipschitz, and let $\mathcal{R}$ be an arbitrary $\Delta$-refinement of $\mathcal{B}$. In order to apply Claim 27, we require that the labeling $S$ satisfies the property that every block $R$ in the refinement contains at least one point from $S$. The probability that a block $R$ contains no points from a random sample $\mathbf{S}$ is $(1 - |R|)^n \le (1 - \Delta/2)^n$, since $|R| \ge \Delta/2$. Now take a union bound over all blocks in the partition, of which there are at most $2/\Delta$. Hence the probability that there exists a block in the partition that does not have a sample from $S$ is $\frac{2}{\Delta}(1 - \Delta/2)^n$. Let

$$F_n = \{(G, S) \in \mathfrak{G}_n \times [0, 1]^n : |R \cap S| > 1 \text{ for all } R \in \mathcal{R}\}.$$

As per above, we have $\Lambda_{W,n}(F_n) = \frac{2}{\Delta}(1 - \Delta/2)^n$, which tends to 0 as $n \to \infty$.

By Claim 27, for every $(G, S) \in H_n \setminus F_n$, we have that, for all $i \ne j \in [n] \times [n]$, writing $S = (x_1, \ldots, x_n)$:

$$|\hat{Q}_{ij} - M(x_i, x_j)| \le 4\Delta\mathsf{c} + \tilde{\epsilon} = \epsilon/2,$$

where $\hat{Q}$ is the merge estimate between nodes $i$ and $j$, described at the beginning of the section. The clustering method $f$ uses $\hat{Q}$ to construct the clustering $\mathsf{C}_{\hat{Q}}$ Therefore, by Claim 23, the merge distortion $d(M, \hat{M}_{f(G) \circ S})$ is bounded above by $\epsilon$ on the set $H_n \setminus F_n$. Since $\Lambda_{W,n}(H_n) \to 1$ and $\Lambda_{W,n}(F_n) \to 0$ as $n \to \infty$, we have $\Lambda_{W,n}(H_n \setminus F_n) \to 1$ as $n \to \infty$ and have thus proven the claim. $\square$

# E  Neighborhood smoothing methods

Theorem 1 states a sufficient condition under which an estimator $\hat{\mathbf{P}}$ of the edge probability matrix leads to a consistent clustering algorithm. In particular, if the graphon $W$ is piecewise Lipschitz, and if for any $\epsilon > 0$,

$$\lim_{n \to \infty} \mathbb{P}(\max_{i \ne j} |\mathbf{P}_{ij} - \hat{\mathbf{P}}_{ij}| > \epsilon) = 0$$

then one consistent clustering algorithm is that which applies single linkage clustering to the estimate $\hat{\mathbf{P}}$. In this section, we analyze a modification of the edge probability estimator introduced in[21] and show that it satisfies the above condition. Combining this result with Theorem 1 shows that the single linkage clustering applied this estimate of the edge probability matrix is a consistent clustering algorithm.

## E.1  The method of Zhang et al. [21]

The aim of the neighborhood smoothing method of Zhang et al. [21] is to estimate the random edge probability matrix $\mathbf{P}$. In particular, the method defines a distance $d(i, i')$ between the columns of the random adjacency matrix $\mathbf{A}$ as such:

$$d(i, i') = \frac{1}{n} \max_{k \ne i, i'} |\langle \mathbf{A}_i - \mathbf{A}_{i'}, \mathbf{A}_k \rangle| = \max_{k \ne i, i'} |(\mathbf{A}^2/n)_{ik} - (\mathbf{A}^2/n)_{i'k}|.$$

The neighborhood $\mathcal{N}_i(\mathbf{A})$ of node $i$ then consists of all nodes $i'$ such that $d(i, i')$ is below the $h$-th quantile of $\{d(i, k)\}_{k \ne i}$, where $h$ is a parameter of the algorithm. Note that $\mathcal{N}_i(\mathbf{A})$ is a random variable, as the neighborhood around node $i$ depends on the random adjacency matrix $\mathbf{A}$. For simplicity, however, we will often omit the explicit dependence on $\mathbf{A}$.

The estimate of the probability of the edge $(i, j)$, written $\hat{\mathbf{P}}_{ij}$, is then computed by smoothing over the neighborhoods $\mathcal{N}_i$ and $\mathcal{N}_j$:

$$\hat{\mathbf{P}}_{ij} = \frac{1}{2} \left( \frac{1}{|\mathcal{N}_i|} \sum_{i' \in \mathcal{N}_i} \mathbf{A}_{i'j} + \frac{1}{|\mathcal{N}_j|} \sum_{j' \in \mathcal{N}_j} \mathbf{A}_{ij'} \right).$$

If it is assumed that $W \in \mathscr{W}_{\mathcal{B}}^{\mathsf{c}}$, and $h$ is set to be $C_0 \sqrt{\log n/n}$ for arbitrary constant $C_0$, where $n$ is the size of the sampled graph, then the method is consistent in the sense that, for any $\epsilon > 0$, as $n \to \infty$

$$\mathbb{P}\left( \frac{1}{n^2} \|\hat{\mathbf{P}} - \mathbf{P}\|_F^2 > \epsilon \right) \to 0$$

## E.2 Our modification

In order to construct an algorithm which is a consistent estimator of the graphon cluster tree in the sense made precise above, we need for the edge probability estimator to be consistent in a stronger sense. In particular, we need that for any $\epsilon > 0$, as $n \to \infty$

$$\mathbb{P}\left(\max_{i \neq j} |\hat{\mathbf{P}}_{ij} - \mathbf{P}_{ij}| > \epsilon\right) \to 0.$$

In order to show that the neighborhood smoothing method satisfies such a notion of consistency, one might attempt to apply a concentration inequality to bound the difference between $\frac{1}{|\mathcal{N}_i|}\sum_{i' \in \mathcal{N}_i} \mathbf{A}_{i'j}$ and $\mathbf{P}_{ij}$. The problem with this approach, however, is that such concentration results require an assumption of statistical independence that is not satisfied by the neighborhoods as defined; that is, the terms of the sum $\sum_{i' \in \mathcal{N}_i} \mathbf{A}_{i'j}$ are not statistically independent. It is true that, unconditioned, $\mathbf{A}_{i'j}$ and $\mathbf{A}_{i''j}$ are independent Bernoulli random variables. However, once we condition on the event $i' \in \mathcal{N}_i$ and $i'' \in \mathcal{N}_i$, the random variables $\mathbf{A}_{i'j}$ and $\mathbf{A}_{i''j}$ are no longer independent.

More precisely, we are interested in

$$\mathbb{P}(\mathbf{A}_{i'j}, \mathbf{A}_{i''j} \mid i', i'' \in \mathcal{N}_i) = \frac{\mathbb{P}(i', i'' \in \mathcal{N}_i \mid \mathbf{A}_{i'j}, \mathbf{A}_{i''j})\mathbb{P}(\mathbf{A}_{i'j}, \mathbf{A}_{i''j})}{\mathbb{P}(i', i'' \in \mathcal{N}_i)}. \tag{2}$$

The denominator of the RHS is a normalization constant which does not depend on $\mathbf{A}_{i'j}$ or $\mathbf{A}_{i''j}$. Moreover, the entries of $\mathbf{A}$ are independent when unconditioned, and so $\mathbb{P}(\mathbf{A}_{i'j}, \mathbf{A}_{i''j}) = \mathbb{P}(\mathbf{A}_{i'j})\mathbb{P}(\mathbf{A}_{i''j})$. The difficulty is in

$$\mathbb{P}(i', i'' \in \mathcal{N}_i \mid \mathbf{A}_{i'j}, \mathbf{A}_{i''j}).$$

Intuitively, the event $i' \in \mathcal{N}_i$ depends on $\mathbf{A}_{i'j}$, and, likewise, $i'' \in \mathcal{N}_i$ depends on $\mathbf{A}_{i''j}$. The reason is that $i' \in \mathcal{N}_i$ when $d(i, i')$ is small. But $d(i, i')$ depends on $\mathbf{A}_{i'j}$, since

$$d(i, i') = \max_{k \neq i, i'} |(\mathbf{A}^2/n)_{ik} - (\mathbf{A}^2/n)_{i'k}|,$$

$$= \frac{1}{n} \max_{k \neq i, i'} \left|\sum_{\ell=1}^{n} \mathbf{A}_{k\ell}\left(\mathbf{A}_{i\ell} - \mathbf{A}_{i'\ell}\right)\right|.$$

and so $\mathbf{A}_{i'j}$ enters the sum and $d(i, i')$ depends on it. In the extreme case, suppose there are two nodes $i'$ and $i''$ such that the row vectors $\mathbf{A}_{i'}$ and $\mathbf{A}_{i''}$ are identical except in their $j$th component. Then the only difference between $d(i, i')$ and $d(i, i'')$ comes from the difference in $\mathbf{A}_{i'j}$ and $\mathbf{A}_{i''j}$. Hence it is clear that $d(i, i')$ and $d(i, i'')$ depend on the values of $\mathbf{A}_{i'j}$ and $\mathbf{A}_{i''j}$, and, by extension, the events $i' \in \mathcal{N}_i$ and $i'' \in \mathcal{N}_i$ are *not* independent of $\mathbf{A}_{i'j}$ and $\mathbf{A}_{i''j}$.

Our modification of the algorithm is to change the way in which neighborhoods are constructed so that statistical independence is preserved. Instead of constructing a neighborhood for each node $i$, we construct a neighborhood $\mathcal{N}_{i\backslash j}$ for each *ordered pair* $(i, j)$ by using a parameterized distance function $d_j$ which ignores all information about node $j$. More precisely, let $\partial_j \mathbf{A}$ represent the matrix obtained by setting the $j$th row and column of $\mathbf{A}$ to zero. Then for every node $j$ we define

$$d_j(i, i') = \max_{k \neq i, i', j} |([\partial_j \mathbf{A}]^2/n)_{ik} - ([\partial_j \mathbf{A}]^2/n)_{i'k}|,$$

$$= \frac{1}{n} \max_{k \neq i, i', j} \left|\sum_{\substack{\ell=1 \\ \ell \neq j}}^{n} \mathbf{A}_{k\ell}\left(\mathbf{A}_{i\ell} - \mathbf{A}_{i'\ell}\right)\right|.$$

The important thing to note here is that $\mathbf{A}_{i'j}$ does not appear in $d_j(i, i')$, and, since the other entries of $\mathbf{A}$ are independent of $\mathbf{A}_{i'j}$, we have that $d_j(i, i')$ is statistically independent of $\mathbf{A}_{i'j}$. Therefore the event $i' \in \mathcal{N}_{i\backslash j}$ is independent of $\mathbf{A}_{ij}$.

We are now interested in the quantity:

$$\mathbb{P}(\mathbf{A}_{i'j}, \mathbf{A}_{i''j} \mid i', i'' \in \mathcal{N}_{i\backslash j}) = \frac{\mathbb{P}(i', i'' \in \mathcal{N}_{i\backslash j} \mid \mathbf{A}_{i'j}, \mathbf{A}_{i''j})\mathbb{P}(\mathbf{A}_{i'j}, \mathbf{A}_{i''j})}{\mathbb{P}(i', i'' \in \mathcal{N}_{i\backslash j})}, \tag{3}$$

where we are using the parameterized distance $d_j$ to build the neighborhood $\mathcal{N}_{i \setminus j}$. In this case, we apply the independence argument above to see that

$$\mathbb{P}(i', i'' \in \mathcal{N}_{i \setminus j} \mid \mathbf{A}_{i'j}, \mathbf{A}_{i''j}) = \mathbb{P}(i', i'' \in \mathcal{N}_{i \setminus j}).$$

Therefore the denominator cancels with the term in the numerator, and we have

$$\mathbb{P}(\mathbf{A}_{i'j}, \mathbf{A}_{i''j} \mid i', i'' \in \mathcal{N}_{i \setminus j}) = \mathbb{P}(\mathbf{A}_{i'j}, \mathbf{A}_{i''j}) = \mathbb{P}(\mathbf{A}_{i'j})\mathbb{P}(\mathbf{A}_{i''j}). \tag{4}$$

Therefore $\mathbf{A}_{i'j}$ and $\mathbf{A}_{i''j}$ are independent even when conditioning on the event $i' \in \mathcal{N}_{i \setminus j}$ and $i'' \in \mathcal{N}_{i \setminus j}$. This allows us to apply a concentration inequality to bound each entry of $\hat{P} - P$, and a max norm result follows after a simple union bound.

In total, the modified neighborhood smoothing procedure is as follows: Fix some neighborhood size parameter $h$ and let $q_{i \setminus j}(h)$ denote the $h$-th quantile of the set $\{d_j(i, i') : i' \neq i, j\}$. Construct the neighborhood $\mathcal{N}_{i \setminus j}$ by setting

$$\mathcal{N}_{i \setminus j} = \{i' \neq i, j : d_j(i, i') \leq q_{i \setminus j}(h)\}.$$

Then set

$$\hat{P}_{ij} = \frac{1}{2}\left( \frac{1}{|\mathcal{N}_{i \setminus j}|} \sum_{i' \in \mathcal{N}_{i \setminus j}} A_{i'j} + \frac{1}{|\mathcal{N}_{j \setminus i}|} \sum_{j' \in \mathcal{N}_{j \setminus i}} A_{ij'} \right).$$

We will show that this estimator of the edge probability matrix is consistent in max-norm.

### E.3   Claims

There are two major components to the analysis. First, we show that, with high probability, each neighborhood $\mathcal{N}_{i \setminus j}$ consists only of nodes $i'$ for which $\|\mathbf{P}_i - \mathbf{P}_{i'}\|_\infty < \epsilon$, with $\epsilon \to 0$ as $n \to \infty$; The formal statement of this result is made in Claims 30 and 31 below. This is an extension of the analysis in [21], where it is shown that the neighborhood $\mathcal{N}_i$ consists only of nodes $i'$ for which $1/n \|\mathbf{P}_i - \mathbf{P}_{i'}\|_2 < \epsilon$, with $\epsilon \to 0$ as $n \to \infty$. The procedure for proving this result parallels that of [21], however, the modifications we make to the algorithm – namely, the deletion of a node from the graph – mean that the claims in that paper do not directly transfer. Much of the analysis consists of making the minor changes necessary to show that analogous versions of the claims in [21] hold for our modified algorithm.

The second part of the analysis uses concentration inequalities to derive the consistency result. In particular, Claim 32 shows that smoothing within neighborhoods produces an estimate of the edge probability matrix which is close within max-norm, provided that each neighborhood consists only of nodes which are sufficiently similar in the sense described above. Theorem 3 puts these two claims together to derive the main result.

The technical details of the analysis are in Appendix E.5. In particular, all proofs of the following claims can be found there.

#### E.3.1   Sample requirements

The analysis will require the notion of a block partition and $\Delta$-refinement as defined in Definition 7 and Definition 13, respectively, both in Appendix D.1. If $\mathcal{R}$ is a block partition and $x \in [0, 1]$, we write $\mathcal{R}(x)$ to denote the block $R \in \mathcal{R}$ which contains $x$. Some of the following results will include an assumption that there are "enough" samples in each block of a partition. We formalize this notion as follows:

**Definition 15.** *If* $\mathbf{S}$ *is an ordered set of random samples from the uniform distribution on the unit interval, and* $\mathcal{B}$ *is any block partition, we say that* $\mathbf{S}$ *is a* $\rho$*-dense sample in* $\mathcal{B}$ *if for any block* $B \in \mathcal{B}$,*

$$\frac{|B \cap \mathbf{S}|}{n} > (1 - \rho)\mu(B).$$

If we fix any $\rho$ and a $\Delta$-block partition, a random sample $\mathbf{S}$ will be $\rho$-dense with high probability as the size of the sample $n \to \infty$, as the following result shows:

**Claim 28.** *Let* $\mathcal{B}$ *be a* $\Delta$*-block partition. Let* $\rho < 1$. *Then with probability* $1 - \frac{2}{\Delta}e^{-2n\rho^2\Delta^2}$, $\mathbf{S}$ *is a* $\rho$*-dense sample of* $\mathcal{B}$. *That is, for all* $B \in \mathcal{B}$ *simultaneously,*

$$\frac{|B \cap \mathbf{S}|}{n} > (1 - \rho)|B|.$$

### E.3.2 The adjacency column distance

In the previous section detailing our modified neighborhood smoothing algorithm, we introduced the following distance $d_j(i, i')$ between columns of the adjacency matrix. For a square matrix $M$, let $\partial_v M$ denote the matrix obtained by replacing the $v$-th row and column of the matrix $M$ with zeros. Then we define

$$d_j(i, i') = \max_{k \neq i, i'} \left| \left[ \left( \partial_j \mathbf{A} \right)^2 / n \right]_{ik} - \left[ \left( \partial_j \mathbf{A} \right)^2 / n \right]_{i'k} \right|.$$

This pattern – the maximum elementwise difference of normalized squared matrices – will reoccur in the analysis. We therefore define:

**Definition 16.** *Let $M_1$ and $M_2$ be $n \times n$ matrices. We define*

$$D(M_1, M_2) = \max_{i,j} \left| \left[ M_1^2 / n \right]_{ij} - \left[ M_2^2 / n \right]_{ij} \right|.$$

A key observation in the analysis of [21] is that if $\mathbf{A}$ is sampled from $P$, then $\mathbb{P}(D(\mathbf{A}, P) < \epsilon) \to 0$ as $n \to \infty$. In our analysis, however, we will work with $\partial_k \mathbf{A}$ and $\partial_k P$, which are the adjacency and edge probability matrices with the $k$th row and column set to zero. We therefore have a slightly modified claim:

**Claim 29.** *Let $P$ be an arbitrary $n \times n$ edge probability matrix. Let $C_2 > 0$ be an arbitrary constant and suppose $n$ is large enough that $\sqrt{\frac{(C_2+2)\log n}{n}} \leq 1$. Then, with probability $1 - 2n^{-C_2/4}$ over random adjacency matrices $\mathbf{A}$ sampled from $P$, for all $k \in [n]$ simultaneously,*

$$D(\partial_k \mathbf{A}, \partial_k P) = \max_{i \neq j} \left| \left[ (\partial_k \mathbf{A})^2 / n \right]_{ij} - \left[ (\partial_k P)^2 / n \right]_{ij} \right| \leq \sqrt{\frac{(C_2 + 2)\log n}{n}} + \frac{6}{n}.$$

### E.3.3 Composition of neighborhoods

Another key step in the analysis of [21] is that, with high probability, for any $i'$ in the neighborhood of node $i$, $1/n \left\| P_{i'} - P_i^2 \right\|_2 = O(\sqrt{\log n / n})$. We derive a similar result for our modified neighborhoods:

**Claim 30.** *Let $W \in \mathscr{W}_{\mathcal{B}}^{\mathsf{c}}$ and let $\mathcal{R}$ be a $\Delta$-refinement of $\mathcal{B}$. Suppose $S$ is a $\rho$-dense sample of $\mathcal{R}$ and let $P$ be the induced edge probability matrix. Suppose $A$ is an adjacency matrix such that $D(\partial_k A, \partial_k P) < \epsilon$ for every $k \in [n]$. Pick $0 < h \leq \rho\Delta$, and construct for every pair $i, j$ a neighborhood $\mathcal{N}_{i \backslash j}$ as described above, including all nodes within the $h$-th quantile. Then for all $i, j$ and any $i' \in \mathcal{N}_{i \backslash j}$ we have*

$$\frac{1}{n} \|P_i - P_{i'}\|_2^2 \leq 6\mathsf{c}\Delta + 8\epsilon + \frac{5}{n}.$$

Additionally, we prove that neighborhoods are composed of nodes whose corresponding columns of $P$ are close in $\infty$-norm. This follows from the previous claim after leveraging the piecewise Lipschitz condition.

**Claim 31.** *Let $W \in \mathscr{W}_{\mathcal{B}}^{\mathsf{c}}$ and let $\mathcal{R}$ be a $\Delta$-refinement of $\mathcal{B}$. Suppose $S$ is a $\rho$-dense sample of $\mathcal{R}$. Then for any $\epsilon \geq 4\rho\Delta^3\mathsf{c}^2$, if $i \neq j$ are such that $\frac{1}{n} \left\| P_i - P_j \right\|_2^2 \leq \epsilon$, then $\|P_i - P_j\|_\infty^2 \leq \frac{4\epsilon}{\rho\Delta}$.*

### E.3.4 Main result

Intuitively, if every neighborhood $\mathcal{N}_{i \backslash j}$ is composed of nodes whose corresponding columns of $P$ are close in $\infty$-norm, and whose $j$th elements are statistically independent, we may apply a concentration inequality to conclude that the estimate $\hat{\mathbf{P}}_{ij}$ is close to $P_{ij}$. The following claim makes this precise.

**Claim 32.** *Let $W \in \mathscr{W}_{\mathcal{B}}^{\mathsf{c}}$ and let $\mathcal{R}$ be a $\Delta$-refinement of $\mathcal{B}$. Let $S \in [0, 1]^n$ be fixed, and let $P$ be the edge probability matrix induced by $S$. Assume that with probability $1 - \delta$ over graphs generated from $P$, that for all $i \neq j$ simultaneously, $\|P_i - P_{i'}\|_\infty < \epsilon$ for all $i' \in \mathcal{N}_{i \backslash j}$. Then with probability at least $(1 - \delta)\left[ 1 - 2n(n-1)e^{-2hnt^2} \right]$,*

$$\max_{ij} \left| \hat{\mathbf{P}}_{ij} - P_{ij} \right| < \epsilon + t.$$

We combine all of the previous claims to derive our main result.

**Theorem 3.** *Let $W \in \mathcal{W}_{\mathcal{B}}^{\mathsf{c}}$. Let $\mathbf{P}$ be the random edge probability matrix arising by sampling a graph of size n from W according to the graphon sampling procedure, and denote by $\hat{\mathbf{P}}$ the estimated edge probability using our modified neighborhood smoothing method. Then*

$$\max_{i \neq j} \left| \hat{\mathbf{P}}_{ij} - \mathbf{P}_{ij} \right| = O_p \left( \left[ \frac{\log n}{n} \right]^{1/6} \right).$$

## E.4 Supplementary claims

The following claims will be used in the proofs of Appendix E.5, and are gathered here for convenience. The proofs of these claims are located in Appendix E.5 as well.

**Claim 33.** *Let $\mathcal{R}_2$ be a block partition. Suppose $\mathcal{R}_1$ is a $\Delta$-refinement of $\mathcal{R}_2$. If a S is a $\rho$-dense sample of $\mathcal{R}_1$, then it is also a $\rho$-dense sample of $\mathcal{R}_2$.*

**Claim 34.** *Let M be an $n \times n$ matrix with values in $[0, 1]$. Then for any distinct $u, u', v \in [n]$,*

$$\|(\partial_v M)_u - (\partial_v M)_{u'}\|_2^2 \geq \|M_u - M_{u'}\|_2^2 - 1$$

**Claim 35.** *Let M be an $n \times n$ symmetric matrix with values in $[0, 1]$. Then for any distinct $i, j, k \in [n]$,*

$$\left[ M^2 \right]_{ij} - 1 \leq \left[ (\partial_k M)^2 \right]_{ij} \leq \left[ M^2 \right]_{ij}.$$

**Claim 36.** *Let M be an $n \times n$ symmetric matrix. Then for any distinct $i, i' \in [n]$,*

$$\|M_i - M_{i'}\|_2^2 = \left( M^2 \right)_{ii} - 2 \left( M^2 \right)_{ii'} + \left( M^2 \right)_{i'i'}.$$

**Claim 37.** *Let $W \in \mathcal{W}_{\mathcal{B}}^{\mathsf{c}}$. Suppose $\mathcal{R}$ is a $\Delta$-refinement of $\mathcal{B}$. Let $S = (x_1, \dots, x_n)$ be a fixed sample. Fix $\Delta > 0$ and assume that $|\mathcal{R}(x_i) \cap S| \geq 4$ for every $i \in [n]$. Let P be the edge probability matrix induced by S. Let A be an adjacency matrix, and suppose that A is such that $D(\partial_k A, \partial_k P) < \epsilon$ for all $k \in [n]$. Then for all $i \neq j \neq k$ simultaneously,*

$$2d_k(i, j) + \frac{1}{n} + 4\mathsf{c}\Delta + 4\epsilon \geq \frac{1}{n} \left\| P_i - P_j \right\|_2^2$$

*for $d_k$ computed w.r.t. A.*

**Claim 38.** *Let $W \in \mathcal{W}_{\mathcal{B}}^{\mathsf{c}}$. Fix a sample $S = (x_1, \dots, x_n)$ and let P be the induced edge probability matrix. Suppose that $\mathcal{R}$ is a $\Delta$-refinement of $\mathcal{B}$. Now suppose that nodes $x_i$ and $x_{i'}$ are from the same $\mathcal{R}(x_{i''})$ for some $i''$. Furthermore, suppose that A is an adjacency matrix with the property that $D(\partial_j A, \partial_j P) \leq \epsilon$ for all $j \in [n]$. Then for all $j \neq i, i'$,*

$$d_j(i, i') \leq \mathsf{c}\Delta + 2\epsilon + \frac{2}{n}.$$

## E.5 Proofs

**Claim 28.** *Let $\mathcal{B}$ be a $\Delta$-block partition. Let $\rho < 1$. Then with probability $1 - \frac{2}{\Delta} e^{-2n\rho^2\Delta^2}$, $\mathbf{S}$ is a $\rho$-dense sample of $\mathcal{B}$. That is, for all $B \in \mathcal{B}$ simultaneously,*

$$\frac{|B \cap \mathbf{S}|}{n} > (1 - \rho)|B|.$$

*Proof.* Let B be an arbitrary block in the partition $\mathcal{B}$. Since $\mathcal{B}$ is a $\Delta$-partition, the size of any block is between $\Delta/2$ and $\Delta$. Therefore there are at most $2/\Delta$ blocks in $\mathcal{B}$.

The membership of any given sample in B is a Bernoulli trial with probability $|B|$ of success. Applying Hoeffding's inequality:

$$\mathbb{P}\left( \left| \frac{1}{n} |B \cap \mathbf{S}| - |B| \right| > \epsilon \right) < e^{-2n\epsilon^2}.$$

Choose $\epsilon = \rho\Delta$. This gives

$$\mathbb{P}\left(\left|\frac{1}{n}\,|B \cap \mathbf{S}| - |B|\right| > \rho\Delta\right) < e^{-2n\rho^2\Delta^2},$$

which implies

$$\mathbb{P}\left(\frac{1}{n}\,|B \cap \mathbf{S}| > |B| - \rho\Delta\right) < e^{-2n\rho^2\Delta^2}.$$

Now, $|B| \leq \Delta$, so that for any arbitrary $B$ it is true that

$$\mathbb{P}\left(\frac{1}{n}\,|B \cap \mathbf{S}| > |B| - \rho|B|\right) < e^{-2n\rho^2\Delta^2}.$$

The result follows by applying a union bound over all blocks of the partition, of which there are at most $2/\Delta$. $\qquad\square$

**Claim 29.** *Let $P$ be an arbitrary $n \times n$ edge probability matrix. Let $C_2 > 0$ be an arbitrary constant and suppose $n$ is large enough that $\sqrt{\frac{(C_2+2)\log n}{n}} \leq 1$. Then, with probability $1 - 2n^{-C_2/4}$ over random adjacency matrices $\mathbf{A}$ sampled from $P$, for all $k \in [n]$ simultaneously,*

$$D(\partial_k \mathbf{A}, \partial_k P) = \max_{i \neq j}\left|\left[(\partial_k \mathbf{A})^2/n\right]_{ij} - \left[(\partial_k P)^2/n\right]_{ij}\right| \leq \sqrt{\frac{(C_2+2)\log n}{n}} + \frac{6}{n}.$$

*Proof.* The proof of Lemma 5.2 in [21] establishes that, given the above assumptions, with probability $1 - 2n^{C_2/4}$,

$$D(\mathbf{A}, P) = \max_{i \neq j}\left|\left[\mathbf{A}^2/n\right]_{ij} - \left[P^2/n\right]_{ij}\right| \leq \sqrt{\frac{(C_2+2)\log n}{n}} + \frac{4}{n}.$$

From Claim 35, for all $k$,

$$\left|\left[\mathbf{A}^2/n\right]_{ij} - \left[(\partial_k \mathbf{A})^2/n\right]_{ij}\right| \leq \frac{1}{n},$$

$$\left|\left[P^2/n\right]_{ij} - \left[(\partial_k P)^2/n\right]_{ij}\right| \leq \frac{1}{n},$$

and so, with probability $1 - 2n^{-C_2/4}$,

$$\max_{i \neq j}\left|\left[(\partial_k \mathbf{A})^2/n\right]_{ij} - \left[(\partial_k P)^2/n\right]_{ij}\right| \leq \max_{i \neq j}\left|\left[\mathbf{A}^2/n\right]_{ij} - \left[P^2/n\right]_{ij}\right| + \frac{2}{n}$$

$$\leq \sqrt{\frac{(C_2+2)\log n}{n}} + \frac{6}{n}.$$

$\qquad\square$

**Claim 30.** *Let $W \in \mathscr{W}_{\mathcal{B}}^{\mathsf{c}}$ and let $\mathcal{R}$ be a $\Delta$-refinement of $\mathcal{B}$. Suppose $S$ is a $\rho$-dense sample of $\mathcal{R}$ and let $P$ be the induced edge probability matrix. Suppose $A$ is an adjacency matrix such that $D(\partial_k A, \partial_k P) < \epsilon$ for every $k \in [n]$. Pick $0 < h \leq \rho\Delta$, and construct for every pair $i, j$ a neighborhood $\mathcal{N}_{i\backslash j}$ as described above, including all nodes within the $h$-th quantile. Then for all $i, j$ and any $i' \in \mathcal{N}_{i\backslash j}$ we have*

$$\frac{1}{n}\,\|P_i - P_{i'}\|_2^2 \leq 6\mathsf{c}\Delta + 8\epsilon + \frac{5}{n}.$$

*Proof.* We start by applying Claim 37, which yields

$$\frac{1}{n}\,\|P_i - P_{i'}\|_2^2 \leq 2d_j(i, i') + \frac{1}{n} + 4\mathsf{c}\Delta + 4\epsilon$$

Now we upper bound $d_j(i, i')$. Since we have assumed that $h \leq \rho$, at least a fraction $h$ of the nodes are within $i$'s partition in the refinement. Therefore the distance between any two nodes in the

neighborhood is bounded above by the maximum distance between two nodes in this partition. This was computed in Claim 38, such that:

$$\frac{1}{n} \|P_i - P_{i'}\|_2^2 \leq 2\left(\mathsf{c}\Delta + 2\epsilon + \frac{2}{n}\right) + \frac{1}{n} + 4\mathsf{c}\Delta + 4\epsilon$$

$$= 6\mathsf{c}\Delta + 8\epsilon + \frac{5}{n}.$$

$\square$

**Claim 31.** *Let $W \in \mathscr{W}_{\mathcal{B}}^{\mathsf{c}}$ and let $\mathcal{R}$ be a $\Delta$-refinement of $\mathcal{B}$. Suppose $S$ is a $\rho$-dense sample of $\mathcal{R}$. Then for any $\epsilon \geq 4\rho\Delta^3\mathsf{c}^2$, if $i \neq j$ are such that $\frac{1}{n}\left\|P_i - P_j\right\|_2^2 \leq \epsilon$, then $\|P_i - P_j\|_\infty^2 \leq \frac{4\epsilon}{\rho\Delta}$.*

*Proof.* Suppose $\epsilon \geq 4\rho\Delta^3\mathsf{c}^2$. Define $\alpha = \sqrt{4\epsilon/(\rho\Delta)}$ and suppose that $\|P_i - P_j\|_\infty > \alpha$. This implies that there exists a $k$ such that $|P_{ik} - P_{jk}| > \alpha$. Consider any $k' \in \mathcal{R}(k)$. Then

$$|P_{ik'} - P_{jk'}| = \left|(P_{ik'} - P_{ik}) + P_{ik} - (P_{jk'} - P_{jk}) - P_{jk}\right|$$

$$= \left|(P_{ik'} - P_{ik}) + (P_{jk} - P_{jk'}) + (P_{ik} - P_{jk})\right|$$

Since $|x_k - x_{k'}| < \Delta$ by virtue of being in the same block $\mathcal{R}(x_k)$, we have $|P_{ik'} - P_{ik}| \leq \Delta\mathsf{c}$. But by assumption, $\Delta \leq \alpha/(4\mathsf{c})$. Therefore, $|P_{ik'} - P_{ik}| \leq \alpha/4$. Similarly, $|P_{jk'} - P_{jk}| \leq \alpha/4$. The last term satisfies $|P_{ik} - P_{jk}| > \alpha$. Therefore the entire quantity must be at least:

$$> \alpha/2.$$

Now consider

$$\frac{1}{n}\left\|P_i - P_j\right\|_2^2 = \frac{1}{n}\sum_l (P_{il} - P_{jl})^2$$

$$\geq \frac{1}{n}\sum_{k' \in \mathcal{R}(k)} (P_{ik'} - P_{jk'})^2$$

But, as established above, each term in the sequence is at least $\alpha/2$, and so:

$$> \frac{1}{n}\sum_{k' \in \mathcal{R}(k)} \alpha^2/4$$

Since $S$ is assumed to be a $\rho$-dense sample of $\mathcal{R}$, there are at least $\rho\Delta n$ elements in $\mathcal{R}(k)$. Therefore:

$$\geq \frac{\rho\Delta\alpha^2}{4} = \epsilon$$

The claim follows from the contrapositive. $\square$

**Claim 32.** *Let $W \in \mathscr{W}_{\mathcal{B}}^{\mathsf{c}}$ and let $\mathcal{R}$ be a $\Delta$-refinement of $\mathcal{B}$. Let $S \in [0,1]^n$ be fixed, and let $P$ be the edge probability matrix induced by $S$. Assume that with probability $1 - \delta$ over graphs generated from $P$, that for all $i \neq j$ simultaneously, $\|P_i - P_{i'}\|_\infty < \epsilon$ for all $i' \in \mathcal{N}_{i\backslash j}$. Then with probability at least $(1 - \delta)\left[1 - 2n(n-1)e^{-2hnt^2}\right]$,*

$$\max_{ij}\left|\hat{\mathbf{P}}_{ij} - P_{ij}\right| < \epsilon + t.$$

*Proof.* Consider an arbitrary ordered pair of nodes $i \neq j$. The neighborhood $\mathcal{N}_{i\backslash j}$ is a random variable, since it depends on the random adjacency matrix $\mathbf{A}$. Define $\ell_{i\backslash j}$ to be the amount by which our smoothed estimate computed using $\mathcal{N}_{i\backslash j}$ differs from $P_{ij}$:

$$\ell_{i\backslash j} = \left|P_{ij} - \frac{1}{\mathcal{N}_{i\backslash j}}\sum_{i' \in \mathcal{N}_{i\backslash j}} \mathbf{A}_{i'j}\right|.$$

Note that $\ell_{i \backslash j}$ is itself a random variable, and we seek to compute

$$\mathbb{P}\left(\max_{i \neq j} \ell_{i \backslash j} < \tilde{\epsilon}\right),$$

where it will be assumed that $\tilde{\epsilon} > \epsilon$.

Denote by $\mathcal{N}_{i \backslash j}$ the subset of $2^{[n]}$ consisting of all possible values of the neighborhood $\mathcal{N}_{i \backslash j}$ over all graphs on $[n]$. Denote by $\mathcal{N}_{i \backslash j}^{\epsilon}$ the subset of $\mathcal{N}_{i \backslash j}$ consisting of neighborhoods with the property that that if $i'$ is in the neighborhood, then $\|P_i - P_{i'}\|_\infty < \epsilon$. Then

$$\mathbb{P}\left(\max_{i \neq j} \ell_{i \backslash j} < \tilde{\epsilon}\right) \geq \mathbb{P}\left(\max_{i \neq j} \ell_{i \backslash j} < \tilde{\epsilon} \,\middle|\, \forall\, i \neq j,\ \mathcal{N}_{i \backslash j} \in \mathcal{N}_{i \backslash j}^{\epsilon}\right) \mathbb{P}\left(\forall\, i \neq j,\ \mathcal{N}_{i \backslash j} \in \mathcal{N}_{i \backslash j}^{\epsilon}\right)$$

$$\geq \mathbb{P}\left(\max_{i \neq j} \ell_{i \backslash j} < \tilde{\epsilon} \,\middle|\, \forall\, i \neq j,\ \mathcal{N}_{i \backslash j} \in \mathcal{N}_{i \backslash j}^{\epsilon}\right)(1 - \delta).$$

We now lower bound the probability that an arbitrary pair $u \neq v$ is such that $\ell_{u \backslash v} < \tilde{\epsilon}$. The result will then follow from a union bound. That is, we would like to compute, for arbitrary $u \neq v$, the probability

$$\mathbb{P}\left(\ell_{u \backslash v} < \tilde{\epsilon} \,\middle|\, \forall\, i \neq j,\ \mathcal{N}_{i \backslash j} \in \mathcal{N}_{i \backslash j}^{\epsilon}\right) = \mathbb{P}\left(\ell_{u \backslash v} < \tilde{\epsilon} \,\middle|\, \mathcal{N}_{u \backslash v} \in \mathcal{N}_{u \backslash v}^{\epsilon}\right)$$

We decompose this quantity as a sum over all neighborhoods in $\mathcal{N}_{u \backslash v}^{\epsilon}$:

$$= \sum_{N \in \mathcal{N}_{u \backslash v}^{\epsilon}} \mathbb{P}\left(\ell_{u \backslash v} < \tilde{\epsilon} \,\middle|\, \mathcal{N}_{u \backslash v} = N\right) \mathbb{P}\left(\mathcal{N}_{u \backslash v} = N \,\middle|\, \mathcal{N}_{u \backslash v} \in \mathcal{N}_{u \backslash v}^{\epsilon}\right)$$

We now claim that, conditioned on a particular neighborhood $N$, the random variables $\mathbf{A}_{u_1 v}$ and $\mathbf{A}_{u_2 v}$ are independent. We may then apply Hoeffding's inequality to conclude:

$$\mathbb{P}\left(\left|\frac{1}{|N|} \sum_{u' \in N}(\mathbf{A}_{u' v} - P_{u' v})\right| > t\right) < e^{-2hnt^2}.$$

Where we have used the fact that there are at least $hn$ nodes in the neighborhood $N$. By the assumption that $|P_{uv} - P_{u' v}| < \epsilon$ for any $u \in N$, we have:

$$\mathbb{P}\left(\left|P_{uv} - \frac{1}{|N|} \sum_{u' \in N}\mathbf{A}_{u' v}\right| > t + \epsilon\right) < e^{-2hnt^2}.$$

So that

$$\mathbb{P}\left(\ell_{u \backslash v} < \tilde{\epsilon} \,\middle|\, \forall\, i \neq j,\ \mathcal{N}_{i \backslash j} \in \mathcal{N}_{i \backslash j}^{\epsilon}\right) = \sum_{N \in \mathcal{N}_{u \backslash v}^{\epsilon}} \mathbb{P}\left(\ell_{u \backslash v} < \tilde{\epsilon} \,\middle|\, \mathcal{N}_{u \backslash v} = N\right) \mathbb{P}\left(\mathcal{N}_{u \backslash v} = N \,\middle|\, \mathcal{N}_{u \backslash v} \in \mathcal{N}_{u \backslash v}^{\epsilon}\right)$$

$$> \left(1 - e^{-2hnt^2}\right) \sum_{N \in \mathcal{N}_{u \backslash v}^{\epsilon}} \mathbb{P}\left(\mathcal{N}_{u \backslash v} = N \,\middle|\, \mathcal{N}_{u \backslash v} \in \mathcal{N}_{u \backslash v}^{\epsilon}\right)$$

$$= \left(1 - e^{-2hnt^2}\right)$$

Now, returning to:

$$\mathbb{P}\left(\max_{i \neq j} \ell_{i \backslash j} < \tilde{\epsilon}\right) \geq \mathbb{P}\left(\max_{i \neq j} \ell_{i \backslash j} < \tilde{\epsilon} \,\middle|\, \forall\, i \neq j,\ \mathcal{N}_{i \backslash j} \in \mathcal{N}_{i \backslash j}^{\epsilon}\right)(1 - \delta)$$

We apply a union bound over all $2n(n - 1)$ ordered pairs to obtain:

$$> (1 - \delta)\left(1 - 2n(n - 1)\left[1 - e^{-2hnt^2}\right]\right)$$

$\square$

**Theorem 3.** *Let $W \in \mathcal{W}_{\mathcal{B}}^{\mathsf{c}}$. Let $\mathbf{P}$ be the random edge probability matrix arising by sampling a graph of size n from W according to the graphon sampling procedure, and denote by $\hat{\mathbf{P}}$ the estimated edge probability using our modified neighborhood smoothing method. Then*

$$\max_{i \neq j} \left| \hat{\mathbf{P}}_{ij} - \mathbf{P}_{ij} \right| = O_p \left( \left[ \frac{\log n}{n} \right]^{1/6} \right).$$

*Proof.* The mechanism of the proof involves a translation from the $L^2$ result of [21] to our desired max-norm result. To accomplish this, we will make use of two discretizations at different scales. First, define arbitrary constants $\alpha_2, \alpha_\infty > 0$ and $0 < \rho < 1$ such that $\rho \cdot \alpha_2 > \frac{1}{2}$, and let

$$\Delta_2(n) = \alpha_2 \sqrt{\frac{\log n}{n}}, \qquad \Delta_\infty(n) = \alpha_\infty \left( \frac{\log n}{n} \right)^{1/6},$$

for any $n \geq 2$. For each $n \geq 2$, let $\mathcal{R}_\infty(n)$ be an arbitrary $\Delta_\infty(n)$-refinement of $\mathcal{B}$, and let $\mathcal{R}_2(n)$ be an arbitrary $\Delta_2(n)$-refinement of $\mathcal{R}_\infty(n)$. In what follows we will drop the functional notation, as the dependence of these quantities on $n$ should be clear.

Let $\mathbf{S}$ be a random sample of $[0, 1]$. Then, according to Claim 28, $\mathbf{S}$ is $\rho$-dense in $\mathcal{R}_2$ with probability

$$1 - \frac{2}{\Delta_2} e^{-2n\rho^2 \Delta_2^2} = 1 - 2\alpha_2 \sqrt{\frac{n}{\log n}} e^{-2n\rho^2 \alpha_2^2 \frac{\log n}{n}}$$

$$= 1 - 2\alpha_2 \sqrt{\frac{n}{\log n}} n^{-2\alpha_2^2 \rho^2}$$

$$\geq 1 - 2\alpha_2 \sqrt{n} \cdot n^{-2\alpha_2^2 \rho^2}$$

$$= 1 - 2\alpha_2 n^{\frac{1}{2} - 2\alpha_2^2 \rho^2}$$

Since $\rho \cdot \alpha_2 > 1/2$ by assumption, this is a decreasing function in $n$.

We have so-far shown that a sample is "good" with high probability in the sense that it is $\rho$-dense in $\mathcal{R}_2$. We now show that, assuming the sample $\mathbf{S} = S$ is a fixed, $\rho$-dense sample of $\mathcal{R}_2$, the estimate $\hat{P}$ is good in max-norm with high probability over random graphs sampled according to the distribution induced by $S$.

We begin by showing that, with high probability, the neighborhood around node $i$ contains only nodes $i'$ such that $P_i$ and $P_{i'}$ are close in 2-norm, which will follow from combining Claims 30 and 29. We will use this result to invoke Claim 31, which says that, for $i'$ in the neighborhood of $i$, $P_i$ and $P_{i'}$ are close in $\infty$-norm. This will in turn satisfy the assumptions of Claim 32, which shows that $\hat{P}$ is close to $P$.

First, we combine Claims 30 and 29 to show that, with high probability, $\frac{1}{n} \|P_i - P_{i'}\|_2^2$ is small when $i'$ is in $\mathcal{N}_{i \backslash j}$. Fix an arbitrary constant $C_2 > 0$ and suppose that $n$ is large enough that $\sqrt{(C_2 + 2) \log n / n} \leq 1$. Then Claim 29 says that, with probability $1 - 2n^{-C_2/4}$ over random adjacency matrices $\mathbf{A}$ generated by $P$, for all $k \in [n]$ simultaneously,

$$D(\partial_k \mathbf{A}, \partial_k P) \leq \sqrt{\frac{(C_2 + 2) \log n}{n}} + \frac{6}{n}.$$

Using $\mathcal{R}_2$ as the partition in Claim 30, we find that this implies that the adjacency matrix $\mathbf{A}$ is such that for any $i, j$ and $i' \in \mathcal{N}_{i \backslash j}$,

$$\frac{1}{n} \|P_i - P_{i'}\|_2^2 \leq 6c\Delta_2 + 8 \left( \sqrt{\frac{(C_2 + 2) \log n}{n}} + \frac{6}{n} \right) + \frac{5}{n}$$

$$\leq 6c\alpha_2 \sqrt{\frac{\log n}{n}} + 8 \sqrt{\frac{(C_2 + 2) \log n}{n}} + \frac{53}{n}$$

$$= \left( 6c\alpha_2 + 8 \sqrt{C_2 + 2} \right) \sqrt{\frac{\log n}{n}} + \frac{53}{n}$$

$$\leq \tilde{\alpha}_2 \sqrt{\frac{\log n}{n}}$$

where $\tilde{\alpha}_2$ is an arbitrary constant greater than $6c\alpha_2 + 8\sqrt{C_2 + 2}$, and assuming that $n$ is large enough that $53/n \le \tilde{\alpha}_2 - 6c\alpha_2 + 8\sqrt{C_2 + 2}$.

Now we may invoke Claim 31 using $\mathcal{R}_\infty$ as the refinement of $\mathcal{B}$. Define $\gamma = \max\{\tilde{\alpha}_2, 4\rho\alpha_\infty^3 c^2\}$ and let $\tilde{\epsilon} = \gamma \sqrt{\frac{\log n}{n}}$. Then, from the previous result, for any $i' \in \mathcal{N}_{i\setminus j}$, $\frac{1}{n}\|P_i - P_{i'}\|_2^2 \le \gamma \sqrt{\frac{\log n}{n}}$. Furthermore,

$$\tilde{\epsilon} = \gamma \sqrt{\frac{\log n}{n}} \ge 4\rho\alpha_\infty^3 c^2 \sqrt{\frac{\log n}{n}} = 4\rho c^2 \left[\alpha_\infty \left(\frac{\log n}{n}\right)^{1/6}\right]^3 = 4\rho c^2 \Delta_\infty^3$$

and so we may use the claim to conclude that, with probability at least $1 - 2n^{-C_2/4}$ over graphs generated from $P$, for all $i, j \in [n]$ and any $i' \in \mathcal{N}_{i\setminus j}$,

$$\|P_i - P_{i'}\|_\infty^2 \le \frac{4\gamma}{\rho\Delta_\infty} \sqrt{\frac{\log n}{n}} = \frac{4\gamma}{\rho \cdot \alpha_\infty} \left(\frac{\log n}{n}\right)^{1/3}.$$

Now we may apply Claim 32. Let $\alpha_t$ be an arbitrary constant, and choose

$$t = \alpha_t \left(\frac{\log n}{n}\right)^{1/6}.$$

Then, with probability

$$\left(1 - 2n^{-C_2/4}\right)\left(1 - n^{2 - 2h\alpha_t \left(\frac{n}{\log n}\right)^{2/3}}\right),$$

it holds that

$$\max_{ij} \left|\hat{\mathbf{P}}_{ij} - P_{ij}\right| < \left(\alpha_t + \sqrt{\frac{4\gamma}{\rho \cdot \alpha_\infty}}\right)\left(\frac{\log n}{n}\right)^{1/6}.$$

The probability over all samples and graphs is therefore

$$\left(1 - 2n^{-C_2/4}\right)\left(1 - n^{2 - 2h\alpha_t \left(\frac{n}{\log n}\right)^{2/3}}\right)\left(1 - 2\alpha_2 \sqrt{\frac{n}{\log n}} n^{-2\alpha_2^2\rho^2}\right).$$

$\square$

**Claim 33.** *Let $\mathcal{R}_2$ be a block partition. Suppose $\mathcal{R}_1$ is a $\Delta$-refinement of $\mathcal{R}_2$. If a $S$ is a $\rho$-dense sample of $\mathcal{R}_1$, then it is also a $\rho$-dense sample of $\mathcal{R}_2$.*

*Proof.* Suppose $S$ is a $\rho$-dense sample of $\mathcal{R}_1$. Take any block $R \in \mathcal{R}_2$. Then $R$ is the disjoint union of blocks in $\mathcal{R}_1$:

$$R = R_1 \cup \ldots \cup R_t$$

where $R_i \in \mathcal{R}_1$. Each $R_i$ is such that $|R_i| \le \Delta$. Therefore:

$$\frac{|R \cap S|}{n} = \sum_i \frac{|R_i \cap S|}{n} \ge \sum_i (1 - \rho)|R_i| = (1 - \rho)\sum_i |R_i| = (1 - \rho)|R|.$$

Therefore $S$ is a $\rho$-dense sample of $\mathcal{R}_2$. $\square$

**Claim 34.** *Let $M$ be an $n \times n$ matrix with values in $[0, 1]$. Then for any distinct $u, u', v \in [n]$,*

$$\|(\partial_v M)_u - (\partial_v M)_{u'}\|_2^2 \ge \|M_u - M_{u'}\|_2^2 - 1$$

*Proof.*

$$\begin{aligned}
\|(\partial_v M)_u - (\partial_v M)_{u'}\|_2^2 &= \sum_t ((\partial_v M)_{ut} - (\partial_v M)_{u't})^2 \\
&= \sum_t (M_{ut} - M_{u't})^2 - (M_{uv} - M_{u'v})^2 \\
&= \|M_u - M_{u'}\|_2^2 - (M_{uv} - M_{u'v})^2 \\
&\ge \|M_u - M_{u'}\|_2^2 - 1
\end{aligned}$$

$\square$

**Claim 35.** *Let M be an n×n symmetric matrix with values in [0, 1]. Then for any distinct i, j, k ∈ [n],*

$$\left[M^2\right]_{ij} - 1 \le \left[(\partial_k M)^2\right]_{ij} \le \left[M^2\right]_{ij}.$$

*Proof.* We have

$$\begin{aligned}
\left[(\partial_k M)^2\right]_{ij} &= \sum_{l \ne k} M_{il} M_{lj} \\
&= \sum_l M_{il} M_{lj} - M_{ik} M_{kj} \\
&= \left[M^2\right]_{ij} - M_{ik} M_{kj}
\end{aligned}$$

The product $M_{ik} M_{kj}$ is at most one and at least zero, which proves the claim. $\square$

**Claim 36.** *Let M be an $n \times n$ symmetric matrix. Then for any distinct $i, i' \in [n]$,*

$$\|M_i - M_{i'}\|_2^2 = \left(M^2\right)_{ii} - 2\left(M^2\right)_{ii'} + \left(M^2\right)_{i'i'}.$$

*Proof.* For any $u, v$ we have

$$\left(M^2\right)_{uv} = \sum_k M_{uk} M_{vk}.$$

Therefore,

$$\begin{aligned}
\left(M^2\right)_{ii} - 2\left(M^2\right)_{ii'} + \left(M^2\right)_{i'i'} &= \sum_k M_{ik}^2 - 2\sum_k M_{ik} M_{i'k} + \sum_k M_{i'k}^2 \\
&= \sum_k (M_{ik} - M_{i'k})^2 \\
&= \|M_i - M_{i'}\|_2^2.
\end{aligned}$$

$\square$

**Claim 37.** *Let $W \in \mathscr{W}_{\mathcal{B}}^{\mathsf{c}}$. Suppose $\mathcal{R}$ is a $\Delta$-refinement of $\mathcal{B}$. Let $S = (x_1, \ldots, x_n)$ be a fixed sample. Fix $\Delta > 0$ and assume that $|\mathcal{R}(x_i) \cap S| \ge 4$ for every $i \in [n]$. Let P be the edge probability matrix induced by $S$. Let A be an adjacency matrix, and suppose that A is such that $D(\partial_k A, \partial_k P) < \epsilon$ for all $k \in [n]$. Then for all $i \ne j \ne k$ simultaneously,*

$$2d_k(i, j) + \frac{1}{n} + 4\mathsf{c}\Delta + 4\epsilon \ge \frac{1}{n} \|P_i - P_j\|_2^2$$

*for $d_k$ computed w.r.t. A.*

*Proof.* We may apply Claim 34 to obtain

$$\frac{1}{n} \|P_i - P_j\|_2^2 \le \frac{1}{n} \|(\partial_k P)_i - (\partial_k P)_j\|_2^2 + \frac{1}{n}$$

which may be expanded using Claim 36, yielding:

$$\begin{aligned}
&= \left[(\partial_k P)^2 / n\right]_{ii} - 2\left[(\partial_k P)^2 / n\right]_{ij} + \left[(\partial_k P)^2 / n\right]_{jj} + \frac{1}{n} \\
&\le \left|\left[(\partial_k P)^2 / n\right]_{ii} - \left[(\partial_k P)^2 / n\right]_{ij}\right| + \left|\left[(\partial_k P)^2 / n\right]_{jj} - \left[(\partial_k P)^2 / n\right]_{ij}\right| + \frac{1}{n}
\end{aligned}$$

By virtue of the fact that every block $\mathcal{R}(x_i)$ in the refinement contains at least 4 points, we may find an $x_{\tilde{i}} \in \mathcal{R}(x_i) \cap S$ and $\tilde{j} \in \mathcal{R}(x_j) \cap S$ such that $\tilde{i} \ne i, k$ and $\tilde{j} \ne j, k$. It is clear that $\left[(\partial_k P)^2 / n\right]_{ii}$ differs from $\left[(\partial_k P)^2 / n\right]_{\tilde{i}\tilde{i}}$ by at most $\mathsf{c}\Delta$, and similarly for the other terms. Hence

$$\le \left|\left[(\partial_k P)^2 / n\right]_{\tilde{i}\tilde{i}} - \left[(\partial_k P)^2 / n\right]_{\tilde{i}\tilde{j}}\right| + \left|\left[(\partial_k P)^2 / n\right]_{\tilde{j}\tilde{j}} - \left[(\partial_k P)^2 / n\right]_{\tilde{i}\tilde{j}}\right| + \frac{1}{n} + 4\mathsf{c}\Delta$$

Next we apply the assumption that $D(\partial_k A, \partial_k P) < \epsilon$:

$$\leq \left| \left[ (\partial_k A)^2 / n \right]_{\tilde{i}\tilde{i}} - \left[ (\partial_k A)^2 / n \right]_{ij} \right| + \left| \left[ (\partial_k A)^2 / n \right]_{j\tilde{j}} - \left[ (\partial_k A)^2 / n \right]_{i\tilde{j}} \right| + \frac{1}{n} + 4c\Delta + 4\epsilon$$

$$\leq 2 \max_{l \neq i,j} \left| \left[ (\partial_k A)^2 / n \right]_{il} - \left[ (\partial_k A)^2 / n \right]_{jl} \right| + \frac{1}{n} + 4c\Delta + 4\epsilon$$

We recognize this as:

$$= 2d_k(i,j) + \frac{1}{n} + 4c\Delta + 4\epsilon.$$

$\square$

**Claim 38.** *Let $W \in \mathcal{W}_{\mathcal{B}}^c$. Fix a sample $S = (x_1, \ldots, x_n)$ and let $P$ be the induced edge probability matrix. Suppose that $\mathcal{R}$ is a $\Delta$-refinement of $\mathcal{B}$. Now suppose that nodes $x_i$ and $x_{i'}$ are from the same $\mathcal{R}(x_{i''})$ for some $i''$. Furthermore, suppose that $A$ is an adjacency matrix with the property that $D(\partial_j A, \partial_j P) \leq \epsilon$ for all $j \in [n]$. Then for all $j \neq i, i'$,*

$$d_j(i, i') \leq c\Delta + 2\epsilon + \frac{2}{n}.$$

*Proof.* We have

$$d_j(i, i') = \max_{k \neq i,i'} \left| \left[ (\partial_j A)^2 / n \right]_{ik} - \left[ (\partial_j A)^2 / n \right]_{i'k} \right|$$

Applying the fact that $D(\partial_j A, \partial_j P) \leq \epsilon$:

$$\leq \max_{k \neq i,i'} \left| \left[ (\partial_j P)^2 / n \right]_{ik} - \left[ (\partial_j P)^2 / n \right]_{i'k} \right| + 2\epsilon$$

Applying Claim 35 yields an additional two terms of $1/n$:

$$\leq \max_{k \neq i,i'} \left| \left[ P^2 / n \right]_{ik} - \left[ P^2 / n \right]_{i'k} \right| + 2\epsilon + \frac{2}{n}$$

The fact that $x_i$ and $x_{i'}$ are from the same block of the $\Delta$-refinement implies that $|x_i - x_{i'}| \leq \Delta$. Hence, by smoothness of $W$, we have that $|P_{ik} - P_{i'k}| \leq c\Delta$ for every $k$. It is therefore the case that for any $k$ $\left| \left[ P^2 / n \right]_{ik} - \left[ P^2 / n \right]_{i'k} \right| \leq c\Delta$, as is shown in the proof of Lemma 5.2 in [21]. Therefore:

$$\leq c\Delta + 2\epsilon + \frac{2}{n}.$$

$\square$

# F   Experiments

In this section we apply the graph clustering method proposed in Algorithm 1 to real and synthetic data and discuss the results. The purpose of these experiments is to help the reader develop an intuition for how the clustering method works, and not necessarily to demonstrate superior practical performance. As such, only limited comparisons are made to existing clustering methods.

## F.1   Football dataset

We first apply Algorithm 1 to the football network from [11]. This is a undirected, unweighted graph representing the games played between all NCAA Division I-A American college football teams during the regular season in the year 2000. Each team appears as a node in the graph; an edge exists between two teams if and only if they played one another. The graph, shown in Figure 4, includes 115 nodes (teams) and 613 edges (games).

In this year, the teams in Division I-A were divided into eleven football conferences, excepting five "independent" teams which belonged to no conference in particular. The conferences and their associated teams are shown in Table 1. In general, an American college football team will play the majority of its games against opponents belonging to its own conference – though the team will not usually play every other conference member in the same season. The remaining games on the team's schedule are against out-of-conference opponents. For instance, Ohio State belongs to the Big 10 conference, and in this particular season played conference opponents Iowa, Illinois, Purdue, Michigan, Minnesota, Wisconsin, Michigan State, and Penn State, as well as out-of-conference opponents Miami of Ohio, Arizona, and Fresno State. Because of this connection between conference membership and the scheduling of games, it is reasonable to assume that the graph of football games will exhibit cluster structure. In particular, the clusters of the graph should roughly correspond to the eleven football conferences. As such, we apply the neighborhood smoothing and clustering method to this network and compare the resulting clusters to the eleven football conferences.

The input to the algorithm is the adjacency matrix of the football graph, shown in Figure 5(a). Rearranging the rows and columns of the adjacency matrix according to conference membership as shown in Figure 5(b) reveals the network's cluster structure. Note that the algorithm *does not* have access to this rearranged adjacency or the conference membership of each team; it is shown here only for the convenience of the reader. Smoothing was performed with the neighborhood size parameter $C = 0.09$; the parameter was chosen by hand to produce a good clustering. The output $\hat{P}$ of the network smoothing step is shown in Figure 5(c); this matrix after rearranging by conference membership is shown in Figure 5(d).

The effect of neighborhood smoothing is to propagate trends in scheduling to all teams within a conference. For instance, consider the ACC and Big East conferences. As Figure 5(b) shows, in this season there were five games played between these conferences. Most ACC teams played at least one Big East opponent, but some ACC teams played no Big East opponent. After applying neighborhood smoothing, however, the estimated probability that any ACC team should play any Big East team is uniformly nonzero, as shown in Figure 5. That is, even if an ACC team played no Big East opponent, the algorithm smooths the estimate of the probability of such a game to be consistent with the other teams in the conference.

In the clustering step, single-linkage clustering is applied to $\hat{P}$, interpreting it as a similarity matrix. The resulting dendrogram is shown in Figure 6. In general the clustering recovers the conferences with high accuracy. In addition, because the clustering is a tree and not a flat partitioning of the teams, more structure is evident. For instance, the clusters corresponding to the MW (Mountain West) conference and the Pac 10 are joined at a high level. This is because the Mountain West and Pac 10 are comprised of teams from the western U.S. and who play one another frequently as such.

Figure 4: The network of American college football games played during the 2000 regular season. Each node in the graph represents a team, and each edge represents a game played.

(a) The input adjacency matrix.

(b) The input adjacency matrix, rearranged according to conference membership.

(c) The result of neighborhood smoothing.

(d) The result of neighborhood smoothing, rearranged according to conference membership.

Figure 5: The neighborhood smoothing step as applied to the football network. The smoothing algorithm only has access to the input adjacency matrix as shown in (a), and not to the re-arranged matrix shown in (b).

| ACC | Big 10 | Big 12 | Big East | C-USA | Independent |
|---|---|---|---|---|---|
| Clemson | Illinois | Baylor | BostonCollege | AlabamaBirmingham | CentralFlorida |
| Duke | Indiana | Colorado | MiamiFlorida | Army | Connecticut |
| FloridaState | Iowa | IowaState | Pittsburgh | Cincinnati | Navy |
| GeorgiaTech | Michigan | Kansas | Rutgers | EastCarolina | NotreDame |
| Maryland | MichiganState | KansasState | Syracuse | Houston | UtahState |
| NorthCarolina | Minnesota | Missouri | Temple | Louisville | |
| NorthCarolinaState | Northwestern | Nebraska | VirginiaTech | Memphis | |
| Virginia | OhioState | Oklahoma | WestVirginia | SouthernMississippi | |
| WakeForest | PennState | OklahomaState | | Tulane | |
| | Purdue | Texas | | | |
| | Wisconsin | TexasA&M | | | |
| | | TexasTech | | | |

| MAC | MW | Pac 10 | SEC | Sunbelt | WAC |
|---|---|---|---|---|---|
| Akron | AirForce | Arizona | Alabama | ArkansasState | BoiseState |
| BallState | BrighamYoung | ArizonaState | Arkansas | Idaho | FresnoState |
| BowlingGreenState | ColoradoState | California | Auburn | LouisianaLafayette | Hawaii |
| Buffalo | NevadaLasVegas | Oregon | Florida | LouisianaMonroe | LouisianaTech |
| CentralMichigan | NewMexico | OregonState | Georgia | MiddleTennesseeState | Nevada |
| EasternMichigan | SanDiegoState | SouthernCalifornia | Kentucky | NewMexicoState | Rice |
| Kent | Utah | Stanford | LouisianaState | NorthTexas | SanJoseState |
| Marshall | Wyoming | UCLA | Mississippi | | SouthernMethodist |
| MiamiOhio | | Washington | MississippiState | | TexasChristian |
| NorthernIllinois | | WashingtonState | SouthCarolina | | TexasElPaso |
| Ohio | | | Tennessee | | Tulsa |
| Toledo | | | Vanderbilt | | |
| WesternMichigan | | | | | |

Table 1: The teams belonging to each conference. Note that the dataset from [11] erroneously assigns Texas Christian to C-USA. Texas Christian was in fact in the WAC in the year 2000, and we have made this correction before performing our analysis.

Figure 6: The result of Algorithm 1 as applied to the football network. Nodes joining at higher levels of the tree are more similar. If all of the leaf nodes in a subtree belong to the same conference, every edge in the subtree is marked with the same color. Different colors are used to distinguish such subtrees, but the particular color used is not meaningful. The conference labels in the figure are used to show where the majority of that conference's teams are in the clustering. Not marked is the Sun Belt conference, the majority of whose teams are placed between the Big East and Big 12, and the independent teams which belong to no conference in particular.

(a) The graphon used in the synthetic experiment. The graphon takes on three values: The darkest region has a height of 0.7; the small, medium-dark blocks are of height 0.5; the remaining light area has value 0.1.

(b) The cluster tree of this graphon. The two left-most blocks join at a height of 0.5. These join with the remaining block at 0.1.

(c) An adjacency matrix sampled from the graphon, rearranged for the presentation (the algorithm receives a random permutation of this matrix.

(d) The smoothed estimate of edge probabilites computed from the adjacency matrix at left.

Figure 7

## F.2 Synthetic network sampled from a graphon

In this experiment we apply Algorithm 1 to a network sampled from the graphon shown in Figure 7a. This graphon was chosen to demonstrate a non-trivial case where a simple clustering method may yield the incorrect result. The graphon consists of three large blocks along the diagonal which take value 0.7. The first two of these blocks are joined by a small region whose value is 0.5. As such, the cluster tree of this graphon is as shown in Figure 7b.

The adjacency matrix of a graph sampled from this graphon is shown in Figure 7c. The matrix in the figure has been rearranged in order to show the cluster structure of the graph; The matrix given as input to the smoothing algorithm is a permutation of this matrix. Smoothing was applied with a neighborhood size parameter of $C = 0.1$. The result is shown in Figure 7d.

In the cluster step, single linkage is applied to the smoothed estimate of edge probabilites. The resulting dendrogram is shown in Figure 8a. Three major clusters are evident in the tree, two of which are joined at a noticeably higher level. As we would expect from a consistent clustering method, the dendrogram resembles the ground-truth cluster tree shown in Figure 7b.

On the other hand, one simple approach to network clustering fails. In this approach, we use the pairwise distance between columns of the adjacency matrix as input to single-linkage clustering. That is, for every $i, j \in \{1, \ldots, n\}$, we use the matrix $D$ whose $i, j$ entry is $\|A_i - A_j\|$, where $A_i$ and $A_j$

(a) The result of applying Algorithm 1 to the synthetic network generated from a graphon.

(b) The result of a simple, inconsistent clustering algorithm which applies single-linkage to the pairwise distances between the columns of the adjacency matrix.

Figure 8

are the $i$th and $j$th columns of $A$, respectively, and $\| \cdot \|$ is a suitable norm – here, we use the 2-norm. Such a simple approach can often work in practice; for example, this method works well on the football network in the previous section. However, as the results shown in Figure 8b demonstrate, it does not work as well for recovering the graphon cluster tree. Though the method appears to recover three clusters, it does not join two of them at a significantly higher level. Therefore the resulting tree does not resemble the graphon cluster tree. In fact, is easily seen that this method is not consistent in the sense described earlier.