[Reviews · NeurIPS 2016]

Reviewer 1

Summary

The paper develops a theory of hierarchical clustering for graphs. They assume that graphs are sampled from a graphon, so they develop a theory of hierarchical clustering for graphons first. To do this, the authors first define what it means for a subset of [0,1] to be connected/disconnected at level \lambda in W. This then automatically defines clusters at level \lambda, for any \lambda \in [0,1]. The set of clusters from all levels \lambda \in [0,1] thus has a natural hierarchical structure, which is called the graphon cluster tree. This is the structure that the authors aim to recover using a graph clustering algorithm. To every graphon cluster tree there is also a naturally associated object that the authors call a mergeon (which is itself a graphon). The main result of the paper is to show that the graphon cluster tree of a piece-wise Lipschitz graphon W can be consistently estimated.

Qualitative Assessment

The paper is pretty well written and the results are interesting. Practically speaking I am not sure how relevant the results and algorithms are though; experiments (even synthetic ones) would be desirable. It would also be interesting to consider the non-dense regime. Nonetheless, I think the hierarchical clustering approach is the right one, and this paper lays some solid theoretical foundations for further explorations. Therefore I believe it could be valuable for the NIPS community.

Confidence in this Review

2-Confident (read it all; understood it all reasonably well)


Reviewer 2

Summary

Hierarchical clustering is commonly applied to two types of objects: 1. sets of points 2. graphs (in which case it is usually called hierarchical graph partitioning) What can be said about the statistical properties of hierarchical clustering? In case (1), we can look at the underlying density from which the points are sampled, define a suitable (infinite) "cluster tree" for this density and then assert that a particular hierarchical clustering procedure returns finite trees that converge to this cluster tree in some suitable sense. Recent work by Eldridge et al has used a criterion called "merge distortion" to assess the discrepancy between the target (infinite) tree and the tree estimated from a finite sample. Specific algorithms have been found to be consistent in the sense of having the right limit, and their rates of convergence have been determined. The present paper is interested in extending this methodology to case (2). The starting point is, necessarily, to specify some kind of underlying distribution over graphs. The authors make a very general choice: the underlying distribution is a "graphon". Briefly, this is a generative process over (infinite) graphs specified by a function W: [0,1]^2 --> [0,1]. To sample a graph with n nodes: -- Pick n values x1, .., xn at random from [0,1]. Here, xi is a latent feature associated with node i. -- Put an edge between nodes i and j with probability W(xi, xj). There has recently been quite a lot of interest in this generative process, motivated by the kinds of large graphs that are central to computational social science. The authors define a natural hierarchical clustering associated with a graphon: this is the obvious one, whose clusters are obtained by picking a threshold t, considering points x and x' "connected" if W(x,x') > t, and then identifying clusters with the resulting connected components. One technical complication is to state this definition in a way that is not derailed by sets of measure zero, and the authors do so by setting up an equivalence relation in which measurable sets are considered equivalent if their set difference has measure zero. With the limit object thus defined, the authors proceed to develop a notion of consistency. One complication is the latent variables: in any given graph, the latent variables associated with nodes are not specified in advance; thus they need to be marginalized out. This is the one key difficulty of case (2) (see above) that does not arise in case (1). The authors then use a notion of merge distortion (as in earlier work) to define consistency. Finally, they give an algorithm for estimating the graphon cluster tree that is provably consistent. It is based on first estimating the edge probabilities and then applying single linkage.

Qualitative Assessment

(See summary for main strengths of the paper) There is some related work that the authors should reference carefully: the paper of Steinwart on "Adaptive density level set clustering". Like the present paper, Steinwart takes great pains to define connectedness in a way that is resistant to sets of measure zero. Can the authors comment on how their approach differs? All in all, this is important, exciting, and novel work.

Confidence in this Review

3-Expert (read the paper in detail, know the area, quite certain of my opinion)


Reviewer 3

Summary

The paper introduces a 'definition' of hierarchical clustering for random graph families generated from graphons. Graphons are a natural notion of limit for sequences of dense graphs. The authors give sufficient conditions for a method to be statistically consistent in identifying a clustering and provide an algorithm for it (for simple graph families)

Qualitative Assessment

* Graphons only make sense for generating dense graphs. I'd suggest that the authors discuss this point as in practice one is often concerned with sparse graphs. To be fair this is also a problem with stochastic block-models (or many other models) if one demands perfect clustering. However, a significant area of study in SBMs is about studying sparse blockmodels, by relaxing the notion of clustering to only identifying a partition correlated with the hidden assignments. Overall the paper is well-written. However, given the delicate nature of some of the definitions and claims, some additional details would be helpful. More importantly, more intuitive explanation would be beneficial to the reader. - e.g., What are X_1 and X_2 in Definition 5? - What does a hierarchical collection of subsets mean? (Line 280)

Confidence in this Review

2-Confident (read it all; understood it all reasonably well)


Reviewer 4

Summary

The paper proposed a theoretical framework for studying hierarchical clustering over graphs. The framework is built on the notion of graphons, ie, assuming that the graph is a random variable sampled from a underlying graphon. The paper introduces the notion of graphon cluster tree to denote the ground-truth hierarchical clustering for the graphon, and introduces the notion of mergeon as a compact mathematical representation of the graphon cluster tree. It then introduces a notion of consistency for graph clustering algorithms, and gives sufficient conditions for this. Finally, it gives a practical algorithm satisfying the conditions.

Qualitative Assessment

A mathematical framework for hierarchical clustering on graphs is definitely an interesing problem. This paper proposed a nice and rigorous framework in this line. The framework is also general enough that it includes the stochastic block model, a recent hot topic for graph clustering. The framework is build on the notion of graphon, a relatively recent mathematical notion. Roughly, a graphon is a symmetric function from [0,1]^2 to [0,1], whcih can be viewed as the generalization of the weighted graph with infinite many nodes. In this paper, it is viewed as a way to general random graphs. The cluster tree is as expected the tree formed by the "maximal connected components (above different level)" of the function. The paper considers the basic consistency question: when the algorithm can recover the tree in the limit? It gives simple sufficient conditions on algorithms using density estimators, and showed that a previous density estimator satisfies such conditions. It will be interesting to check the performance of the algorithm proposed in this paper on sythentic and real world data.

Confidence in this Review

2-Confident (read it all; understood it all reasonably well)


Reviewer 5

Summary

The paper formalizes cluster trees for graphons and provides an algorithm that consistently estimates them.

Qualitative Assessment

Overall, an interesting and well-written paper. My main concern is whether clustering in the graphon model is well motivated: the sample points are drawn uniformly from [0,1]. If the communities are (unknown) high dimensional (i.e. d > 1) structures, how could one reasonably embed this into [0, 1]? Although the graphon model seems to supersede the stochastic block model, it is unclear to me how much more general it makes the graph clustering problem. There seems to have been a good amount of care in the technical details. It would have been nice to see some numerical experiments. For definition 1 (connectedness) on line 162, do you mean that A is disconnected if there exists an S s.t. W < \lambda a.e. on S X (A\S) rather than this being the case for all S? *** POST-REBUTTAL *** Unfortunately, it seems that I may not have the background in this area to constructively comment on the usefulness/impact of this work. There seems to be a good amount of recent interest in understanding graphons and their use in analyzing networks and the hierarchical clustering framework in this regime could be of interest. Although some of my concerns regarding the practicality of such a procedure remain, I will upgrade my score in this category and downgrade my confidence.

Confidence in this Review

1-Less confident (might not have understood significant parts)


Reviewer 6

Summary

A notion of consistency of clustering is developed for a large class of random graphs: graphon models. To this end they define the clustering of a graphon, this then yields newly introduced mergeons. These mergeons (which have infinite sample knowledge) are also evaluated on finite samples to establish a point of comparison to the whole mergeon. Since the empirical graphs' vertices have forgotten their original place in the graphon, consistency is measured over all possible embeddings of the vertices into [0,1] and weighted by the likelihood of the graph given this embedding. Finally consistency is proven for an adaptation of the recently introduced clustering by neighborhood smoothing.

Qualitative Assessment

The paper gives an interesting approach to consistency of graph clustering. It seems sound, well written, gives a good overview. However, the possible impact could be discussed much more, maybe giving more examples. According to the rebuttal, experiments will be added to the appendix. According to the paper, the condition used for consistency seems to be not well equipped for other algorithms. Is it possible to give a consistency result for estimators of P which are consistent in say mean squared error? Or can other algorithms be adapted to obtain consistency "in infinity-norm"? Obviously, the authors struggle heavily with the 8 page limit of NIPS and the paper turns out to be a kind of executive summary of their theory. It seems to be sound, yet often I would like a little bit more details to be able to assess the ramifications - without having to look into the appendix. Here are several of such questions: 187 in Definition 2: why is this countable? 204: Can you explain the reasons why this result is non-trivial? [Rebuttal clarified this] 242-244 Definition 4: Could this be also defined without a rho? 246: When is a graphon W also a mergeon for itself? You write later that the stochastic block model has this property. 372: "W has structure": is this an informal statement? At least I cannot find the corresponding definition. Algorithm I: Where are the differences to [18] Finally, some minor comments: 114: "...has the same distribution as G_{***k-1***} 163: "for every" => "there is a" (already changed in the arXiv version) 321: S not referenced here 327: Make the equality S=(x_1,...,x_n) clear in the definition of L_W(S|G) Bibliography: The articles from arXiv should be cited as such

Confidence in this Review

2-Confident (read it all; understood it all reasonably well)